# Neuronal Learning Analysis using Cycle-Consistent Adversarial Networks

## Abstract

Recent advances in neural imaging technologies enable high-quality recordings from hundreds of neurons over multiple days, with the potential to uncover how activity in neural circuits reshapes over learning. However, the complexity and dimensionality of population responses pose significant challenges for analysis. To cope with this problem, existing methods for studying neuronal adaptation and learning often impose strong assumptions on the data or model that may result in biased descriptions of the activity changes. In this work, we avoid such biases by developing a data-driven analysis method for revealing activity changes due to task learning. We use cycle-consistent adversarial networks (Zhu et al., 2017) to learn the unknown mapping between pre- to post-learning neuronal responses. To do so, we develop an end-to-end pipeline to preprocess, train, validate and interpret the unsupervised learning framework with calcium imaging data. We validate our method on two synthetic datasets with known ground-truth transformation, as well as on V1 recordings obtained from behaving mice, where the mice transition from novice to expert-level performance in a visual-based behavioural experiment. We show that our models can identify neurons and spatiotemporal activity patterns relevant to learning the behavioural task, in terms of sub-populations maximising behavioural decoding performance and task characteristics not explicitly used for training the models. Together, our results demonstrate that analysing neuronal learning processes with data-driven deep unsupervised methods can unravel activity changes in complex datasets.

## 1 Introduction

One of the objectives in computational neuroscience is to study the dynamics of neural processing and how neural activity reshapes when learning a task. A major hurdle in this endeavour was the difficulty in obtaining high-quality neural recordings of the same set of neurons across an extended period of learning a task (Stevenson & Kording, 2011; Lütcke et al., 2013; Dhawale et al., 2017). With the advent of modern neural imaging technologies, it is now possible to monitor a large population of neurons over days or even weeks (Williams et al., 2018a; Steinmetz et al., 2021), thus allowing experimentalists to obtain *in vivo* recordings from the same set of neurons across different learning stages.

Significant efforts have been put into extracting interpretable descriptions of how the responses reshape with experience, ranging from latent variable models and domain adaptation models to deep generative models. One popular approach is to apply dimensionality reduction methods, including Principal Component Analysis (PCA), Tensor Component Analysis (TCA), Gaussian Process Factor Analysis (GPFA), Gaussian Process Factor Analysis with Dynamical Structure (GPFADS), and Preferential Subspace Identification (PSID) (Cunningham & Byron, 2014; Williams et al., 2018b; Sani et al., 2021; Yu et al., 2009; Rutten et al., 2020), to learn a set of latent factors that describe experimental variables (Hurwitz et al., 2021a). Using Deep Neural Networks (DNNs) as latent dynamic models has become increasingly popular in recent years. Pandarinath et al. (2018) introduced a Variational Autoencoder (VAE) called Latent Factor Analysis via Dynamical Systems (LFADS) to learn the latent dynamics from single-trial spiking activities. Gao et al. (2016) adapted the framework to work with calcium imaging data in Poisson feed-forward neural network Linear Dynamical System (PfLDS). Hurwitz et al. (2021b) further extended LFADS with an additional linear

decoder to filter irrelevant behaviour dynamics which led to a better low-dimensional representation. In an adjacent setting, numerous works pose the neural analysis task as a domain adaption problem and utilise methods based on Generative Adversarial Networks (GANs), Canonical Correlation Analysis (CCA) and VAEs to learn an aligned latent representation of neural responses obtained over multiple recording sessions or days (Farshchian et al., 2018; Gallego et al., 2020; Jude et al., 2022). While these methods enabled substantial progress in understanding the structure of neuronal activity, they impose strong assumptions inherent in the modelling technique or the animal experiment, such as the linearity assumption in the linear latent variable models and the requirement of an experimental setup with structured trials. On the other hand, a fully data-driven method without strong assumptions about the experimental setup or response statistics could provide an unbiased view of the neural activity.

Fully data-driven methods are challenging because they require large numbers of samples for training. However, the number of trials in neuroscience datasets is typically small. This problem is exacerbated for experiments investigating learning where combinations of experience stages are of interest, e.g. novice and expert animals. Learning the transformation between these stages in a supervised manner would require a pairing of particular trials, compounding the need for large numbers of trials. Instead, we here explore the problem of learning the transformation from one stage of learning to another with unknown sample pairing as an unsupervised translation task. This way, we have more sample pairings at our disposal at the expense of being agnostic to particular trials. Moreover, since we do not take trial information into consideration, the method is applicable to experiments without a clear trial structure (e.g. experiments with freely roaming animals).

In this work, we use cycle-consistent adversarial networks (Zhu et al., 2017), or CycleGAN, to learn the mapping between pre- and post-learning neuronal activities in an unsupervised and data-driven manner. Such a transformation can be useful in follow-up studies to 1) identify neurons that are particularly important for describing the changes in the overall response statistics, not limited to first or second-order statistics; 2) detect response patterns relevant for changes from pre- to post-learning; 3) determine what experimental details are of particular interest for learning. To this end, our work includes the following contributions:

- We introduce two synthetic datasets with ground-truth transformation and one recorded dataset from the primary visual cortex (V1) of behaving mice to validate the method.

- We demonstrate that our method is able to identify neurons and response patterns from the neural recordings that are relevant to the behavioural task in a data-driven manner.

- We perform a decoding analysis on two behavioural variables and show that using the top-30 neurons learned by the models, can achieve similar decoding performance as using all neurons.

- We propose a novel neuron ordering method that can improve the learning performance of convolutional-based neural networks by pre-sorting the spatial order of the neurons as a preprocessing step.

Notations used in this manuscript are listed in Table A.1 for convenience.

## 2  Methods

In this section, we first formalise the problem we aim to tackle in this work, followed by the setup of the animal experiment which we use to obtain pre- and post-learning recordings, and descriptions of two synthetic datasets we introduce to validate our method. We then detail the unsupervised learning framework, as well as the preprocessing, training, and evaluation procedures.

### 2.1  Task setting

Our goal is to model the transformation between pre-learning and post-learning activities. Given two sets of neural recordings $X$ and $Y$ which correspond to the pre-learning and post-learning activities from the same behaving animal, one could learn the transformation between the two sets in a supervised manner, i.e. given a trial $i = 1$ in $x^i \in X$, learn model $G : X \to Y$ to minimise the error in $G(x^i)$ and $y^i \in Y$. However, due to trial-to-trial variability in neuronal responses (Carandini, 2004), as well as external factors that can

influence the recording session (e.g. small changes in lighting or level of attention by the animal), trial-to-trial pairings of $X$ and $Y$ tend to be noisy (i.e. we cannot ensure that trial $i = 1$ in $X$ corresponds to trial $i = 1$ in $Y$). Moreover, combinations of these factors are unlikely to occur multiple times, leading to a small number of samples for training our models. Instead, we investigate the problem of learning the $X \rightarrow Y$ transformation with unknown pairing as an unsupervised translation task. In other words, given the neural recordings of a novice animal, can we translate the responses that correspond to the animal with expert-level performance, and vice versa?

## 2.2 Recorded data

To record neuronal activities with pre- and post-learning responses, we conducted a virtual reality (VR) experiment[1] that follows a similar procedure as in Pakan et al. (2018) and Henschke et al. (2020). Briefly, a head-fixed mouse was placed on a linear treadmill that allows it to move forward and backward. A lick spout and two monitors were placed in front of the treadmill and a virtual corridor with a defined grating pattern was shown to the mouse. A reward (water drop) was available if the mouse licked within the predefined reward location in the virtual corridor (at 120 to 140 cm), in which a black screen is shown as a visual clue, and the mouse learns to utilise both visual information and self-motion feedback to maximise reward. Figure 1 illustrates the experiment setup. The same set of neurons in the primary visual cortex was labelled with the GCaMP6 calcium indicator and monitored throughout 5 days of experiments. The fluorescence signals were then decontaminated and extracted from the calcium imaging data using FISSA (Keemink et al., 2018), and the relative changes in fluorescence ($\Delta F/F_0$) over time were used as a proxy for an action potential. Four mice were trained in the experiment and all mice achieved expert-level performance within 4 days of training. For instance, Mouse 1 took on average $36\%$ less time to complete a trial with a $52\%$ improvement in the received rewards from day 1 to day 4. Trial information of Mouse 1 is shown in Table 1 (see Appendix C for Mouse 2 - 4). This dataset provides excellent insights into how cortical responses reshape with experience, and therefore, we utilise the recordings obtained on the 1st (pre-learning) and 4th day[2] and represent their distributions as $X_{\mathrm{rec}}$ and $Y_{\mathrm{rec}}$, respectively.

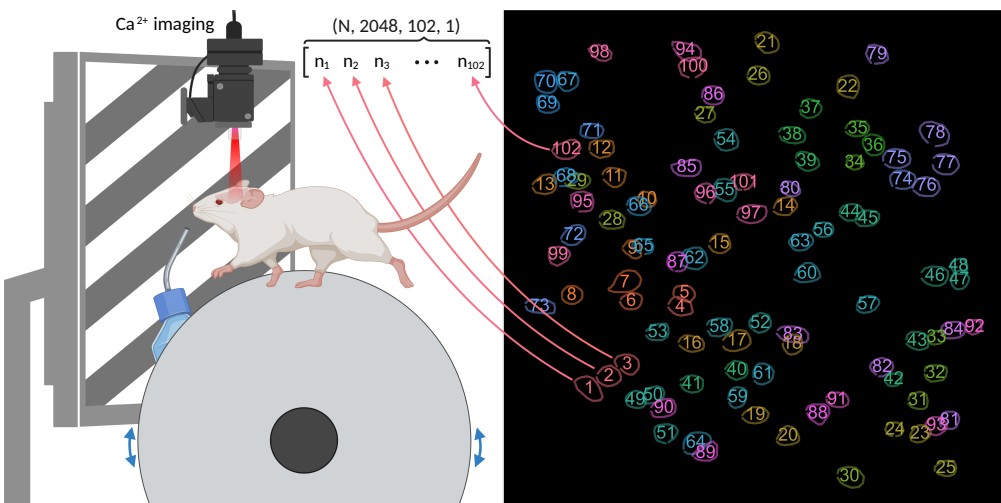

Figure 1: (Left) illustration of the mouse virtual-environment setup. A defined grating pattern is displayed on the monitors and the mouse can move forward and backward in the virtual corridor. When the mouse approaches the reward zone, which was set at 120 cm to 140 cm from the initial start point, the grating pattern would disappear and be replaced with a blank screen. If the mouse licked within the virtual reward zone, then a droplet of water was given to the mouse as a reward. Trials reset at 160 cm. The figure is based on Figure 1 in Pakan et al. (2018). (Right) original coordinates and annotation order of the 102 recorded neurons. i.e. neuron 1 here would be at index 0 in the data matrix, and neuron 65 would be at index 64.

---

[1]Data used in this work will be made publicly available upon acceptance.
[2]Mouse 2 and 3 performed worse on day 5, hence, we use the recordings obtained on day 4 for all mice.

Table 1: Mouse 1 trial information where 102 V1 neurons were monitored across 5 days of training, and the rodent achieved "expert" level by day 4 with a success rate of $> 75\%$ at the task.

| DAY | DURATION | NUM. TRIALS | AVG. TRIAL DURATION | LICKS | REWARDS |
|-----|----------|-------------|---------------------|-------|---------|
| 1 | 894.73s | 129 | 6.94s | 2813 | 140 |
| 2 | 898.68s | 177 | 5.08s | 2364 | 182 |
| 3 | 897.16s | 192 | 4.67s | 2217 | 198 |
| 4 | 898.45s | 203 | 4.43s | 1671 | 213 |
| 5 | 897.25s | 264 | 3.40s | 1298 | 327 |

## 2.3 Synthetic data

To evaluate our model in a setting with known ground truth, we introduce two synthetic datasets: a simulated dataset (Section 2.3.1) and an augmented dataset (Section 2.3.2). As these datasets contain known ground-truth pairing, they allow us to evaluate our method by directly comparing the transformation results with the synthetic labels. Moreover, the augmented dataset further enables us to inspect the transformation results visually.

### 2.3.1 Simulated data

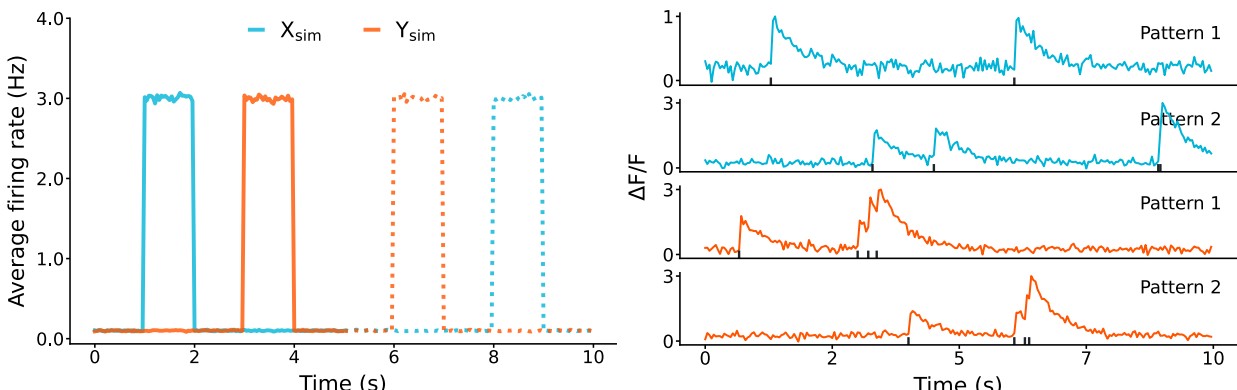

Figure 2: The (Left) average firing pattern of two simulated populations, (Blue) $X_{\text{sim}}$ and (Orange) $Y_{\text{sim}}$. Each population has two firing patterns (i.e. Pattern 1 and 2). For instance, neurons in $X_{\text{sim}}$ are high firing between 1 s to 2 s or 8 s to 9 s. (Right) Two randomly selected spike trains and their convolved calcium-like traces from the two simulated populations. Figure E.1 shows a sample population of $x_{\text{sim}}$ and $y_{\text{sim}}$.

We first simulate the neuronal responses of two populations of size 128. Each trial is sampled from a Poisson distribution with a trial duration of 10 s (i.e. 240 time-steps at 24 Hz) for a total of 1400 trials (i.e. about the same as the Mouse 1 recorded data in Section 2.5). In addition to the background activity at $\sim 0.1$ Hz, each population consists of two distinct firing patterns. Half of the neurons in the first population are highly active ($\sim 3$ Hz) from 1 s to 2 s, and the other half from 8 s to 9 s; whereas neurons in the second population are active from 3 s to 4 s and from 6 s to 7 s. We use an exponential onset and double decay function $f_{\text{Ca}}$, as described in Grewe et al. (2010), to obtain fluorescence-like traces from the spike times $t$:

$$f_{\text{Ca}}(t) = \begin{cases} 0 & \text{for } t \le t_0 \\ \left[ 1 - e^{-(t-t_0)/\tau_{\text{onset}}} \right] \left[ A_1 e^{-(t-t_0)/\tau_1} + A_2 e^{-(t-t_0)/\tau_2} \right] & \text{otherwise} \end{cases} \tag{1}$$

where $A_1$, $\tau_1$, $A_2$ and $\tau_2$ are the amplitude and decay time parameters for the first and second exponential decay; $\tau_{\text{onset}}$ is the action potential onset time. We then add Gaussian noise (signal-to-noise ratio of 10) to the convolved traces to improve realism. We denote the simulated responses from the two populations as $X_{\text{sim}}$ and $Y_{\text{sim}}$, and their overall firing patterns and samples of calcium-like traces are shown in Figure 2. Notably,

we shuffle the neuron index such that the traces appear less structured and thus increase the difficulty for the generators to learn the mapping between the two sets, an example of the shuffled populations is shown in Figure E.1. The simulated dataset allows us to compare the activity patterns of the transformed traces, $\hat{y}_{\text{sim}} = G(x_{\text{sim}})$ and $\hat{x}_{\text{sim}} = F(y_{\text{sim}})$, against the ground truth patterns.

### 2.3.2 Augmented data

In addition to testing the method's capability to learn the transformation between two neuron populations, we would also like to evaluate the method's capacity to recover or generate neural activity from unpaired data. To that end, we introduce an additional synthetic dataset – the augmented dataset $X_{\text{aug}}$ and $Y_{\text{aug}} = \Phi(X_{\text{aug}})$. We construct a handcrafted spatiotemporal transformation that introduces a clear triangular pattern to the day 1 recordings from the VR experiment $X_{\text{rec}}$. For a given neural recording $x \in X$ with shape $(N, H, W)$ where $N$ is the number of samples, $H$ is the time-steps and $W$ is the number of neurons, we define transformation $\Phi$ as:

$$M_{\text{aug}}^{H \times W} = \begin{cases} 0 & i \leq sj, \ i \in H, \ j \in W \\ 1 & \text{otherwise} \end{cases} \tag{2}$$

$$\Phi(x_{\text{aug}}) = M_{\text{aug}} x_{\text{aug}} + 0.25\eta \tag{3}$$

where $M_{\text{aug}}$ is a $H \times W$ matrix where values $(i, j)$, $i \leq sj$ are set to 0 and $s = 8$ is the number of time-steps to mask for each row (neuron). $\Phi$ replaces activities masked by $M_{\text{aug}}$ with Gaussian noise $\eta \in \mathcal{N}(\mu_x, \sigma_x^2)$ with the per-neuron mean and standard deviation of $x_{\text{aug}}$. Importantly, we shuffle the pairing of $X_{\text{aug}}$ and $Y_{\text{aug}}$ such that $x_{\text{aug}}^i \neq \Phi(y_{\text{aug}}^i)$ for any sample $i$ in the training set. This forces the model to learn from unpaired samples but allows us to measure the transformation error in $F(Y_{\text{aug}})$ and $G(X_{\text{aug}})$ against their ground-truth data $X_{\text{aug}}$ and $Y_{\text{aug}}$ using common distance metrics on the test set. An example of $x_{\text{aug}}$ and $y_{\text{aug}}$ is available in Figure 3 (example traces are available in later Sections). Such spatiotemporal augmentation, though biologically unrealistic, allows easy visual verification of the transformation learned by the generators. In addition, one would expect the models to focus on regions surrounding the masked area in $x_{\text{aug}}$ and $y_{\text{aug}}$, thus, allowing us to ensure that the attention gate modules and localisation maps function as intended.

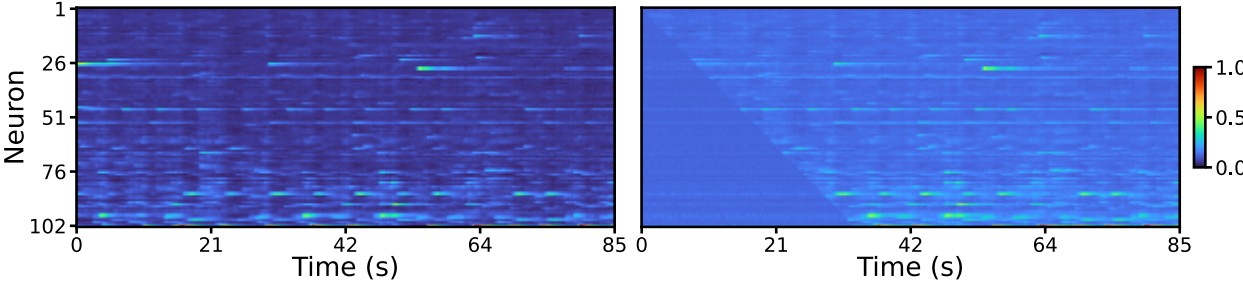

Figure 3: Example of a randomly selected segment from Mouse 1 recordings (Left) $x_{\text{aug}}$ and its corresponding (Right) augmented $y_{\text{aug}} = \Phi(x_{\text{aug}})$. Note that the bottom left corner in $y_{\text{aug}}$ has been masked with noise added to the segment. The `TURBO` colormap is applied to improve readability.

### 2.4 CycleGAN

In this section, we describe the cycle-consistent adversarial network (CycleGAN, Zhu et al. 2017) framework which is a core part of the unsupervised learning mechanism used in this work. Generative Adversarial Networks (GANs, Goodfellow et al. 2014) is a class of deep generative models that have shown promising results in synthesising neuronal activities that capture the low-level statistics of recordings obtained from behaving animals (Molano-Mazon et al., 2018; Ramesh et al., 2019; Li et al., 2020). CycleGAN is an unsupervised framework that utilises two GANs to learn the mapping between two unpaired distributions $X$ and $Y$ via cycle-consistency optimisation.

CycleGAN consists of four networks: generators $G : X \rightarrow Y$ and $F : Y \rightarrow X$ which learn the transformation from $X$ to $Y$ and vice versa; and discriminators $D_X : X \rightarrow [0, 1]$ and $D_Y : Y \rightarrow [0, 1]$ which learn to

distinguish if a given input is part of distribution $X$ and $Y$, respectively. In a forward cycle step ($X \rightarrow \hat{Y} \rightarrow \bar{X}$, illustrated in Figure B.1), we first sample $x \sim X$ and apply transformation $G$ to obtain $\hat{y} = G(x)$. We expect $\hat{y}$ to resemble data from the expert distribution $Y$. Hence $D_Y$ learns to minimise (4) $\mathcal{L}^{D_Y} = -\mathbb{E}_{y \sim Y}[(D_Y(y) - 1)^2] + \mathbb{E}_{x \sim X}[D_Y(G(x))^2]$. Similar to a typical GAN, generator $G$ learns to deceive $D_Y$ with the objective of (5) $\mathcal{L}^G = -\mathbb{E}_{x \sim X}[(D_Y(G(x)) - 1)^2]$. This formulation is the loss function of LSGAN (Mao et al., 2017) when the discriminator labels for generated and real samples are 0 and 1. However, $D_Y$ only verifies whether $\hat{y} \in Y$ but cannot ensure that $\hat{y}$ is the corresponding transformation of $x$. Moreover, since $X$ and $Y$ are not paired, we cannot directly compare $\hat{y}$ with samples in $Y$. To overcome this issue, another transformation is applied to reconstruct $\bar{x} = F(\hat{y})$, and if both transformations $G$ and $F$ are reasonable, then the distance between $X$ and $\bar{X} = F(G(X))$ should be small. Therefore, the generators also optimise this cycle-consistent loss (6) $\mathcal{L}_{\text{cycle}} = \mathbb{E}_{x \sim X}[\| x - F(G(x)) \|] + \mathbb{E}_{y \sim Y}[\| y - G(F(y)) \|]$. In addition, we would expect $\hat{x} = F(x)$ and $\hat{y} = G(y)$ to be in distributions $X$ and $Y$ given that $F : Y \rightarrow X$ and $G : X \rightarrow Y$. Hence we have the identity loss objective (7) $\mathcal{L}_{\text{identity}}^G = \mathbb{E}_{y \sim Y}[\| y - G(y) \|]$. Note that mean absolute error (MAE) was used as the distance function for both cycle-consistent and identity loss in the original work (Zhu et al., 2017).

Taken all together, $G$ optimises the following objectives: (8) $\mathcal{L}_{\text{total}}^G = \mathcal{L}^G + \lambda_{\text{cycle}} \mathcal{L}_{\text{cycle}} + \lambda_{\text{identity}} \mathcal{L}_{\text{identity}}^G$, where $\lambda_{\text{identity}}$ and $\lambda_{\text{cycle}}$ are hyper-parameters for identity and cycle loss coefficients. All four networks are trained jointly ($\mathcal{L}_{\text{total}}^F$ and $\mathcal{L}^{D_X}$ are the same as $\mathcal{L}_{\text{total}}^G$ and $\mathcal{L}^{D_Y}$ but in opposite directions). The framework has shown excellent results in a number of unsupervised translation tasks, including natural language translation (Gomez et al., 2018) and molecular optimisation (Maziarka et al., 2020), to name a few.

## 2.5 Model pipeline

We devised an end-to-end analysis pipeline[3] for calcium signals, including data preprocessing and augmentation, model training and interpretation, and, finally, evaluation of the translated calcium signals and also spike statistics. An illustration of the entire pipeline is shown in Figure 4.

As we want the models to identify patterns relevant to the animal experiment in a completely data-driven manner, we apply little preprocessing to the fluorescence signals. For the augmented and recorded dataset, we first segment the two datasets $X$ and $Y$ with a sliding window of size $H = 2048$ along the temporal dimension (around 85s in wall time). This preprocessing step ignores the trial information in the animal experiment and results in data with shape $(N, H, W)$ where $W$ is the number of neurons and $N$ is the number of segments. Note that the segmentation step is not needed for the simulated dataset since we already sample $N$ trials with shape $(H, W)$. In order to take advantage of the spatiotemporal information in calcium responses using 2-dimensional convolutional neural networks (CNNs), we further convert the segment to shape $(N, H, W, 1)$, effectively treating each segment as a grey-scale image. We normalise each set to the range $[0, 1]$, and divide the datasets into train-validation-test sets with a ratio of 70%:15%:15%.

In addition to the cycle-consistency and identity loss comparison metrics to validate the transformation performance, we also compare the first and second-order statistics of the translated responses $\hat{X} = F(Y)$ and $\hat{Y} = G(X)$ against $X$ and $Y$. To that end, we first infer spike trains from the fluorescence signals using Cascade (Rupprecht et al., 2021), the state-of-the-art spike deconvolution algorithm. We then compute the following spike train similarities and statistics: (1) mean firing rate for evaluating single neuron statistics; (2) pairwise Pearson correlation for evaluating pairwise statistics; (3) pairwise van Rossum distance (Rossum, 2001) for evaluating general spike train similarity. These quantities are evaluated across the whole population for each neuron or neuron pair and we compare the resulting distributions over these quantities obtained from the original and translated data: (a) $X \mid \hat{X} = F(Y)$, (b) $X \mid \bar{X} = F(G(X))$, (c) $Y \mid \hat{Y} = G(X)$ and (d) $Y \mid \bar{Y} = G(F(Y))$. We, therefore, validate the whole spatiotemporal first and second-order statistics as well as general spike train similarities.

Finally, we incorporate two visual explanation methods into our generator architecture and pipeline to uncover patterns in the neural activities, detailed in Section 2.6.1 and Section 2.7. We then evaluate the explainability techniques via a decoding analysis on selected behavioural variables from the VR experiment. Implementation details can be found in Section 2.8.

---

[3]The software codebase is attached as a supplementary file and it will be made publicly available upon acceptance.

All models were trained with the Adam optimiser (Kingma & Ba, 2014) for a maximum of 200 epochs where all models converged. With mixed precision training Micikevicius et al. (2017), the method takes on average 15 hours to fit on a single NVIDIA A100 GPU. We selected the hyper-parameters by a random search (Bergstra & Bengio, 2012) on the augmented dataset, and the same settings were then used in the other two datasets. The hyperparameter search space and final settings are shown in Table B.2 and B.3, respectively.

## 2.6    Networks architecture

In this section, we detail the generator and discriminator architectures. In this work, we propose two modifications to the generators from Zhu et al. (2017): (1) block residual connection – `Model-R`, and (2) attention-gated residual connection – `Model-AG`. Figure 5 illustrates the architecture of the `Model-AG`.

### 2.6.1    Generator

Given a normalised input $x$ with shape $(N, H, W, C)$, the original generator in Zhu et al. (2017) first compresses $x$ with 2 convolution blocks (`DS`$_1$ and `DS`$_2$), followed by 9 residual processing blocks (`RB`$_i$, $i \in \{1, 9\}$), then 2 up-sampling blocks (`US`$_1$ and `US`$_2$) to output a tensor with the same spatial-temporal dimension as the $x$, and finally, a convolution layer with $C$ filters and a kernel size of 1 and $C$ and sigmoid activation. Each down-sampling block (Blue box in Figure 5) uses a 2D strided convolution layer to reduce the spatiotemporal dimensions by a factor of 2, then followed by Instance Normalisation (InstanceNorm, Ulyanov et al. 2016), GELU (Hendrycks & Gimpel, 2016) activation and Spatial Dropout. Each up-sampling block has the same structure as the down-sampling blocks but with a transposed convolution layer instead. Each residual block consists of two convolution blocks with padding added to offset the dimensionality reduction and a skip connection that connects the input to the block with the output of the last convolution block via element-wise addition (Pink box in Figure 5).

As residual connections are known to improve gradient flow, especially in CNNs, thus mitigating the issue of vanishing gradients (He et al., 2016a;b; Huang et al., 2017), the first modification we made to the generator is to add residual connections between corresponding down-sampling blocks and up-sampling blocks, i.e. concat(`RB`$_9$, `DS`$_2$) $\rightarrow$ `US`$_1$ and concat(`US`$_1$, `DS`$_1$) $\rightarrow$ `US`$_2$. Concretely, in the original model, the last residual block `RB`$_9$ outputs $q$ which is then passed to `US`$_1$ directly and upsampled via transposed convolution. Now, $q$ is first concatenated with the output of `DS`$_2$ $a$ in the channel dimension and then passed to `US`$_1$, thus allowing better gradient flow in the increasingly deeper networks. This level-wise residual connection was first introduced and made popular in Ronneberger et al. (2015). We denote this generator architecture as `Model-R`.

The second modification we made to the generator is to adapt the additive attention gate (`AG`, Oktay et al. 2018) module into our residual connections. We denote this modification as `Model-AG`, which is illustrated in the yellow block in Figure 5. Given $q$ and $a$, the outputs of `RB`$_9$ and `DS`$_2$ respectively, `AG`$_1$ first apply a (separate) convolution with a kernel size of 1 and InstanceNorm for each variable, followed by an element-wise summation $s_{\text{AG}} = \text{InstanceNorm}(\text{CONV}_{1\times1}(q)) + \text{InstanceNorm}(\text{CONV}_{1\times1}(a))$ such that overlapping regions in $q$ and $a$ are amplified. We then apply ReLU (Nair & Hinton, 2010) to eliminate negative values in $s_{\text{AG}}$, then learn a sigmoid mask $m_{\text{AG}} = \text{Sigmoid}(\text{InstanceNorm}(\text{CONV}_{1\times1}(\text{ReLU}(s_{\text{AG}}))))$ via a convolution layer with a kernel size of 1 and InstanceNorm which has the same shape as $q$ and $a$. Finally, we apply the sigmoid mask $m_{\text{AG}}$ to $a$ and concatenate with $q$ in the channel dimension and pass its output to `US`$_1$. Conceptually, we learn $m_{\text{AG}}$ to eliminate irrelevant information in $a$ (i.e. representation closer to the input) with respect to $q$ (i.e. representation after a number of low-dimensional processing blocks). If all of $a$ is relevant, then this formulation works like `Model-R`. AG is easy to implement and can be a simple replacement for any block-wise residual concatenation layer. In addition, since $m_{\text{AG}}$ has the same shape as $a$ and $q$, we can overlay the sigmoid mask over either of the two variables and visualise the level of spatial-temporal attention learned by the AG module, thus improving the interpretability of the method.

### 2.6.2    Discriminator

As for the discriminator, we use a PatchGAN-based (Isola et al., 2017) architecture, as it provides more fine-grained discrimination information to the generators instead of the single value discrimination in the

discriminator in vanilla GAN. $D_X$ and $D_Y$ contain 3 down-sampling blocks where each block reduces the spatiotemporal dimension by a factor of 2, like the down-sampling blocks in the generators. For an input sample with shape ($H = 2048, W = 102, C = 1$), the discriminator outputs a sigmoid activated vector with shape $(256, 13, 1)$. Each element has a range $[0, 1]$ where a value closer to 1 suggests that the corresponding patch is from a real sample. The discriminators are kept relatively simple so that the generators would not be overpowered, especially in the initial phase of training.

## 2.7 Visual explanation

The AG modules in `Model-AG` operate in a low-dimensional latent space, which can be difficult to identify from fine-grained patterns in the neural activity. To address this shortcoming, we incorporated an additional technique for assessing feature importance and allowing further visual interpretation of the models. Gradient-weighted Class Activation Mapping (GradCAM), introduced by Selvaraju et al. (2017), is a post hoc algorithm that identifies region(s) in the input (i.e. a natural image) that a CNN classifier deems important. Briefly, GradCAM computes the gradient information flow between logits $y^c$ of class $c$ and the feature map $A^k \in \mathcal{R}^{u \times v}$ of a specific convolutional layer (usually the final layer) with $k$ filters. The gradients are then pooled over the spatial dimensions to calculate the unit importance weights (9) $\alpha_k^c = \frac{1}{N} \sum_i^u \sum_j^v \frac{\partial y^c}{\partial A_{i,j}^k}$ where $N = u \times v$. The GradCAM activation map is the weighted combination of the feature maps followed by ReLU activation to eliminate features with negative influence (10) $M_{\mathrm{GradCAM}} = \mathrm{ReLU}(\sum_k \alpha_k^c A^k)$. Similar to a sigmoid mask in the AG module (see Section 2.6.1), here, we can overlay the activation map $M_{\mathrm{GradCAM}}$ on top of the input to visually interpret region(s) that the model is focussing on. We applied the same method on the two discriminators to monitor the final convolution layer, and instead of a scalar prediction $y_c$, we compute the gradient flow between the feature maps of the target layer and the outputs.

## 2.8 Decoding analysis

The GradCAM visual explanation method make it possible to identify regions or neurons in the responses to which the models are more attentive. For instance, we expect the generators to focus on the activities surrounding the reward zone in the virtual corridor as the grating patterns disappear on the monitors. We hypothesise that a subset of neurons is more informative in the neuronal learning transformation and that the models should learn to be more attentive toward this group of neurons. To that end, we trained a regression model to decode behavioural variables (position and velocity) when provided with calcium responses of (1) all neurons, (2) top-30 (out of 102) neurons according to the activation maps, (3) rest of the neurons and (4) 30 randomly selected neurons. We kept the regression model fairly simple since we are only interested in the change in performance when training the model with different combinations of neurons. Here, we trained a recurrent neural network (RNN) decoder which consists of a GRU-layer (Cho et al., 2014) with 128 units followed by a fully-connected layer to output a scalar. The model was trained to optimise the mean-squared error (`MSE`) between its predictions and the behavioural variables using Adam (Kingma & Ba, 2014). Finally, for each variable-neuron combination, we fit the decoder 20 times, each with a different random seed, and compare their performance in terms of $R^2$ on the test set. If the selected neurons are indeed more influential, then in contrast to using (1) all neurons, we expect a drop in performance when the model is trained with the (3) rest of the neurons. Moreover, the decoding accuracy when provided with (2) top-30 neurons should be significantly different from (4) selecting 30 neurons randomly.

## 2.9 Neuron ordering

As discussed in Section 2.6, we are using convolutional-based networks for the generators and discriminators. It has been shown that CNNs with a smaller kernel can often perform as well or even better than models with larger kernels while maintaining fewer trainable parameters thus easier to train (He et al., 2016a; Li et al., 2021). Nevertheless, a small kernel can potentially limit the receptive field of the model, or the region in the input that the model is exposed to in each convolution step (Araujo et al., 2019). In addition, the recordings obtained from the VR experiment were annotated based on how visible the neurons were in the calcium image, rather than ordered in a particular statistical manner (see Figure 1). This could potentially restrict CNNs with a small receptive field to learn meaningful spatial-temporal information from the population

responses. To mitigate this issue, we propose a novel procedure to pre-sort $X$ and $Y$, such that neurons that are highly correlated or relevant are nearby in their ordering. A naive approach is to sort the neurons by their firing rate or pairwise correlation, where the neuron with the highest firing rate or the neuron that, on average, is most correlated to other neurons is ranked first in the input array. However, it is possible that not all high-firing neurons or the most correlated neurons are the most influential in the learning process. This calls for an automated and data-driven approach to rank neurons in a meaningful order. Deep autoencoders have shown excellent results in feature extraction and representation learning (Gondara, 2016; Wang et al., 2016; Tschannen et al., 2018), and we can take advantage of their unsupervised feature learning ability.

We employed a deep autoencoder `AE` which learns to reconstruct calcium signals in $X$ and $Y$ jointly. The `AE` model consists of three (encoder) down-sampling and (decoder) up-sampling convolution blocks, and a bottleneck layer of dimension $(256, 128)$. The down-sampling block consists of a 1D convolution layer followed by InstanceNorm, GELU activation, and Spatial Dropout, whereas a 1D transpose convolution is used in the up-sampling block instead. We fit the model by minimising the reconstruction loss (11) $\mathcal{L}_{AE} = \texttt{MSE}(X, \texttt{AE}(X)) + \texttt{MSE}(Y, \texttt{AE}(Y))$. Then we compute the per-neuron reconstruction error on the validation set and sort the neurons in ascending order. That is, we rearrange the order in the neuron ($3^{\text{rd}}$) dimension of the data matrix of shape $(N, H, W, C = 1)$ where $H = 2048$ and $W = 102$ for $X_{\text{rec}}$ and $Y_{\text{rec}}$, as shown in Figure 1. It is important to note that the proposed neuron ordering process is an optional data preprocessing step that allows 2D convolution-based models to take advantage of the spatial information presented in neuronal responses and is not mandatory for the rest of this work to function. We also compare our method against neurons ordered by their original annotation, as well as neurons ordered by their average firing rate, and pairwise correlation. Furthermore, to demonstrate that 2D convolution can indeed better learn the spatial structure in neuronal responses, we added a 1D variant of `Model-AG` (denote as `Model-AG-1D`) as a baseline model which applies 1D convolution over the temporal dimension of the data and disregards spatial information in the neural recordings.

### 2.10 Baseline models

Here, we introduce three simple models for baseline comparisons: (1) `Identity` model: the `Identity` model performs no operation on the input, i.e. $X = \texttt{Identity}(X)$. We expect the generators to translate $X$ and $Y$ to representations that are closer to $Y$ and $X$, respectively. (2) `Linear` model: the `Linear` model is a linear autoencoder that learns a low-dimensional representation from $X$ via principal component analysis (PCA), followed by a linear decoder to project the latent variable to $Y$. We convert the input responses $X$ to shape $(N \times H, W)$ and apply PCA with $N_{\text{PCA}}$ components that can explain $\sim 95\%$ of the variance over a single time frame. The linear decoder is then fit to minimise the reconstruction error between targets $Y$ and projected outputs $\hat{Y}$. (3) `VAE` model: the `VAE` model is a vanilla variational autoencoder (Kingma & Welling, 2013). Briefly, the encoder learns to approximate the distribution $q(z|x)$ where $z$ is a latent representation given input $x$. The decoder then samples from the latent distribution $p(z)$ and generates $p(y|z)$. The architectures of the encoder and decoder are fairly simple. The encoder consists of three 2D convolution layers with 32 filters, a kernel size of 4, a stride size of 2, and GELU activation then followed by a dense layer that outputs the mean and log-variance of the latent space $z$ with a latent dimension of 4. Almost mirroring the encoder, the decoder inputs the latent variable and outputs $\hat{y}$ with the same dimension as input $x$ via three 2D transpose convolution layers followed by a 2D convolution layers with a kernel size of 3 and a stride size of 1. `VAE` learns to maximise the evidence lower bound on the marginal likelihood $\log p(y)$ which can be expressed as minimising (12) $\mathcal{L}_{VAE} = \log p(y|z) + \log p(z) - \log q(z|x)$ (Kingma et al., 2019). We optimise `VAE` and the decoder of `Linear` using Adam (Kingma & Ba, 2014) with early stopping for a maximum of 200 epochs. For completeness, we also fit the three baseline models on the opposite transformation $Y \rightarrow X$.

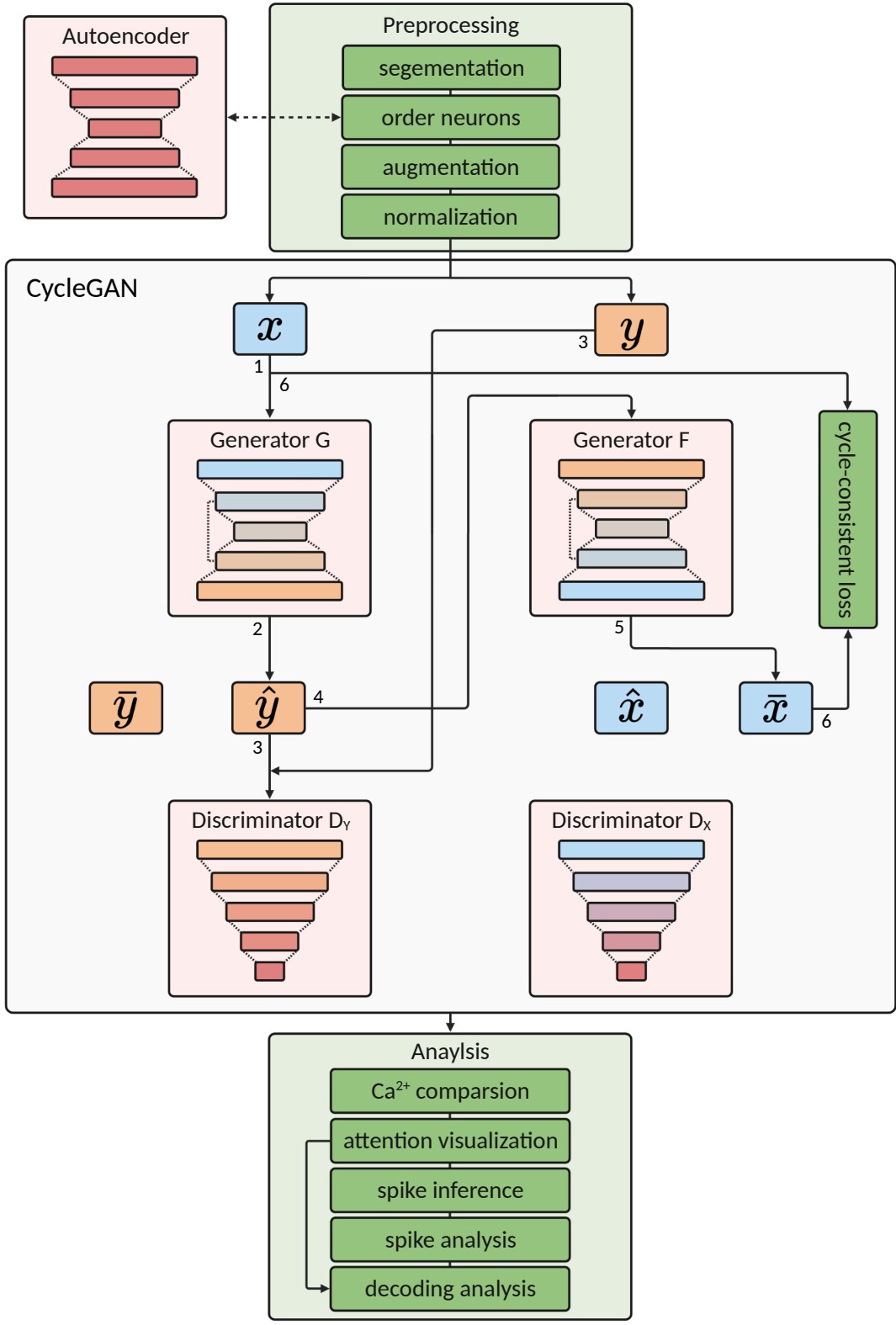

Figure 4: Illustration of the complete pipeline used in this work. Black directed lines represent the flow of data and the numbers indicate its order. Note that only the forward cycle step $X \to Y \to X$ is shown here for better readability.

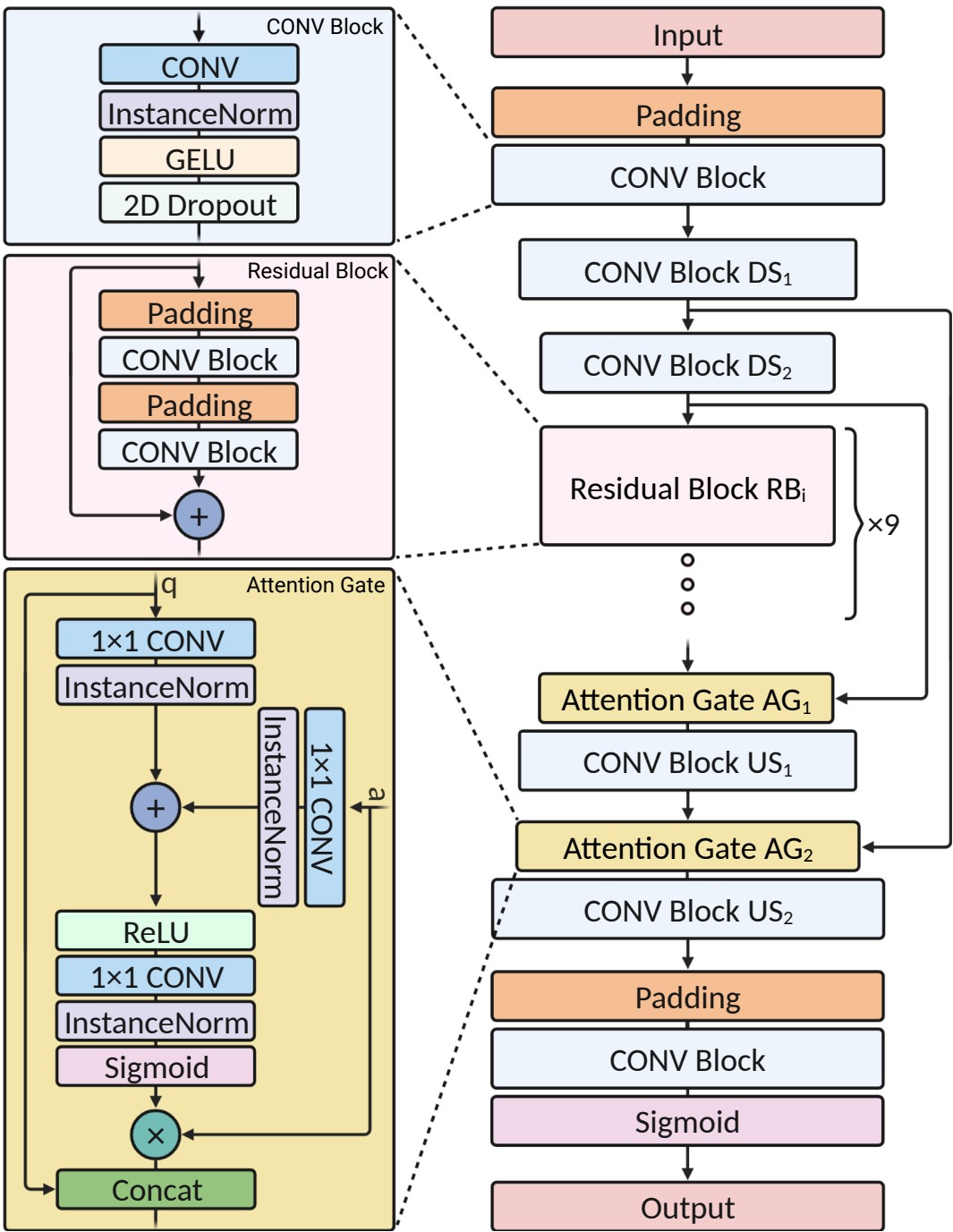

Figure 5: Architecture diagram of Model-AG generator. $+$ and $\times$ denote addition and element-wise multiplication respectively. Note that if we replace the AG module with a concatenation layer that concatenates $a$ and $q$ in the channel dimension, then architecture is the same as Model-R.

# 3 Results

The main objective of the framework is to identify a meaningful transformation between two neuronal datasets in a data-driven manner. We first assessed the framework's ability to learn the mapping between two data distributions using two synthetic datasets with known ground truth pairing. We then applied the same method to recorded data obtained from the primary visual cortex of behaving mice in the VR experiment.

## 3.1 Simulated data

We first validate our method on the simulated dataset to test its capability to learn the transformation between two calcium response populations with distinct activity patterns. In order to increase the difficulty of the mapping task, neurons in $X_{\text{sim}}$ and $Y_{\text{sim}}$ were shuffled jointly prior to model training (see Figure E.1). To quantify the transformation results, we compute the coefficient of determination $R^2$ between the average ground-truth and the average translated responses (Table 2). The baseline model `Linear` failed to capture the $X_{\text{sim}} \to Y_{\text{sim}}$ transformation with a $R^2 = 0.1738$ and is worse than the `Identity` model. `VAE` and `Model-AG` trained with the LSGAN objective, on the other hand, are able to learn the transformations very well with $R^2 \geq 0.9$ in both directions. The average activity patterns translated by `Model-AG` are shown in Figure 6.

Table 2: The $R^2$ between the average ground truth and average translated responses in the test set. The average responses of `Model-AG` and `Linear` are shown in Figure 6 and Figure E.2.

| MODEL | $X_{\text{SIM}}$ VS $F(Y_{\text{SIM}})$ | $Y_{\text{SIM}}$ VS $G(X_{\text{SIM}})$ |
|---|---|---|
| IDENTITY | 0.1994 | 0.1994 |
| LINEAR ($N_{PCA} = 115$) | 0.1738 | 0.1786 |
| VAE | 0.9401 | 0.9288 |
| MODEL-AG & LSGAN | 0.9680 | 0.9845 |

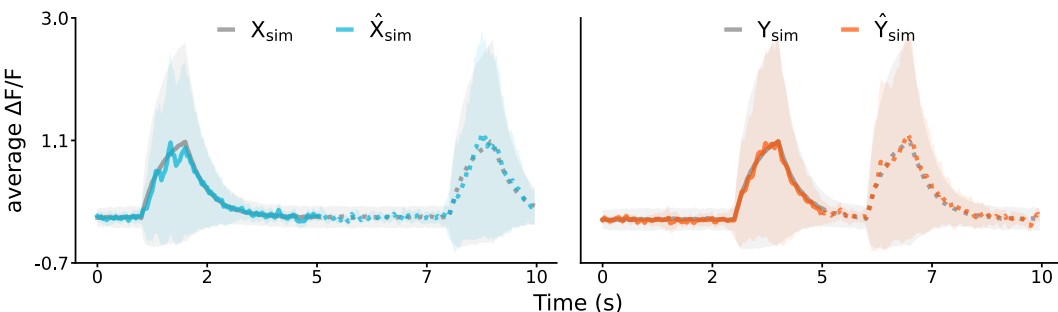

Figure 6: The average response patterns ($\Delta F/F$) in (Left) $\hat{X}_{\text{sim}}$ against $X_{\text{sim}}$, and (Right) $\hat{Y}_{\text{sim}}$ against $Y_{\text{sim}}$ by `Model-AG` trained with the LSGAN objective. The solid and dotted lines indicate the two activity patterns in each population, the grey and coloured lines correspond to the average simulated and translated responses, and the shaded areas show their variance. Table 2 shows the transformation results in terms of $R^2$.

## 3.2 Augmented data

Next, we fit our models on the augmented dataset, with known handcrafted augmentation, to show that our method is capable of learning subtle differences and recovering responses from unpaired calcium traces. Figure 9 shows calcium signals of the forward and backward cycle transformation from 3 neurons in a population, each with a different level of masked activities. Without paired samples, $G$ was able to learn the augmentation $\Phi$ and mask out the appropriate regions in $x_{\text{aug}}$, and conversely, $F$ was able to recover responses from the masked regions in $y_{\text{aug}}$. For instance, $F$ was able to recover the 4 spikes of fluorescence signals from the augmented input in Neuron 98 (bottom panel in Figure 9). Given that the difference between $X_{\text{aug}}$ and $Y_{\text{aug}}$ is replacing the responses of higher-index neurons with noise, we expect $D_Y$ to

discriminate based on activities around the augmentation region. The activation map of $D_Y(y_{\text{aug}})$, shown in Figure 8, indeed indicates a high level of attention along the edge of the diagonal region. On the other hand, since no augmentation was done on the input to $D_X$, the localisation map does not appear to have a particular structural area of focus. Interestingly, once we overlay the reward zones onto the input segment, we observed that the areas of focus learned by $D_X$ are loosely aligned with the reward zones. Note that reward zones are external task-relevant regions that are expected to shape the neural activity in V1 as the visual patterns change when the mouse enters the reward zone. These findings suggested that $D_Y$ learned to distinguish an input by predominantly focusing on the edge of the masking area, whereas $D_X$ learned distinctive patterns from higher-index neurons around the reward zones. We then inspected the AG modules learned by the two generators, shown in Figure 7. We observed that both generators ignore responses in the augmentation region, which is likely caused by the fact that information in that area is not relevant to either transformation. $G$ learns to replace the calcium traces from the to-be-masked region with noise, hence ignoring the information from that region; $F$ learns to recover responses in the masked region, as the information from the masked region is noise and hence not relevant for the model to learn from.

$X_{\text{aug}}$ and $Y_{\text{aug}}$ are paired in the test set, which enables us to directly evaluate the transformation result with common distance metrics, making this a good testbed to compare different generator architectures and neuron ordering methods. Results on the intermediate transformations $F(Y_{\text{aug}})$ and $G(X_{\text{aug}})$ are summarised in Table 3, and the complete results are available in Table F.1. Sensibly, all methods performed better in the $X_{\text{aug}} \xrightarrow{\Phi} Y_{\text{aug}}$ transformation as the models have to augment the responses with noise, whereas in the opposite direction, the models have to learn the reconstruction of the augmented activities. The baseline models `Linear` and `VAE` failed to learn the augmentation with results that are worse than `Identity`, i.e. $\texttt{MAE}(X_{\text{aug}}, \texttt{Linear}(Y_{\text{aug}})) = 0.2423$, $\texttt{MAE}(X_{\text{aug}}, \texttt{VAE}(Y_{\text{aug}})) = 0.2558$ and $\texttt{MAE}(X_{\text{aug}}, \texttt{Identity}(Y_{\text{aug}})) = 0.1328$; `Model-AG` trained with the LSGAN objectives, on the other hand, achieved $\texttt{MAE}(X_{\text{aug}}, \hat{X}_{\text{aug}}) = 0.1081$, outperformed the baseline models and as well as `Model-R` (i.e. $\texttt{MAE}(X_{\text{aug}}, \hat{X}_{\text{aug}}) = 0.1191$). Next, we evaluated different neuron ordering strategies. `Model-AG-1D`, which disregards spatial information in the data, performed noticeably worse than its 2D counterpart, suggesting that the spatial structure in the neural activities is indeed essential and learnable by the models. Overall, models trained on sorted neurons achieved better results with ordering neurons by `AE` reconstruction loss being the most performant.

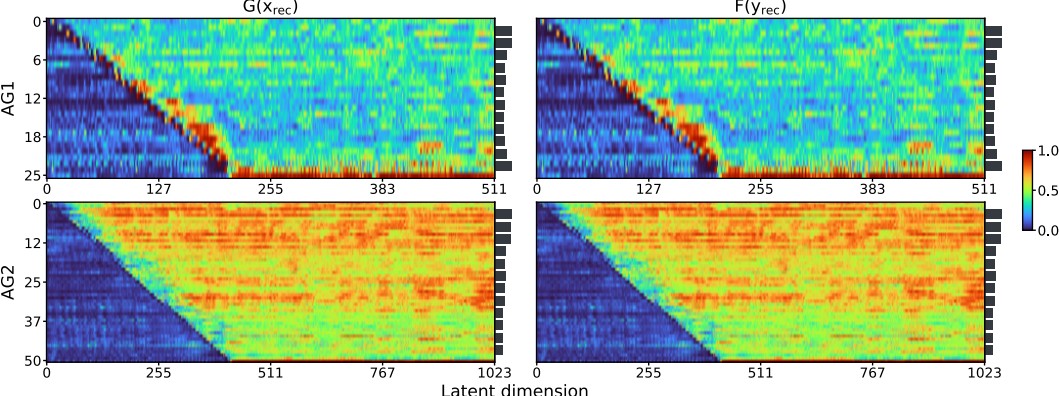

Figure 7: Attention masks $AG_1$ and $AG_2$ from `Model-AG` generators (Left) $G$ and (Right) $F$ where $AG_1$ and $AG_2$ learn the sigmoid attention masks with shape $(512, 26)$ and $(1024, 51)$, respectively. The attention masks showed that the generators learned to ignore the augmentation region in latent space. The histogram on the right of each panel shows the attention distribution over the $2^{\text{nd}}$ (i.e. neuron) dimension of the mask.

## 3.3 Recorded data

The results from the two synthetic datasets demonstrated the framework's capability in learning the unknown mappings in calcium traces. Moreover, the visual explanation methods indicated that the networks are indeed learning meaningful features. Next, we evaluated our method on the $(X_{\text{rec}})$ pre-learning and $(Y_{\text{rec}})$ post-

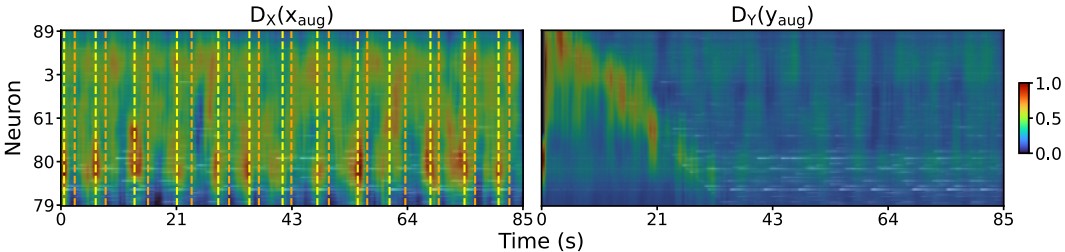

Figure 8: Activation maps of (Left) $D_X(x_{\text{aug}})$ and (Right) $D_Y(y_{\text{aug}})$ overlaid their respective input. As expected, $D_Y$ was attentive toward activities along the diagonal masking area as the most prominent feature of $y_{\text{aug}}$ is the augmentation region. Interestingly, $D_X$ focused on neuronal activities surrounding the reward zones (indicated by the yellow and orange vertical dotted lines). Neurons in `AE` order.

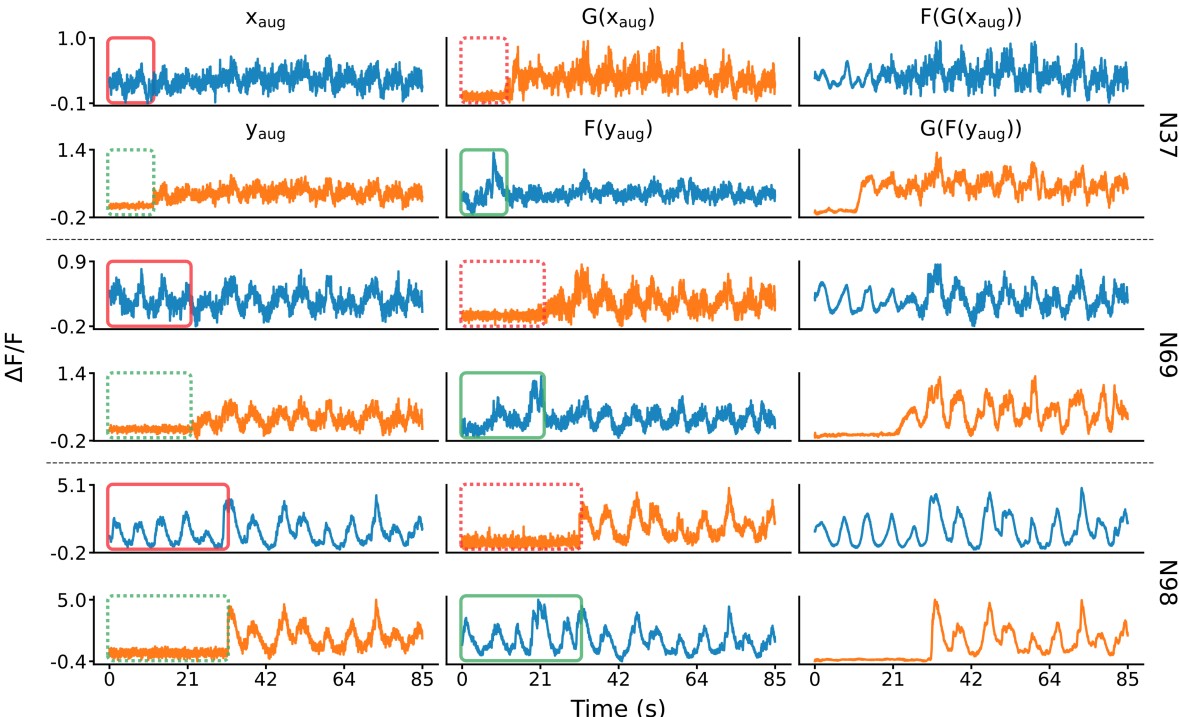

Figure 9: Forward and backward cycle steps of neurons 37, 69, and 98 (top, middle, and bottom of the population) from a randomly selected test segment. $G$ learns to perturb the initial part of the traces (i.e. from red solid box to red dotted box) and $F$ learns to reconstruct augmented regions in the traces (i.e. from green dotted box to green solid box). We expect the traces in the green solid box to resemble signals in the yellow solid box and the yellow dotted box in the green dotted box. Example transformation on the entire population is available in Figure F.1.

learning recordings obtained from mouse V1. Figure 10 shows the cycle transformation of 3 randomly selected neurons. Visually, $G$ and $F$ were able to reconstruct $\bar{x}_{\text{rec}} = F(G(x_{\text{rec}}))$ and $\bar{y}_{\text{rec}} = G(F(y_{\text{rec}}))$, and that the two generators were not simply passing through $x_{\text{rec}}$ and $y_{\text{rec}}$ in an intermediate step to translate $\hat{y}_{\text{rec}} = G(x_{\text{rec}})$ and $\hat{x}_{\text{rec}} = F(y_{\text{rec}})$. We also compared the cycle-consistent and identity loss to get a better sense of the transformations, shown in Table G.1. The `Model-AG` model trained with the LSGAN objective and `AE` neuron orders achieved a cycle-consistent loss of `MAE`$(X_{\text{rec}}, F(G(X_{\text{rec}}))) = 0.0733$ and `MAE`$(Y_{\text{rec}}, G(F(Y_{\text{rec}}))) = 0.0737$, and identity losses for `MAE`$(Y_{\text{rec}}, G(Y_{\text{rec}})) = 0.0101$ and `MAE`$(Y_{\text{rec}}, G(Y_{\text{rec}})) = 0.0069$, which are significantly better than the identity `MAE`$(X_{\text{rec}}, Y_{\text{rec}}) = 0.3674$. The low identity loss also suggests that the generators can correctly identify whether or not the given input is already in their respective target distributions, hence

Table 3: Transformation errors (`MAE`) in the augmented test set. FR, CORR and AE indicate neurons ordered by firing rate, pairwise correlation and `AE` reconstruction loss, respectively. Full results in Table F.1.

| | $X_{\text{AUG}}$ vs $F(Y_{\text{AUG}})$ | $Y_{\text{AUG}}$ vs $G(X_{\text{AUG}})$ |
|---|---|---|
| IDENTITY | $0.1328 \pm 0.2103$ | $0.1328 \pm 0.2103$ |
| LINEAR $(N_{PCA} = 40)$ | $0.2423 \pm 0.4189$ | $0.2583 \pm 0.4388$ |
| VAE | $0.2558 \pm 0.3885$ | $0.2437 \pm 0.4061$ |
| MODEL-R & LSGAN | $0.1191 \pm 0.2204$ | $0.1268 \pm 0.2294$ |
| MODEL-AG & LSGAN | $0.1081 \pm 0.1816$ | $0.1175 \pm 0.2008$ |
| MODEL-AG-1D & LSGAN | $0.2313 \pm 0.3065$ | $0.2372 \pm 0.3749$ |
| MODEL-AG & LSGAN & FR | $0.1043 \pm 0.2675$ | $0.1154 \pm 0.2164$ |
| MODEL-AG & LSGAN & CORR | $0.1030 \pm 0.1870$ | $0.1193 \pm 0.3396$ |
| MODEL-AG & LSGAN & AE | $0.1024 \pm 0.1856$ | $0.1103 \pm 0.2858$ |

performing no operation. Again, as demonstrated in Section 3.2, `Model-AG-1D` performed measurably worse than 2D convolutional-based models. In all cases, models trained with neurons ordered by `AE` reconstruction error achieved the best losses, and ordering neurons in any meaningful way brings measurable improvements.

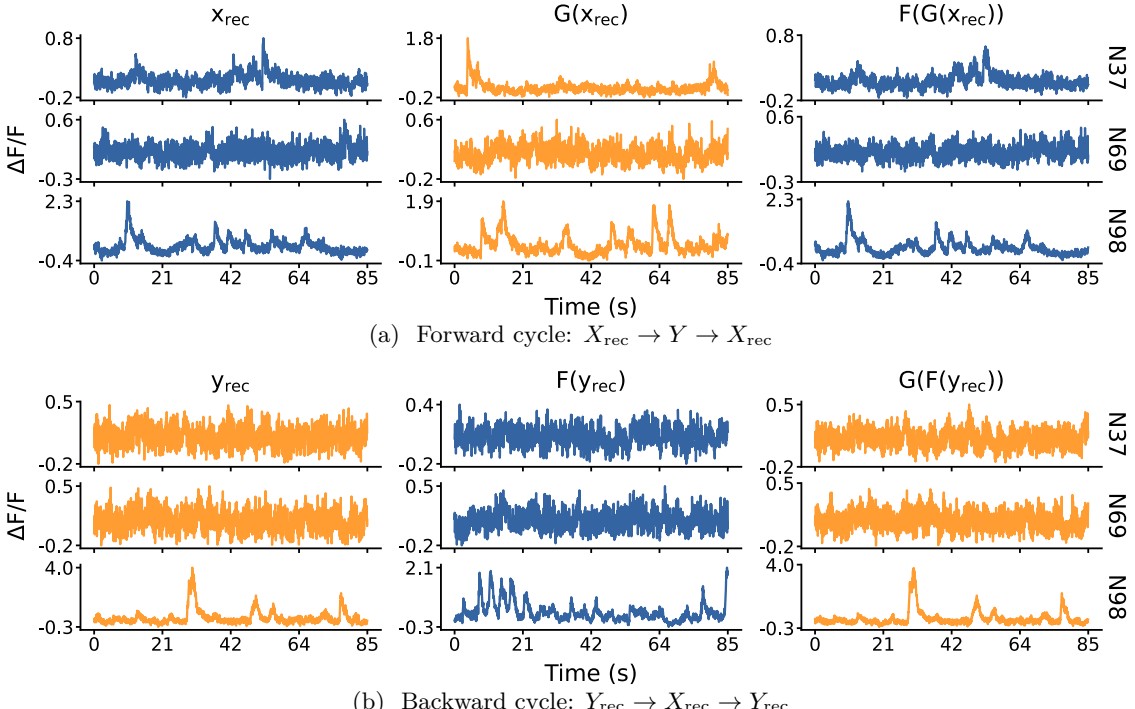

(a) Forward cycle: $X_{\text{rec}} \to Y \to X_{\text{rec}}$

(b) Backward cycle: $Y_{\text{rec}} \to X_{\text{rec}} \to Y_{\text{rec}}$

Figure 10: (a) forward and (b) backward cycle of neurons 37, 69, and 98 from a randomly selected test sample. The transformation of the entire population is available in Figure G.1.

As the recorded data are not paired on a trial-by-trial basis, we cannot compare $X_{\text{rec}}$ with $\hat{X}_{\text{rec}} = F(Y_{\text{rec}})$ nor $Y_{\text{rec}}$ with $\hat{Y}_{\text{rec}} = G(X_{\text{rec}})$ directly. In order to verify that the two intermediate steps are reasonable transformations, we instead compared three commonly used spike train statistics: (a) pairwise correlation, (b) firing rate, and (c) pairwise van Rossum distance. We expected that the distribution of the translated data resembles their corresponding recorded data, i.e. $X_{\text{rec}} \sim F(Y_{\text{rec}})$ and $Y_{\text{rec}} \sim G(X_{\text{rec}})$. In addition, since each spike metric is represented as a distribution over neurons or samples, we quantified the transformation performance by measuring the average KL divergence of each distribution pair. The distribution comparison for each metric is available in Section G.2.1. Table 4 summarises the average KL divergence of the three spike

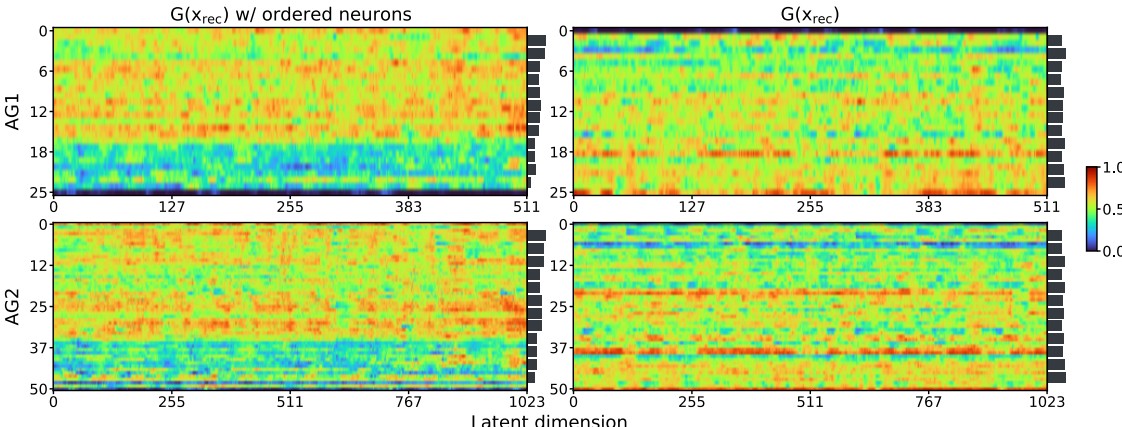

Figure 11: Attention masks $AG_1$ and $AG_2$ from `Model-AG` generator $G$ with (Left column) neurons sorted by `AE` reconstruction error and (Right column) no neuron ordering. The histogram to the right of each panel indicates the spatial attention (i.e. 2nd dimension) intensity. When the responses were ordered, a higher level of attention toward the neurons at the top (i.e. those with lower reconstruction loss) is very prominent across all attention modules, in contrast to the attention distribution being uniform across neurons when neurons were not ordered. The same behaviour is observed in generator $F$, shown in Figure G.2. Note that $AG_1$ and $AG_2$ learn the sigmoid attention masks with shape $(512, 26)$ and $(1024, 51)$, respectively.

statistics in the intermediate transformation step. Overall, models trained with the CycleGAN framework vastly outperformed the baseline models: `Linear` and `VAE` failed to capture the first and second order statistics of the target distributions (Figure G.7 shows the distribution comparisons of $X_{\mathrm{rec}}$ and $\mathtt{VAE}(Y_{\mathrm{rec}})$). On the other hand, `Model-AG` trained with the LSGAN objectives and `AE` neuron order is able to translate the responses with firing rate distributions of $\hat{X}_{\mathrm{rec}}$ and $\hat{Y}_{\mathrm{rec}}$ closely matching $X_{\mathrm{rec}}$ and $Y_{\mathrm{rec}}$, with average KL value of 1.1648 and 1.0697, respectively. When evaluated on pairwise correlation, the model achieved a low KL divergences of $\mathrm{KL}(X_{\mathrm{rec}}, \hat{X}_{\mathrm{rec}}) = 0.0479$ and $\mathrm{KL}(Y_{\mathrm{rec}}, \hat{Y}_{\mathrm{rec}}) = 0.0493$. We expected that each neuronal activity in $\hat{X}_{\mathrm{rec}}$ should resemble a corresponding response in $X$. We, therefore, computed the van Rossum distance between $X$ and $\hat{X}_{\mathrm{rec}}$ for each neuron across the test set, which can be represented in the form of a heatmap. We observed a clear diagonal line of low-intensity values in the heatmaps for most neurons (e.g. Figure G.5 and G.6 for $G$ and $F$). To summarise the spike similarity results, we also computed the KL divergence of pairwise van Rossum distance distributions with $\mathrm{KL}(X_{\mathrm{rec}}, \hat{X}_{\mathrm{rec}}) = 0.2387$ and $\mathrm{KL}(Y_{\mathrm{rec}}, \hat{Y}_{\mathrm{rec}}) = 0.3031$. The spike statistics indicated that the generators can indeed learn the transformation from pre- and post-learning activities, and vice-versa, and translated results can closely match the first and second-order statistics of their corresponding recorded data. We obtained similar results when fitting the same models on data recorded from Mouse 2, 3, and 4 (see Sections G.3, G.4 and G.5).

In the previous section, we were able to identify and interpret the learned features in a relatively straight-forward manner due to the systematic augmentation we introduced into the data. However, visualising and interpreting the attention maps on pre- and post-learning data is more challenging as there are no obvious patterns in the inputs to anticipate. Nevertheless, we expected a higher level of activities in the V1 neurons when the mouse was about to enter or was inside of the reward zone, where the grating pattern on the virtual walls turned blank. Subsequently, the models should learn meaningful features from responses surrounding the reward zones. Figure 11 shows the learned `AG` attention masks of $G$ and $F$ superimposed on the latent inputs. When the neurons were ordered, either by firing rate or autoencoder, we observed that the generators allocate more attention toward neurons that rank higher. This suggests that by grouping neurons in a meaningful manner, the convolutional layers in the generators can extract relevant features more effectively as compared to when neurons were randomly ordered. The spike analysis showed that ordering neurons in a structured manner does indeed yield better results across the board. In most cases, ordering the neurons according to the reconstruction error achieved the best results.

Table 4: The average KL divergence between recorded ($X_{\text{rec}}$ and $Y_{\text{rec}}$), translated ($\hat{X}_{\text{rec}} = F(Y_{\text{rec}})$ and $\hat{Y}_{\text{rec}} = G(X_{\text{rec}})$) distributions in three spike statistics. FR, CORR and AE denote neurons ordered by firing rate, correlation and AE reconstruction error, respectively. The complete results are available in Table G.2.

| | $X_{\text{REC}}$ VS $F(Y_{\text{REC}})$ | $Y_{\text{REC}}$ VS $G(X_{\text{REC}})$ |
|---|---|---|
| (A) PAIRWISE CORRELATION | | |
| IDENTITY | $0.0875 \pm 0.0549$ | $0.0821 \pm 0.0471$ |
| LINEAR ($N_{PCA} = 40$) | $2.9071 \pm 0.3869$ | $4.4880 \pm 0.6209$ |
| VAE | $1.1532 \pm 0.3885$ | $2.2670 \pm 0.4557$ |
| MODEL-AG-1D & LSGAN | $0.2027 \pm 0.1040$ | $0.1901 \pm 0.1003$ |
| MODEL-AG & LSGAN | $0.0552 \pm 0.0419$ | $0.0583 \pm 0.0553$ |
| MODEL-AG & LSGAN & FR | $0.0507 \pm 0.0358$ | $0.0504 \pm 0.0438$ |
| MODEL-AG & LSGAN & CORR | $0.0539 \pm 0.0329$ | $0.0534 \pm 0.0474$ |
| MODEL-AG & LSGAN & AE | $0.0479 \pm 0.0372$ | $0.0493 \pm 0.0448$ |
| (B) FIRING RATE | | |
| IDENTITY | $8.0705 \pm 6.5500$ | $7.7781 \pm 6.7338$ |
| LINEAR ($N_{PCA} = 40$) | $19.5507 \pm 1.6473$ | $19.5703 \pm 1.3806$ |
| VAE | $13.4533 \pm 6.6228$ | $15.9878 \pm 5.6384$ |
| MODEL-AG-1D & LSGAN | $3.5688 \pm 3.8895$ | $3.0572 \pm 3.1114$ |
| MODEL-AG & LSGAN | $1.5401 \pm 1.2491$ | $1.8527 \pm 1.3563$ |
| MODEL-AG & LSGAN & FR | $1.3402 \pm 1.0450$ | $1.6994 \pm 1.4170$ |
| MODEL-AG & LSGAN & CORR | $1.4006 \pm 1.1079$ | $1.4088 \pm 1.0828$ |
| MODEL-AG & LSGAN & AE | $1.1648 \pm 0.7934$ | $1.0697 \pm 0.7689$ |
| (C) PAIRWISE VAN ROSSUM DISTANCE | | |
| IDENTITY | $0.5510 \pm 0.2960$ | $0.3053 \pm 0.1211$ |
| LINEAR ($N_{PCA} = 40$) | $2.0803 \pm 0.2934$ | $3.4271 \pm 0.4390$ |
| VAE | $0.7336 \pm 0.2057$ | $1.9896 \pm 0.5960$ |
| MODEL-AG-1D & LSGAN | $0.3613 \pm 0.1597$ | $0.3764 \pm 0.1565$ |
| MODEL-AG & LSGAN | $0.2790 \pm 0.2186$ | $0.3216 \pm 0.1352$ |
| MODEL-AG & LSGAN & FR | $0.2539 \pm 0.1708$ | $0.3080 \pm 0.1173$ |
| MODEL-AG & LSGAN & CORR | $0.2629 \pm 0.1877$ | $0.2953 \pm 0.1230$ |
| MODEL-AG & LSGAN & AE | $0.2387 \pm 0.1488$ | $0.3031 \pm 0.1138$ |

We then inspected the activation maps of the discriminators, shown in Figure 12. Similar to the activation map of $D_X$ obtained from the synthetic dataset, we observed that both discriminators $D_X$ and $D_Y$ focused on activities surrounding the reward zones in the VR experiment. Nevertheless, the activation patterns shown in the figures are not that clear due to having multiple trials in the segment. We, therefore, derived a procedure to allow better visualisation of the relationship between activation patterns in the Model-AGnd the virtual position (0 to 160 cm) of the rodent. With $D_X$, for instance, we first extract the activation map for each test sample in $X_{\text{rec}}$. Then at each virtual position (in cm), we aggregate the activation value for each neuron over all the activation maps, resulting in a matrix with shape ($W = 102, 161$). Figure 12 shows the normalised positional activation maps for $D_X$ and $D_Y$. The only objective of the discriminators was to distinguish if a given sample was from a particular distribution, and thus the discriminators could have learned trivial features. Instead, $D_X$ focused on a specific group of neurons from 120 to 150 cm, which coincides with the end of the reward zone. Moreover, $D_Y$ learned to focus on a broader group of neurons with activation patterns that were also in alignment with the reward zone. Similarly, we could extract these positional attention maps for $G$ and $F$ following the same procedure, where we monitor the gradient flow between the convolution layer in the RB$_9$ and the output. Likewise, both generators focused on responses towards the middle and end of the reward zone, with $G$ concentrated on a very small subset of neurons.

This suggests that to learn the transformation from post- to pre-learning responses, the activities the mouse exhibits as it approaches the reward zone were deemed more important by the networks.

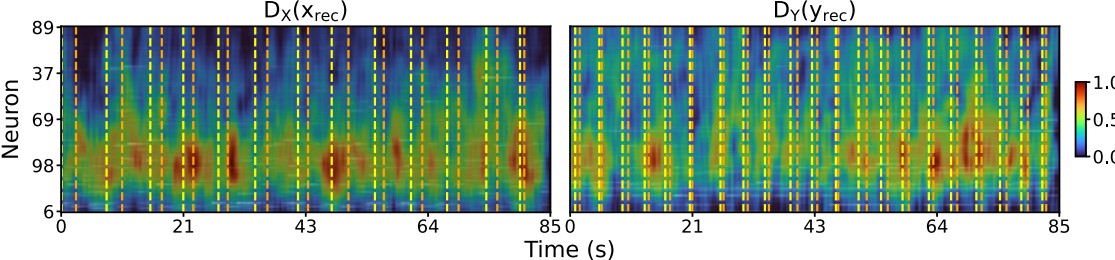

Figure 12: Activation maps of (Left) $D_X(x_{rec})$ and (Right) $D_Y(y_{rec})$ overlaid on their respective inputs. Without providing any trial-relevant information to the models, the discriminators were able to pick up information related to the reward zones (yellow and orange vertical dotted lines). Neurons in AE order.

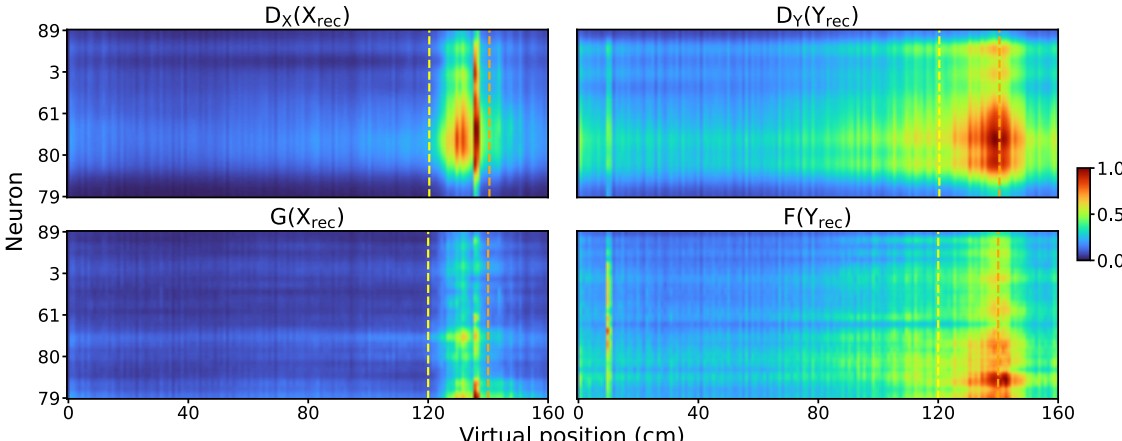

Figure 13: Positional activation maps of the discriminators and generators with respect to the virtual position. Overall, the models were focusing on a subset of responses surrounding the reward zone (yellow and orange dotted lines) in their discrimination and transformation processes. Neurons in AE order.

### 3.3.1 Decoding performance

The positional attention maps highlighted localised neurons in the responses with respect to the virtual position. To investigate our hypothesis that these neurons are more influential in the visual experiment and, subsequently, important for learning, we evaluated the decoding performance (i.e. position and velocity) with different subsets of neurons: (1) all neurons, (2) top-30 neurons, (3) rest of the neurons, and (4) 30 randomly selected neurons (see Section 2.8). Since $G$ and $F$ input $X_{rec}$ and $Y_{rec}$ respectively, the extracted positional activation maps represent different recording sessions. We therefore separately computed the top-30 neurons to decode behaviour variables from Day 1 and Day 4 of the recording. Note that we are interested in the relative change in performance when providing different combinations of neurons, rather than the overall decoding accuracy. Figure 14 shows the decoding results on the two behavioural variables from Day 1 and Day 4. Overall, we observed a substantial drop in performance when the top-30 neurons were removed. In addition, the models trained with the top-30 neurons outperformed the models with 30 randomly selected neurons, with the exception of decoding the mouse velocity on Day 1 recordings, though all decoders performed poorly in this specific task. Our decoding analysis thus shows that the neurons identified in the positional activation maps are more informative in the behavioural decoding tasks.

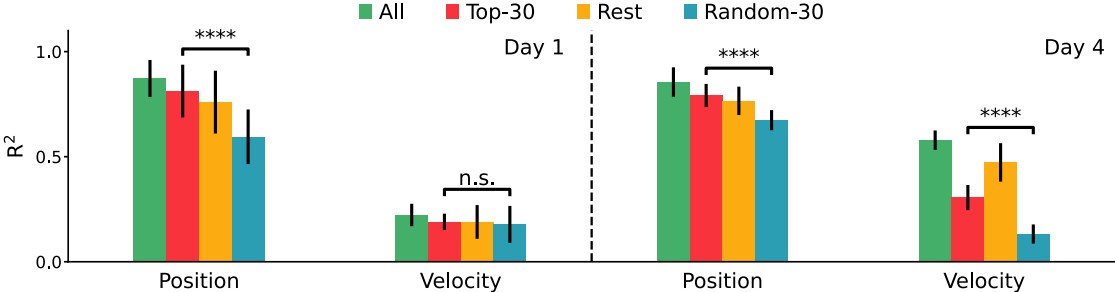

Figure 14: The decoding performance ($R^2$) of (a) virtual position and (b) velocity on Mouse 1 recordings when provided with: (1) all neurons, (2) top-30 neurons, (3) rest of the neurons and (4) 30 random neurons. The decoding accuracy values are listed in Table G.3.

## 4    Discussion

We demonstrated that the CycleGAN (Zhu et al., 2017) framework is a capable data-driven method to model the unknown transformation between pre- and post-learning neural responses. We evaluated our method on two synthetic datasets with known ground-truth transformations, and on V1 recordings obtained from behaving animals. We showed that our method significantly outperforms the baseline models on the recorded dataset. In addition, the AG module and GradCAM visualisation techniques provided the means to inspect region(s) of interest in the calcium responses learned by the generators and discriminators. Finally, we introduced a novel and easy-to-implement neuron ordering method that can assist convolutional-based models to learn better representations of neural data.

Intriguingly, without providing trial information in the training process, the models self-identified activities surrounding the reward zone in the VR experiment to be highly influential. This result aligns with our understanding from previous studies where responses in the visual cortex are shaped by the change in visual cues (Pakan et al., 2018; Henschke et al., 2020). Moreover, the behaviour decoding result showed that a subset of neurons, self-identified by the generators, achieved performance on par with when all neurons were provided, and performed significantly better than randomly selected neurons (see Section 3.3.1). This demonstrates the effectiveness of our method to self-identify neurons that are relevant to the task in contrast to previous works in identifying neuron contribution in decoding accuracy which requires manual iteration over neurons (Montijn et al., 2014). Interestingly, when analysing the firing rate and pairwise correlation of this subset of neurons, we did not notice any significant distinction in their statistics as compared to the rest of the population, suggesting that the models have learned features that cannot be easily captured or identified.

### 4.1    Limitations

The fully data-driven property of the proposed framework also comes with a number of limitations. First and foremost, as with most DNNs, this framework requires a significant amount of data, across the number of trials, the duration of each trial, and the number of neurons. For instance, we experimented with using fewer neurons, such as $W = 4$ or $W = 8$, and there was a significant decrease in performance.

Another notable constraint in our method is the fundamental one-to-one mapping limitation in the CycleGAN framework. The generators learn a deterministic mapping between the two domains and only associate each input with a single output. However, most cross-domain relationships consist of one-to-many or many-to-many mappings. More recently proposed methods, such as Augmented CycleGAN (Almahairi et al., 2018), aim to address such fundamental limitations by introducing auxiliary noise to the two distributions, and are thus able to generate outputs with variations. Nevertheless, these methods are most effective when trained in a semi-supervised manner which is not possible with our unpaired neural activity.

Lastly, a significant portion of the neuronal activity validation in Section 3.3 was performed in spike trains inferred from the recorded and generated calcium responses using Cascade (Rupprecht et al., 2021), which is

a recently introduced method that has outperformed the existing model-based algorithms. However, reliable spike inference from calcium signals remains an active area of research (Theis et al., 2016). For instance, Vanwalleghem et al. (2020) demonstrated that spiking activities could be missed due to the implicit non-negativity assumption in calcium imaging data which exists in many deconvolution algorithms, including Cascade. Nonetheless, Cascade was used to deconvolve calcium signals for all datasets and thus all inferred spike trains were subject to the same bias.

## 4.2 Conclusion

The CycleGAN framework's capacity to self-identify patterns in neural recordings makes it useful for analysing large-scale and unstructured animal experiments. As interpretability techniques for deep unsupervised models progress, and neural recordings in different learning phases from behaving animals have become readily available, there is potential for novel insights into fundamental learning mechanisms.

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

# A    Appendix

Table A.1: List of notations and their descriptions used in this manuscript.

| Term | Colour | Description |
|---|---|---|
| $N$ | | The number of segments. |
| $H$ | | The duration (in time-steps) of each segment. |
| $W$ | | The number of neurons. |
| $X, Y$ | | Two unpaired data distributions. |
| $G, F$ | | generators that learn the mapping of $X \to Y$ and $Y \to X$. |
| $D_X, D_Y$ | | Discriminators that learn to distinguish if samples are of distribution $X$ and $Y$, respectively. |
| $x, y$ | | Sample from $X$ and $Y$. |
| $\hat{X}, \hat{Y}$ | | the intermediate distributions $\hat{X} = G(X)$ and $\hat{Y} = F(Y)$. |
| $\bar{X}, \bar{Y}$ | | the cycle distributions $\bar{X} = F(\hat{Y})$ and $\bar{Y} = F(\hat{X})$. |
| $X_{\text{sim}}$ | ■ | First group of simulated neurons. |
| $Y_{\text{sim}}$ | ■ | Second group of simulated neurons. |
| $\Phi$ | | The handcrafted augmentation to replace neural activity with random noise. |
| $X_{\text{aug}}$ | ■ | V1 recordings obtained from the 1st day of the VR experiment. |
| $Y_{\text{aug}}$ | ■ | Augmented $X_{\text{aug}}$ using the transformation function $\Phi$, see Section 2.3.2. |
| $X_{\text{rec}}$ | ■ | V1 recordings obtained from the 1st day (i.e. pre-learning) of the VR experiment. |
| $Y_{\text{rec}}$ | ■ | V1 recordings obtained from the 4th day (i.e. post-learning) of the VR experiment. |
| VAE | | Variational autoencoder. |
| CNN | | Convolution neural network. |
| RNN | | Recurrent neural network. |
| Identity | | Identity baseline model. |
| Linear | | A linear autoencoder consists of a PCA encoder and a linear decoder. |
| $N_{PCA}$ | | The number of PCA components to explains 95% of variance. |
| VAE | | A VAE baseline that learns to generate $X \to Y$. |
| AE | | An autoencoder that learns to reconstruction calcium responses and we pre-sort neurons based on the the reconstruction loss. |
| Model-R | | The generator architecture with level-wise residual connection (Ronneberger et al., 2015), see Section 2.6.1. |
| $\text{US}_i, \text{DS}_i$ | | The $i^{\text{th}}$ down-sampling and up-sampling block in Model-R. |
| $\text{RB}_i$ | | The $i^{\text{th}}$ residual block in Model-R. |
| Model-AG | | The proposed Model-R generator architecture with Additive Attention Gate module (Oktay et al., 2018), see Section 2.6.1. |
| $\text{AG}_i$ | | The $i^{\text{th}}$ Additive Attention Gate module (layer) in Model-AG. |
| Model-AG-1D | | Model-AG with 1D convolutional layers instead of 2D. |
| MAE | | Mean absolute error. |
| MSE | | Mean squared error. |
| $R^2$ | | The coefficient of determination or R-squared. |

## B   CycleGAN

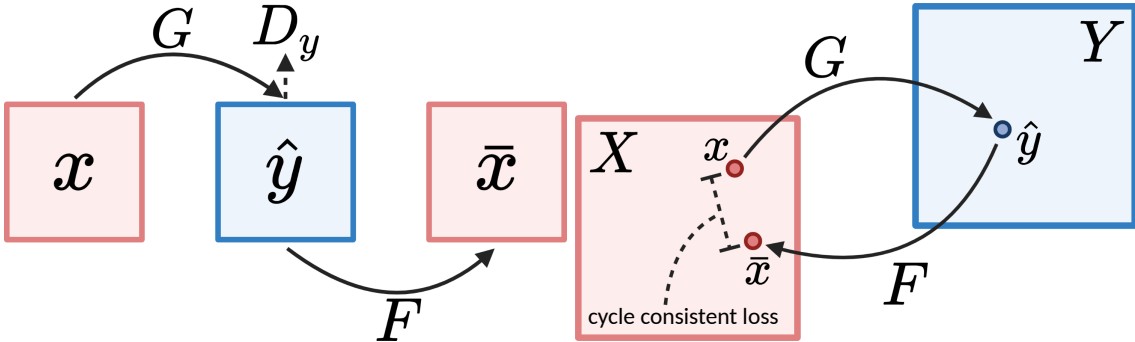

Figure B.1: Illustration of (Left) the data flow and (Right) the cycle-consistency in a forward cycle $X \to Y \to X$. $G$ and $F$ are generators that learn the transformation of $X \to Y$ and $Y \to X$ respectively. We first sample $x \sim X$, then apply transformation $G$ to obtain $\hat{y} = G(x)$. To ensure $\hat{y}$ resembles distribution $Y$, we train discriminator $D_Y$ to distinguish generated samples from real samples. However, even if $\hat{y}$ is of distribution $Y$, we cannot verify that $\hat{y}$ is the direct correspondent of $x$. Hence, we apply transformation $F$ which convert $\bar{x} = G(\hat{y})$ back to $X$. If both $F$ and $G$ are reasonable transformations, then the cycle consistency $\texttt{MAE}(x, \bar{x})$ should be minimal. The backward cycle $Y \to X \to X$ is a mirrored but opposite operation that runs concurrently with the forward cycle. Illustration re-created from Figure 3 in Zhu et al. (2017).

Table B.1: The objectives for $G$ and $D_Y$ in GAN, LSGAN, WGANGP and DRAGAN formulations (Goodfellow et al., 2014; Mao et al., 2017; Arjovsky et al., 2017; Kodali et al., 2017). $\lambda_{\text{GP}}$ denotes the gradient penalty coefficient in WGANGP and DRAGAN, $\epsilon$ is the $[0, 1]$ linear interpolation coefficient for WGANGP and $c$ is the Gaussian standard deviation for DRAGAN. The objectives for $F$ and $D_X$ are symmetric to $G$ and $D_Y$.

| MODEL | LOSS FUNCTIONS OF $G$ AND $D_Y$ |
|---|---|
| GAN | $\mathcal{L}^G = -\underset{x \sim X}{\mathbb{E}}\Big[\log(D_Y(G(x)\Big]$ |
| | $\mathcal{L}^{D_Y} = -\underset{y \sim Y}{\mathbb{E}}\Big[\log(D_Y(y))\Big] - \underset{x \sim X}{\mathbb{E}}\Big[\log(1 - D_Y(G(x)))\Big]$ |
| LSGAN | $\mathcal{L}^G = -\underset{x \sim X}{\mathbb{E}}\Big[(D_Y(G(x)) - 1)^2\Big]$ |
| | $\mathcal{L}^{D_Y} = -\underset{y \sim Y}{\mathbb{E}}\Big[(D_Y(y) - 1)^2\Big] + \underset{x \sim X}{\mathbb{E}}\Big[D_Y(G(x))^2\Big]$ |
| WGANGP | $\mathcal{L}^G = -\underset{x \sim X}{\mathbb{E}}\Big[D_Y(G(x))\Big]$ |
| | $\mathcal{L}^{D_Y} = \underset{x \sim X}{\mathbb{E}}\Big[D_Y(G(x))\Big] - \underset{y \sim Y}{\mathbb{E}}\Big[D_Y(y)\Big]$ |
| | $\qquad + \lambda_{\text{GP}} \underset{x \sim X, y \sim Y}{\mathbb{E}}\Big[\big(\parallel \nabla D(\epsilon y + (1 - \epsilon)G(x)) \parallel_2 - 1\big)^2\Big]$ |
| DRAGAN | $\mathcal{L}^G = \underset{x \sim X}{\mathbb{E}}\Big[\log(1 - D_Y(G(x)))\Big]$ |
| | $\mathcal{L}^{D_Y} = -\underset{y \sim Y}{\mathbb{E}}\Big[\log(D_y(y))\Big] - \underset{x \sim X}{\mathbb{E}}\Big[\log(1 - D_Y(G(x)))\Big]$ |
| | $\qquad + \lambda_{\text{GP}} \underset{y \sim Y, z \sim \mathcal{N}(0,c)}{\mathbb{E}}\Big[\big(\parallel \nabla D(y + z) \parallel_2 - 1\big)^2\Big]$ |

Table B.2: The hyperparameters search space. $\alpha_G$ and $\alpha_D$ are the learning rates of the generators and discriminators. $\lambda_{\text{GP}}$ is the gradient penalty coefficient for WGANGP and DRAGAN and $c$ is the Gaussian variance hyper-parameter in DRAGAN. $n_D$ is the number of discriminator updates for each generator update.

| HYPERPARAMETERS | GAN | LSGAN | WGANGP | DRAGAN |
|---|---|---|---|---|
| FILTERS | | POWER OF 2 FROM 2 TO 128 | | |
| KERNEL SIZE | | 1 TO 7, INTEGER | | |
| STRIDE SIZE | | [2, 3, 4] | | |
| ACTIVATION | | [TANH, RELU, LEAKY RELU, GELU] | | |
| NORMALISATION | | [BATCHNORM, LAYERNORM, INSTANCENORM] | | |
| SPATIAL DROPOUT | | MULTIPLE OF 0.05 FROM 0 TO 0.8) | | |
| $\lambda_{\text{CYCLE}}$ | | INTEGER FROM 1 TO 20 | | |
| $\lambda_{\text{IDENTITY}}$ | | INTEGER FROM 1 TO 20 | | |
| $\lambda_{\text{GP}}$ | N/A | N/A | INTEGER FROM 1 TO 20 | |
| $c$ | N/A | N/A | N/A | INTEGER FROM 1 TO 20 |
| $n_D$ | | 1 TO 5, INTEGER | | |
| $\alpha_G$ | | LOG UNIFORM FROM 1E-5 TO 1E-3 | | |
| $\alpha_D$ | | LOG UNIFORM FROM 1E-5 TO 1E-3 | | |
| DISTANCE FUNCTION | | MEAN ABSOLUTE ERROR | | |

Table B.3: The hyperparameters used for each objective formulation. $\alpha_G$ and $\alpha_D$ are the learning rates of the generators and discriminators. $beta_1$ and $beta_2$ are the exponential decay rates for the 1st and 2nd moment estimates, and $\epsilon_{\text{adam}}$ is the small constant value for numerical stability in Adam optimiser (Kingma & Ba, 2014). $\lambda_{\text{GP}}$ is the gradient penalty coefficient for WGANGP and DRAGAN and $c$ is the Gaussian variance hyper-parameter in DRAGAN. $n_D$ is the number of discriminator updates for each generator update.

| HYPERPARAMETERS | GAN | LSGAN | WGANGP | DRAGAN |
|---|---|---|---|---|
| FILTERS | | 32 | | |
| KERNEL SIZE | | 4 | | |
| STRIDE SIZE | | 2 | | |
| ACTIVATION | | GELU (HENDRYCKS & GIMPEL, 2016) | | |
| NORMALIZATION | | INSTANCENORM (ULYANOV ET AL., 2016) | | |
| SPATIAL DROPOUT | | 0.25 | | |
| WEIGHT INITIALISATION | | RANDOM NORMAL $\mathcal{N}(0, 0.02)$ | | |
| $\lambda_{\text{CYCLE}}$ | | 10 | | |
| $\lambda_{\text{IDENTITY}}$ | | 5 | | |
| $\lambda_{\text{GP}}$ | N/A | N/A | 10 | 10 |
| $c$ | N/A | N/A | N/A | 10 |
| $n_D$ | 1 | 1 | 5 | 1 |
| BATCH SIZE | | 2 | | |
| $\alpha_G$ | | 0.0001 | | |
| $\alpha_D$ | | 0.0004 | | |
| $\beta_1$ | | 0.9 | | |
| $\beta_2$ | | 0.999 | | |
| $\epsilon_{\text{ADAM}}$ | | 1E-7 | | |
| DISTANCE FUNCTION | | MEAN ABSOLUTE ERROR | | |

## C  Mouse information

The trial information of Mouse 2 to 4 in the VR experiment (see Section 2.2).

Table C.1: Mouse 2 performance in the experiment where 59 V1 neurons were monitored across 5 days of training.

| DAY | DURATION | NUM. TRIALS | AVG. TRIAL DURATION | LICKS | REWARDS |
|---|---|---|---|---|---|
| 1 | 897.84s | 61 | 14.72s | 1038 | 75 |
| 2 | 892.29s | 107 | 8.34s | 1572 | 115 |
| 3 | 898.20s | 196 | 4.58s | 2330 | 204 |
| 4 | 897.07s | 199 | 4.51s | 1338 | 304 |
| 5 | 895.48s | 122 | 7.34s | 1069 | 157 |

Table C.2: Mouse 3 performance in the experiment where 21 V1 neurons were monitored across 5 days of training.

| DAY | DURATION | NUM. TRIALS | AVG. TRIAL DURATION | LICKS | REWARDS |
|---|---|---|---|---|---|
| 1 | 895.19s | 68 | 13.16s | 919 | 98 |
| 2 | 897.11s | 76 | 11.80s | 1369 | 86 |
| 3 | 898.85s | 131 | 6.86s | 1146 | 173 |
| 4 | 895.78s | 177 | 5.06s | 1065 | 302 |
| 5 | 898.56s | 190 | 4.73s | 2334 | 196 |

Table C.3: Mouse 4 performance in the experiment where 32 V1 neurons were monitored across 5 days of training.

| DAY | DURATION | NUM. TRIALS | AVG. TRIAL DURATION | LICKS | REWARDS |
|---|---|---|---|---|---|
| 1 | 895.06s | 147 | 6.09s | 1239 | 192 |
| 2 | 898.54s | 300 | 3.00s | 1024 | 487 |
| 3 | 895.91s | 215 | 4.17s | 1982 | 220 |
| 4 | 896.83s | 227 | 3.95s | 2493 | 230 |
| 5 | 897.88s | 299 | 3.00s | 1750 | 303 |

# D   Neuron ordering

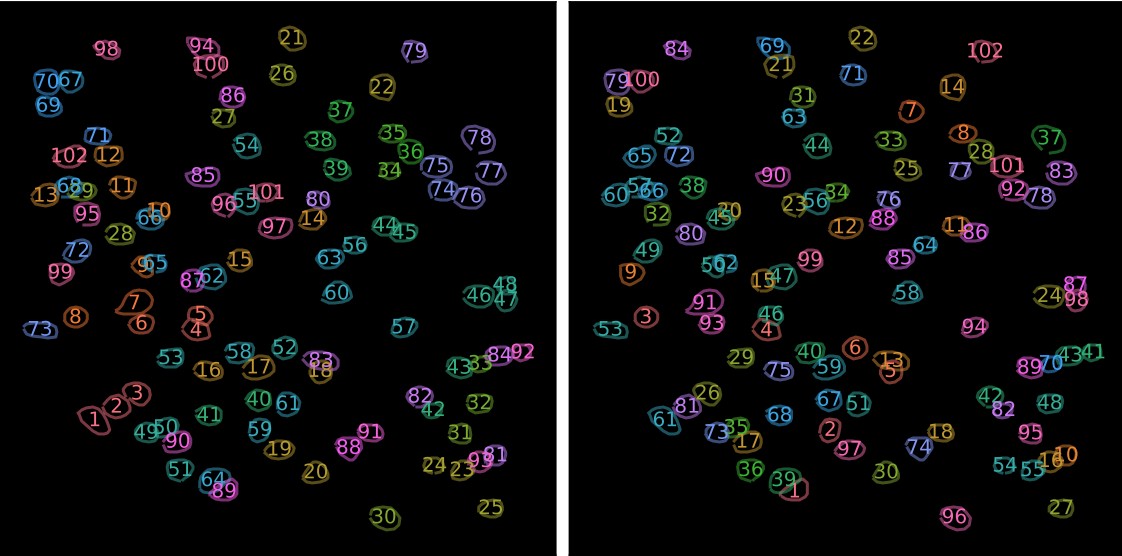

Figure D.1: Neuron ordering based on (Left) original annotation, (Right) autoencoder reconstruction loss. The original order was based on how visible the neuron was in the calcium image, hence not sorted in a particular manner. We proposed to train an autoencoder that learns to reconstruct $X$ and $Y$ jointly, and sort neurons based on the average reconstruction error on the validation set (see Section 2.9).

Table D.1: Mouse 1 neuron ordering based on the (OG) original annotation, (FR) firing rate, (CORR) average pairwise correlation, and (AE) autoencoder reconstruction loss.

| METHOD | ORDER |
|---|---|
| OG | 1, 2, 3, 4, 5, 6, 7, 8, 9, 10, 11, 12, 13, 14, 15, 16, 17, 18, 19, 20, 21, 22, 23, 24, 25, 26, 27, 28, 29, 30, 31, 32, 33, 34, 35, 36, 37, 38, 39, 40, 41, 42, 43, 44, 45, 46, 47, 48, 49, 50, 51, 52, 53, 54, 55, 56, 57, 58, 59, 60, 61, 62, 63, 64, 65, 66, 67, 68, 69, 70, 71, 72, 73, 74, 75, 76, 77, 78, 79, 80, 81, 82, 83, 84, 85, 86, 87, 88, 89, 90, 91, 92, 93, 94, 95, 96, 97, 98, 99, 100, 101, 102 |
| FR | 18, 14, 12, 30, 8, 15, 36, 4, 21, 19, 3, 7, 43, 33, 20, 42, 13, 6, 11, 39, 2, 22, 75, 28, 55, 100, 31, 62, 10, 67, 63, 54, 17, 40, 52, 46, 99, 88, 61, 77, 57, 34, 85, 41, 27, 98, 84, 47, 65, 73, 5, 1, 44, 101, 58, 80, 16, 29, 87, 9, 26, 83, 92, 74, 24, 45, 49, 23, 97, 48, 68, 60, 71, 76, 59, 53, 70, 89, 25, 93, 32, 56, 66, 81, 72, 94, 38, 64, 79, 82, 50, 51, 96, 90, 37, 86, 95, 91, 102, 69, 35, 78 |
| CORR | 36, 27, 46, 28, 39, 30, 42, 20, 92, 10, 18, 11, 67, 14, 4, 33, 19, 77, 75, 13, 24, 99, 8, 43, 65, 101, 63, 7, 25, 44, 12, 76, 80, 9, 47, 3, 34, 71, 87, 52, 22, 1, 85, 61, 84, 29, 45, 31, 93, 100, 5, 58, 57, 17, 74, 21, 96, 55, 82, 91, 2, 48, 6, 56, 83, 62, 49, 16, 26, 81, 97, 53, 73, 94, 89, 59, 40, 95, 23, 32, 54, 66, 98, 72, 35, 88, 15, 41, 50, 60, 90, 70, 78, 68, 69, 86, 38, 51, 64, 79, 37, 102 |
| AE | 89, 59, 8, 4, 18, 52, 37, 35, 99, 81, 44, 97, 83, 22, 87, 93, 90, 91, 69, 10, 100, 21, 96, 46, 39, 3, 25, 36, 53, 20, 86, 95, 38, 101, 50, 51, 78, 11, 64, 58, 92, 82, 84, 54, 66, 5, 62, 32, 72, 9, 61, 71, 73, 24, 23, 55, 68, 60, 17, 13, 1, 65, 27, 56, 102, 29, 40, 41, 94, 33, 26, 12, 49, 88, 16, 80, 34, 76, 70, 28, 2, 42, 77, 98, 63, 45, 48, 14, 43, 85, 7, 74, 6, 57, 31, 30, 19, 47, 15, 67, 75, 79 |

## E   Simulation data

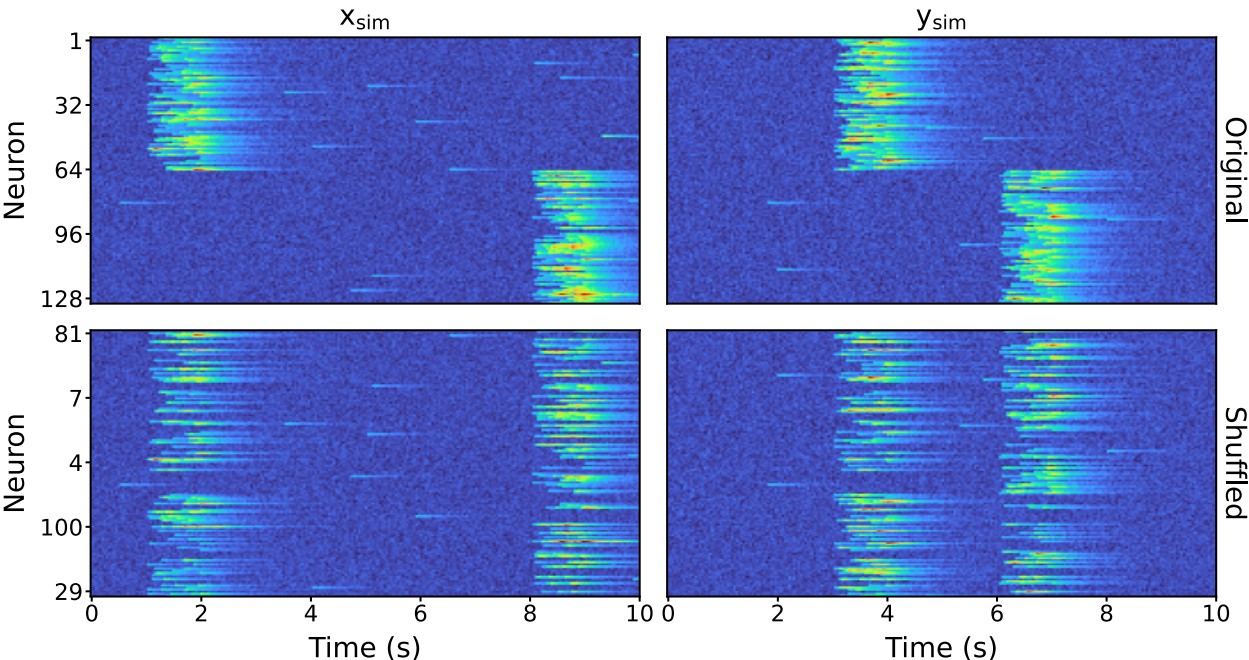

Figure E.1: The top panels show the calcium-like traces of all 128 neurons from $x_{sim} \sim X_{sim}$ and $y_{sim} \sim Y_{sim}$, and the bottom panels show the same population with neuron ordered shuffled, which are then feed into the unsupervised learning framework. The shuffling process is added to increase the difficulties for the generators to learn the transformation as the responses are less structured. `TURBO` colourmap is used to improve visibility.

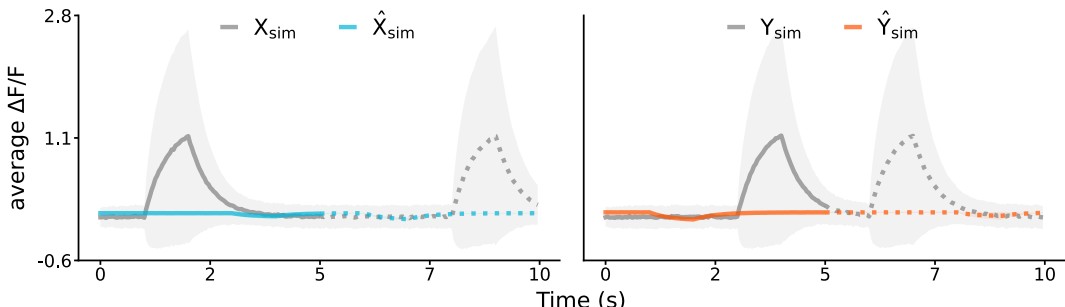

Figure E.2: The average response patterns ($\Delta F/F$) in (Left) $X_{sim}$ against $\hat{X}_{sim}$, and (Right) $Y_{sim}$ against $\hat{Y}_{sim}$ by the `Linear` model. The solid and dotted lines indicate the two activity patterns in each population, the grey and coloured lines correspond to the average simulated and translated responses, and the shaded areas show their variance. Table 2 shows the transformation results in terms of $R^2$.

## F  Augmented data

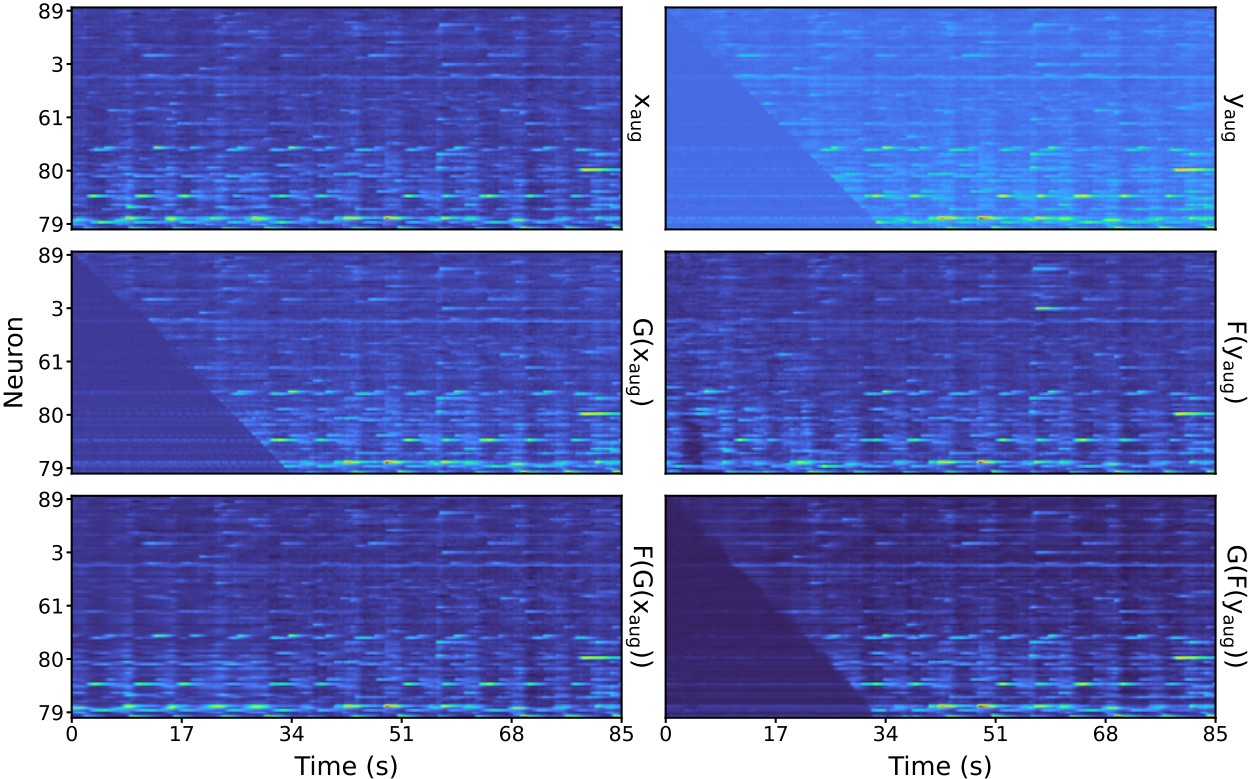

Figure F.1: The (Left column) forward and (Right column) backward cycle of the entire neuron population from a randomly selected trial, using the `Model-AG` architecture and trained with LSGAN objectives. $G$ learns the augmentation function $\Phi$ (described in Section 2.3.2) which mask out the lower left corner of input $x_{\mathrm{aug}}$, whereas $F$ learns to recover the masked regions from unpaired data. `TURBO` colourmap is used to improve visibility.

Table F.1: Test performance (in MAE) in the two intermediate transformations $X_{\text{aug}}$ vs $F(Y_{\text{aug}})$ and $Y_{\text{aug}}$ vs $G(X_{\text{aug}})$, and the two cycle transformations $X_{\text{aug}}$ vs $F(G(X_{\text{aug}}))$ and $Y_{\text{aug}}$ vs $G(F(Y_{\text{aug}}))$ on the augmented dataset. `Linear` ($N_{\text{PCA}} = 40$), `VAE` and `Identity` are baseline models which do not rely on cycle-consistency, whereas `Model-AG-1D`, which uses 1D convolutions, disregards the neuron spatial structure and is added to verify that 2D convolutions do indeed take advantage of the spatial information in neural responses. We evaluate the CycleGAN framework with different (A) generator architectures, (B) GAN objective formulations, and (C) neuron ordering. FR, CORR, and AE indicate neurons ordered by their firing rate, correlation, and `AE` reconstruction loss, respectively. Models trained with WGANGP and DRAGAN failed to learn either of the intermediate transformations (and cycle transformations for WGANGP). Moreover, $D_X(F(Y_{\text{aug}}))$ and $D_Y(G(X_{\text{aug}}))$ were neither informative nor impactful to the overall objective. Generators focused on optimising the cycle-consistent loss instead. It is likely that the gradient penalty term in DRAGAN and WGANGP further complicates the already perplexing overall optimisation objective, hence hindering the discriminators from learning meaningful features and being overpowered by the generators.

| | $X_{\text{AUG}}$ vs $F(Y_{\text{AUG}})$ | $X_{\text{AUG}}$ vs $F(G(X_{\text{AUG}}))$ | $Y_{\text{AUG}}$ vs $G(X_{\text{AUG}})$ | $Y_{\text{AUG}}$ vs $G(F(Y_{\text{AUG}}))$ |
|---|---|---|---|---|
| IDENTITY | $0.1328 \pm 0.2103$ | $0$ | $0.1328 \pm 0.2103$ | $0$ |
| LINEAR | $0.2423 \pm 0.4189$ | N/A | $0.2583 \pm 0.4388$ | N/A |
| VAE | $0.2558 \pm 0.3885$ | N/A | $0.2437 \pm 0.4061$ | N/A |
| (A) LSGAN OBJECTIVE AGAINST DIFFERENT GENERATOR ARCHITECTURES | | | | |
| MODEL-R | $0.1191 \pm 0.2204$ | $0.0253 \pm 0.0481$ | $0.1268 \pm 0.2294$ | $0.0281 \pm 0.0567$ |
| MODEL-AG | $0.1081 \pm 0.1816$ | $0.0589 \pm 0.0592$ | $0.1175 \pm 0.2008$ | $0.0645 \pm 0.0700$ |
| (B) MODEL-AG ARCHITECTURE AGAINST DIFFERENT OBJECTIVES | | | | |
| GAN | $0.1753 \pm 0.2221$ | $0.2009 \pm 0.2538$ | $0.2246 \pm 0.2708$ | $0.2200 \pm 0.2494$ |
| WGANGP | $3.9094 \pm 2.8441$ | $2.5724 \pm 2.0950$ | $3.9362 \pm 2.7241$ | $1.7874 \pm 1.3328$ |
| DRAGAN | $3.0314 \pm 2.4206$ | $0.1681 \pm 0.2496$ | $3.8994 \pm 2.7172$ | $0.1302 \pm 0.1673$ |
| (C) MODEL-AG AND LSGAN OBJECTIVE AGAINST DIFFERENT NEURON ORDERING | | | | |
| MODEL-AG-1D | $0.2313 \pm 0.3065$ | $0.1839 \pm 0.1828$ | $0.2372 \pm 0.3749$ | $0.1712 \pm 0.2872$ |
| FR | $0.1043 \pm 0.2675$ | $0.0514 \pm 0.0632$ | $0.1154 \pm 0.2164$ | $0.0649 \pm 0.0814$ |
| CORR | $0.1030 \pm 0.1870$ | $0.0760 \pm 0.0995$ | $0.1193 \pm 0.3396$ | $0.0559 \pm 0.0766$ |
| AE | $0.1024 \pm 0.1856$ | $0.1111 \pm 0.2260$ | $0.1103 \pm 0.2858$ | $0.1267 \pm 0.2633$ |

# G   Recorded data

## G.1   Mouse 1 results

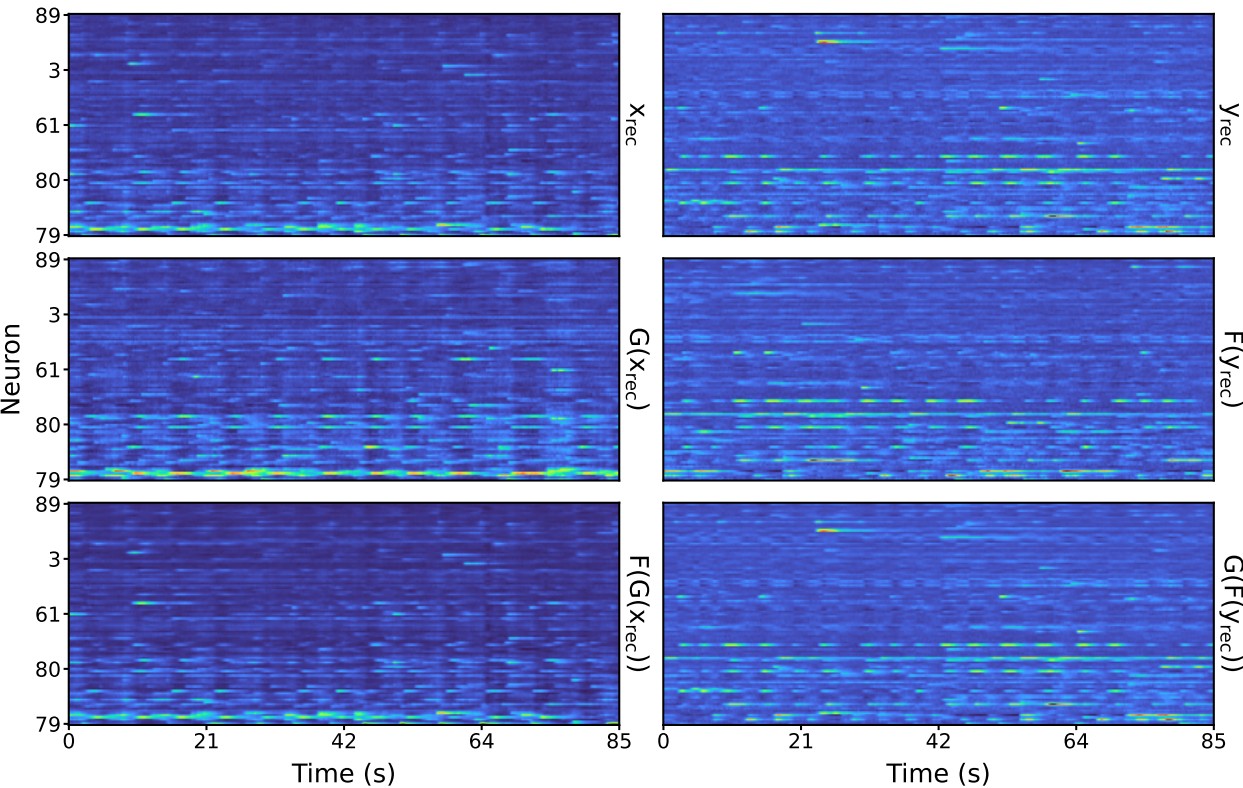

Figure G.1: The (Left column) forward and (Right column) backward cycle of the entire population from a randomly selected segment $x_{\text{rec}} \sim X_{\text{rec}}$ and $y_{\text{rec}} \sim Y_{\text{rec}}$. The model was trained with `Model-AG` generators using the LSGAN objective on the recorded dataset and neurons were in `AE` order. `TURBO` colourmap is used to improve visibility.

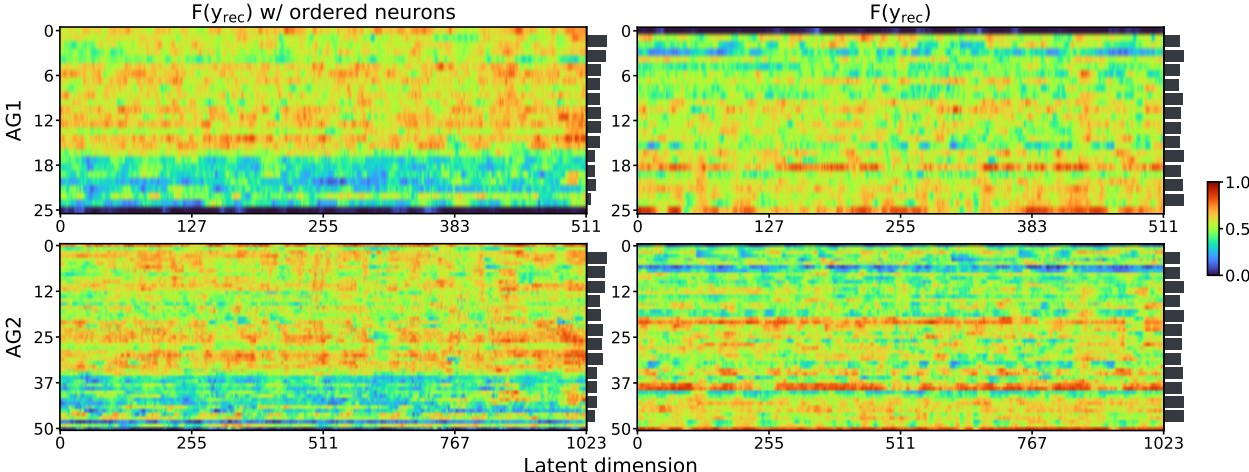

Figure G.2: Attention masks $AG_1$ and $AG_2$ from generator $F$ with (Left column) neurons sorted by `AE` reconstruction error and (Right column) no neuron ordering. The histogram to the right of each panel indicates the spatial attention intensity learned by the attention module.

Table G.1: Cycle-consistent and identity loss in the test set, where neurons were ordered by (OG) original annotation, (FR) firing rate, (CORR) pairwise correlation, (AE) AE reconstruction loss, respectively. For reference, $\texttt{MAE}(X_{\text{rec}}, Y_{\text{rec}}) = 0.3674 \pm 0.0236$ in the test set.

| ORDER | $X_{\text{REC}}$ VS $F(X_{\text{REC}})$ | $X_{\text{REC}}$ VS $F(G(X_{\text{REC}}))$ | $Y_{\text{REC}}$ VS $G(Y_{\text{REC}})$ | $Y_{\text{REC}}$ VS $G(F(Y_{\text{REC}}))$ |
|---|---|---|---|---|
| MODEL-AG-1D | $0.1502 \pm 0.0064$ | $0.1806 \pm 0.0077$ | $0.1463 \pm 0.0149$ | $0.1811 \pm 0.0163$ |
| OG | $0.0123 \pm 0.0015$ | $0.0874 \pm 0.0037$ | $0.0101 \pm 0.0010$ | $0.0766 \pm 0.0025$ |
| FR | $0.0108 \pm 0.0013$ | $0.0760 \pm 0.0030$ | $0.0070 \pm 0.0005$ | $0.0752 \pm 0.0028$ |
| CORR | $0.0111 \pm 0.0012$ | $0.0778 \pm 0.0028$ | $0.0089 \pm 0.0022$ | $0.0757 \pm 0.0024$ |
| AE | $0.0101 \pm 0.0012$ | $0.0733 \pm 0.0025$ | $0.0069 \pm 0.0007$ | $0.0737 \pm 0.0027$ |

Table G.2: The average KL divergence between recorded and translated distributions in (a) pairwise correlation, (b) firing rate, and (c) pairwise van Rossum distance. We repeated the same experiments with different neuron ordering methods as well as Model-AG-1D which uses 1D convolutions instead of 2D convolutions and does not take neuron spatial information into consideration. The PCA encoder in Linear consists of $N_{\text{PCA}} = 40$ components to capture 9% of the variance in the input. OG, FR, CORR, and AE indicate neurons ordered by their original annotation, firing rate, correlation, and AE reconstruction loss, respectively.

| | $X_{\text{REC}}$ VS $F(Y_{\text{REC}})$ | $X_{\text{REC}}$ VS $F(G(X_{\text{REC}}))$ | $Y_{\text{REC}}$ VS $G(X_{\text{REC}})$ | $Y_{\text{REC}}$ VS $G(F(Y_{\text{REC}}))$ |
|---|---|---|---|---|
| (A) PAIRWISE CORRELATION | | | | |
| IDENTITY | $0.0875 \pm 0.0549$ | $0$ | $0.0821 \pm 0.0471$ | $0$ |
| LINEAR | $2.9071 \pm 0.3869$ | N/A | $4.4880 \pm 0.6209$ | N/A |
| VAE | $1.1532 \pm 0.3885$ | N/A | $2.2670 \pm 0.4557$ | N/A |
| MODEL-AG-1D | $0.2027 \pm 0.1040$ | $0.4715 \pm 0.2051$ | $0.1901 \pm 0.1003$ | $0.4149 \pm 0.2194$ |
| OG | $0.0552 \pm 0.0419$ | $0.0754 \pm 0.0353$ | $0.0583 \pm 0.0553$ | $0.0174 \pm 0.0110$ |
| FR | $0.0507 \pm 0.0358$ | $0.0266 \pm 0.0146$ | $0.0504 \pm 0.0438$ | $0.0267 \pm 0.0176$ |
| CORR | $0.0539 \pm 0.0329$ | $0.0339 \pm 0.0176$ | $0.0534 \pm 0.0474$ | $0.0205 \pm 0.0133$ |
| AE | $0.0479 \pm 0.0372$ | $0.0329 \pm 0.0163$ | $0.0493 \pm 0.0448$ | $0.0283 \pm 0.0206$ |
| (B) FIRING RATE | | | | |
| IDENTITY | $8.0705 \pm 6.5500$ | $0$ | $7.7781 \pm 6.7338$ | $0$ |
| LINEAR | $19.5507 \pm 1.6473$ | N/A | $19.5703 \pm 1.3806$ | N/A |
| VAE | $13.4533 \pm 6.6228$ | N/A | $15.9878 \pm 5.6384$ | N/A |
| MODEL-AG-1D | $3.5688 \pm 3.8895$ | $7.9101 \pm 5.3517$ | $3.0572 \pm 3.1114$ | $8.3185 \pm 5.5950$ |
| OG | $1.5401 \pm 1.2491$ | $2.0442 \pm 2.0936$ | $1.8527 \pm 1.3563$ | $1.4697 \pm 1.1412$ |
| FR | $1.3402 \pm 1.0450$ | $1.2658 \pm 1.0784$ | $1.6994 \pm 1.4170$ | $1.4152 \pm 1.2221$ |
| CORR | $1.4006 \pm 1.1079$ | $1.5450 \pm 1.0786$ | $1.4088 \pm 1.0828$ | $1.4674 \pm 1.3505$ |
| AE | $1.1648 \pm 0.7934$ | $1.4022 \pm 1.2734$ | $1.0697 \pm 0.7689$ | $1.2705 \pm 1.1148$ |
| (C) PAIRWISE VAN ROSSUM DISTANCE | | | | |
| IDENTITY | $0.5510 \pm 0.2960$ | $0$ | $0.3053 \pm 0.1211$ | $0$ |
| LINEAR | $2.0803 \pm 0.2934$ | N/A | $3.4271 \pm 0.4390$ | N/A |
| VAE | $0.7336 \pm 0.2057$ | N/A | $1.9896 \pm 0.5960$ | N/A |
| MODEL-AG-1D | $0.3613 \pm 0.1597$ | $0.8045 \pm 0.1846$ | $0.3764 \pm 0.1565$ | $1.3897 \pm 0.8256$ |
| OG | $0.2790 \pm 0.2186$ | $0.1878 \pm 0.0477$ | $0.3216 \pm 0.1352$ | $0.1581 \pm 0.0664$ |
| FR | $0.2539 \pm 0.1708$ | $0.1003 \pm 0.0514$ | $0.3080 \pm 0.1173$ | $0.1536 \pm 0.0663$ |
| CORR | $0.2629 \pm 0.1877$ | $0.1905 \pm 0.0485$ | $0.2953 \pm 0.1230$ | $0.1797 \pm 0.0696$ |
| AE | $0.2387 \pm 0.1488$ | $0.1041 \pm 0.0376$ | $0.3031 \pm 0.1138$ | $0.1328 \pm 0.0592$ |

### G.2 Decoding performance

Table G.3: The decoding performances ($R^2$) in (a) virtual position and (b) velocity on Mouse 1 recordings from Day 1 and Day 4 of the VR experiment. We trained RNN regression models on the calcium responses of (1) all 102 neurons, (2) top-30 neurons, (3) the rest of the neurons, and (4) 30 randomly selected neurons. We fit the regression models 20 times using different random seeds and compute the p-value between (2) and (4). As the GradCAM activation map from $G$ and $F$ are extracted with respect to inputs $X_{\mathrm{rec}}$ and $Y_{\mathrm{rec}}$, and thus we selected the (2) top-30 neurons for Day 1 and Day 4 separately, each according to the positional activation maps from the two generators (see Figure 13). The best result for each decoding task is shown in bold.

| DAY | (1) ALL | (2) TOP-30 | (3) REST | (4) RANDOM-30 | P-VALUE |
|-----|---------|------------|----------|---------------|---------|
| (A) VIRTUAL POSITION | | | | | |
| 1 | $0.8721 \pm 0.0883$ | $0.8123 \pm 0.1255$ | $0.7597 \pm 0.1494$ | $0.5947 \pm 0.1300$ | $6.0858 \times 10^{-6}$ (****) |
| 4 | $0.8555 \pm 0.0698$ | $0.7912 \pm 0.0551$ | $0.7657 \pm 0.0676$ | $0.6736 \pm 0.0479$ | $2.3413 \times 10^{-8}$ (****) |
| (B) VELOCITY | | | | | |
| 1 | $0.2225 \pm 0.0533$ | $0.1900 \pm 0.0387$ | $0.1894 \pm 0.0802$ | $0.1778 \pm 0.0879$ | $0.5827$ (N.S.) |
| 4 | $0.5786 \pm 0.0461$ | $0.3053 \pm 0.0600$ | $0.4728 \pm 0.0915$ | $0.1320 \pm 0.0455$ | $3.1032 \times 10^{-12}$ (****) |

### G.2.1 Mouse 1 spike analysis

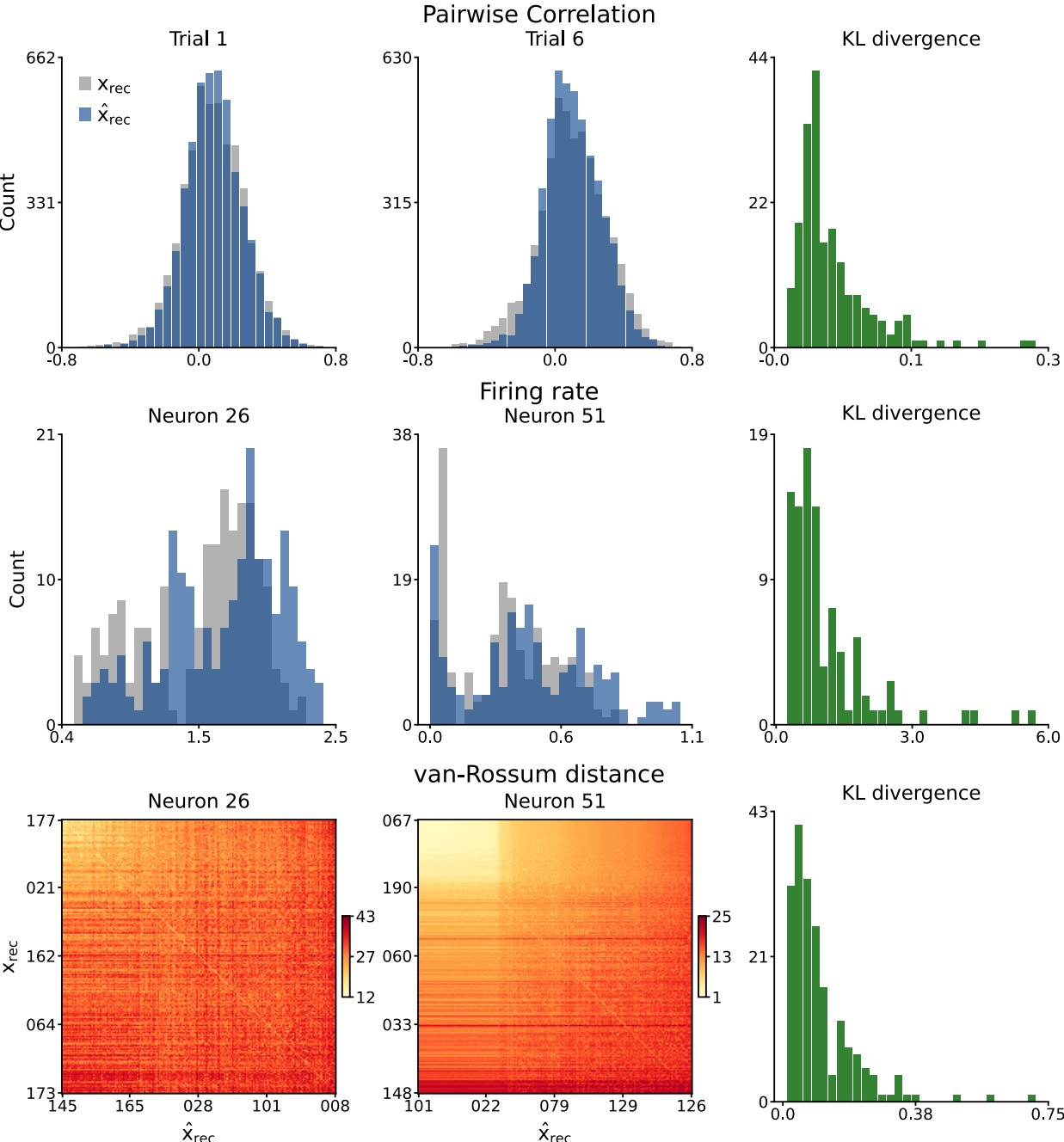

Figure G.3: Spike statistics comparison between $X_{\mathrm{rec}}$ and $\hat{X}_{\mathrm{rec}} = F(Y_{\mathrm{rec}})$ of (Top row) pairwise correlation from 2 randomly selected samples, (Middle row) firing rate of 2 randomly selected neurons and (Bottom row) van Rossum distance of 2 randomly selected segments. The 3rd column shows the KL divergence of each metric and Table G.2 shows the mean and standard deviation of the KL divergence comparisons. Neurons in AE order and the ground-truth $X_{\mathrm{rec}}$ is in grey colour.

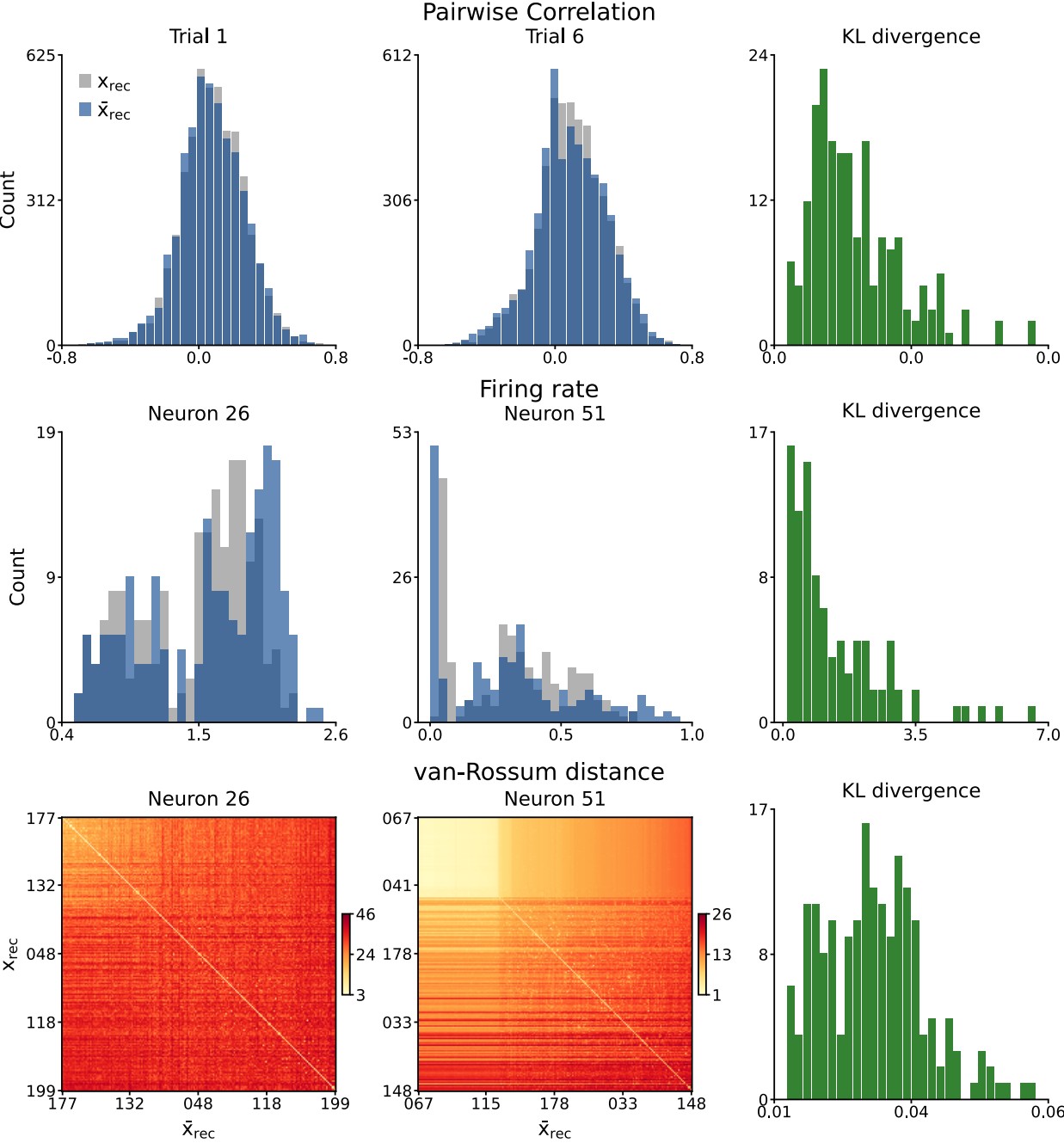

Figure G.4: Spike statistics comparison between $X_{\text{rec}}$ and $\bar{X}_{\text{rec}} = F(G(X_{\text{rec}}))$ of (Top row) pairwise correlation from 2 randomly selected samples, (Middle row) firing rate of 2 randomly selected neurons and (Bottom row) van Rossum distance of 2 randomly selected segments. The $3^{\text{rd}}$ column shows the KL divergence of each metric and Table G.2 shows the mean and standard deviation of the KL divergence comparisons. Neurons in `AE` order and the ground-truth $X_{\text{rec}}$ is in grey colour.

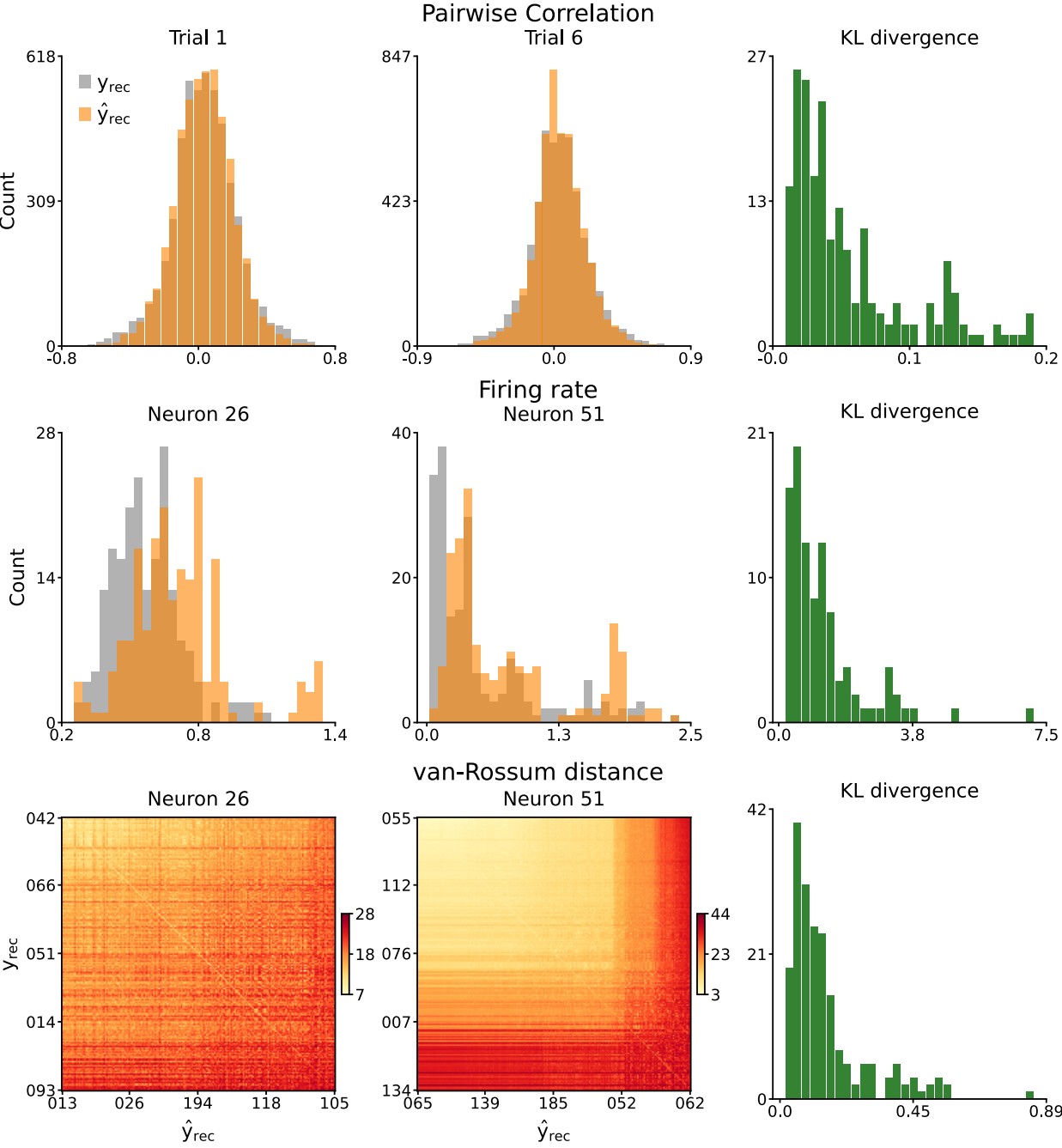

Figure G.5: Spike statistics comparison between $Y_{\text{rec}}$ and $\hat{Y}_{\text{rec}} = G(X_{\text{rec}})$ of (Top row) pairwise correlation from 2 randomly selected samples, (Middle row) firing rate of 2 randomly selected neurons and (Bottom row) van Rossum distance of 2 randomly selected segments. The 3$^{\text{rd}}$ column shows the KL divergence of each metric and Table G.2 shows the mean and standard deviation of the KL divergence comparisons. Neurons in AE order and the ground-truth $Y_{\text{rec}}$ is in grey colour.

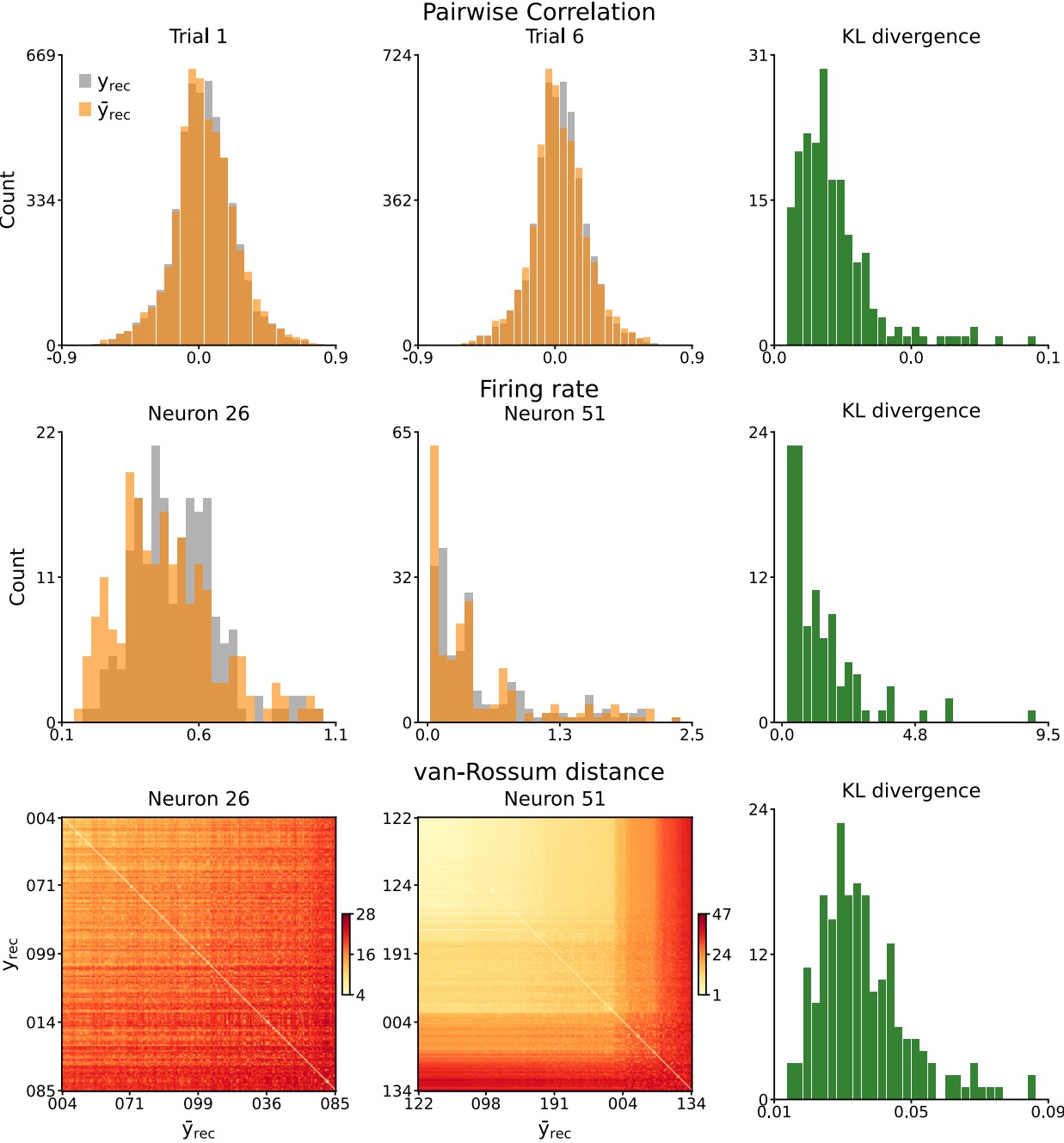

Figure G.6: Spike statistics comparison between $Y_{rec}$ and $\bar{Y}_{rec} = F(G(Y_{rec}))$ of (Top row) pairwise correlation from 2 randomly selected samples, (Middle row) firing rate of 2 randomly selected neurons and (Bottom row) van Rossum distance of 2 randomly selected segments. The 3$^{rd}$ column shows the KL divergence of each metric and Table G.2 shows the mean and standard deviation of the KL divergence comparisons. Neurons were ordered by AE reconstruction loss and the ground-truth $Y_{rec}$ is in grey colour.

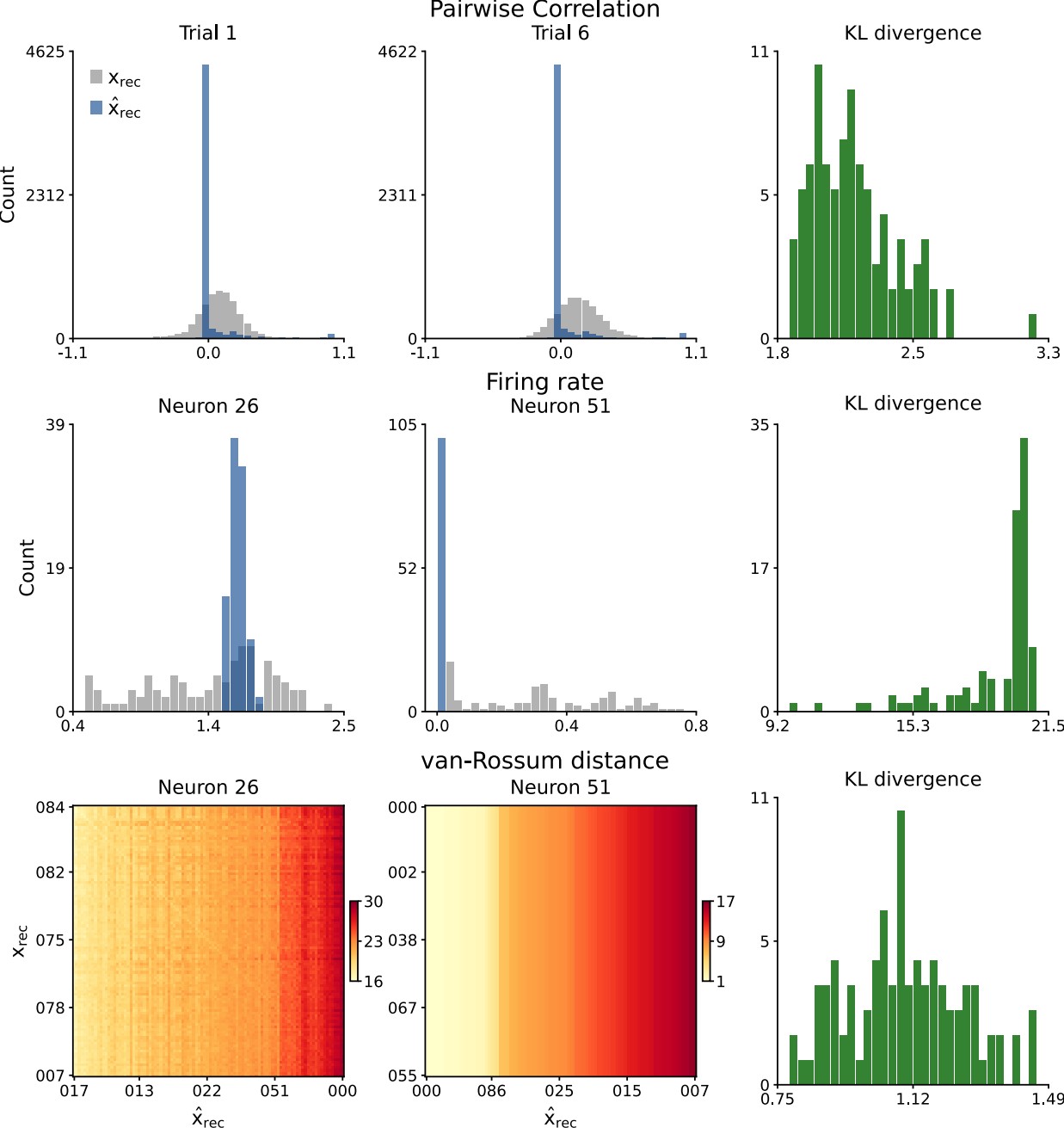

Figure G.7: Spike statistics comparison between $X_{rec}$ and $\texttt{VAE}(Y_{rec})$. (Top row) pairwise correlation from 2 randomly selected samples, (Middle row) firing rate of 2 randomly selected neurons and (Bottom row) van Rossum distance of 2 randomly selected segments. The $3^{rd}$ column shows the KL divergence of each metric and Table G.2 shows the mean and standard deviation of the KL divergence comparisons. Ground-truth $Y_{rec}$ is in grey colour.

### G.3 Mouse 2 results

Table G.4: Cycle-consistent and identity loss of `Model-AG` on Mouse 2 recordings, where neurons were ordered by (OG) original annotation, (FR) firing rate, (AE) `AE` reconstruction error. For reference, $\mathrm{MAE}(X_{\mathrm{rec}}, Y_{\mathrm{rec}}) = 0.6057 \pm 0.1146$ in the test set. The lowest loss in each category is marked in bold.

| ORDER | $X_{\mathrm{REC}}$ VS $F(X_{\mathrm{REC}})$ | $X_{\mathrm{REC}}$ VS $F(G(X_{\mathrm{REC}}))$ | $Y_{\mathrm{REC}}$ VS $G(Y_{\mathrm{REC}})$ | $Y_{\mathrm{REC}}$ VS $G(F(Y_{\mathrm{REC}}))$ |
|---|---|---|---|---|
| OG | $0.1292 \pm 0.0168$ | $0.5875 \pm 0.1050$ | $0.0923 \pm 0.0064$ | $0.4416 \pm 0.0763$ |
| FR | $0.1276 \pm 0.0152$ | $0.5794 \pm 0.1055$ | $0.0894 \pm 0.0048$ | $0.4396 \pm 0.0793$ |
| AE | $0.1030 \pm 0.0099$ | $0.5692 \pm 0.1008$ | $0.0101 \pm 0.0018$ | $0.4378 \pm 0.0769$ |

Table G.5: The average KL divergence between generated and recorded distributions of Mouse 2 in (a) pairwise correlation, (b) firing rate, and (c) population pairwise van Rossum distance. We compare `Model-AG` results with different neuron ordering methods. Note that we added the identity model (first row of each sub-table) as a baseline where we should obtain perfect cycle reconstruction.

| | $X_{\mathrm{REC}}$ VS $F(Y_{\mathrm{REC}})$ | $X_{\mathrm{REC}}$ VS $F(G(X_{\mathrm{REC}}))$ | $Y$ VS $G(X_{\mathrm{REC}})$ | $Y_{\mathrm{REC}}$ VS $G(F(Y_{\mathrm{REC}}))$ |
|---|---|---|---|---|
| (A) PAIRWISE CORRELATION | | | | |
| IDENTITY | $0.6528 \pm 0.4980$ | **0** | $0.4583 \pm 0.4366$ | **0** |
| OG | $0.5523 \pm 0.4251$ | $0.1617 \pm 0.0715$ | $0.1212 \pm 0.0833$ | $0.0499 \pm 0.0266$ |
| FR | $0.5639 \pm 0.4679$ | $0.1951 \pm 0.1031$ | $0.1126 \pm 0.0831$ | $0.0399 \pm 0.0231$ |
| AE | $0.5209 \pm 0.5554$ | $0.0582 \pm 0.0361$ | $0.1231 \pm 0.0988$ | $0.0352 \pm 0.0228$ |
| (B) FIRING RATE | | | | |
| IDENTITY | $8.3096 \pm 6.1580$ | $0$ | $5.5783 \pm 5.8451$ | $0$ |
| OG | $1.2881 \pm 1.1147$ | $2.5786 \pm 2.7222$ | $1.5782 \pm 1.2217$ | $1.6722 \pm 1.3286$ |
| FR | $1.2181 \pm 0.9909$ | $2.4912 \pm 2.5037$ | $1.3656 \pm 1.1475$ | $1.1767 \pm 1.0625$ |
| AE | $0.8087 \pm 0.5764$ | $1.1326 \pm 1.3149$ | $1.2521 \pm 0.9649$ | $1.0592 \pm 1.0722$ |
| (C) PAIRWISE VAN ROSSUM DISTANCE | | | | |
| IDENTITY | $1.3894 \pm 2.0529$ | $0$ | $1.1240 \pm 1.5159$ | $0$ |
| OG | $1.3392 \pm 1.6653$ | $0.5782 \pm 0.9743$ | $0.6043 \pm 0.5250$ | $0.2497 \pm 0.2443$ |
| FR | $1.2464 \pm 1.7505$ | $0.5946 \pm 0.9352$ | $0.5638 \pm 0.4181$ | $0.1977 \pm 0.1234$ |
| AE | $0.6946 \pm 0.5687$ | $0.1996 \pm 0.3232$ | $0.5287 \pm 0.3897$ | $0.1775 \pm 0.0959$ |

### G.4 Mouse 3 results

Table G.6: Cycle-consistent and identity loss of `Model-AG` on Mouse 3 recordings, where neurons were ordered by (OG) original annotation, (FR) firing rate, (AE) `AE` reconstruction error. For reference, $\mathrm{MAE}(X_{\mathrm{rec}}, Y_{\mathrm{rec}}) = 0.4764 \pm 0.1520$ in the test set.

| ORDER | $X_{\mathrm{REC}}$ vs $F(X_{\mathrm{REC}})$ | $X_{\mathrm{REC}}$ vs $F(G(X_{\mathrm{REC}}))$ | $Y_{\mathrm{REC}}$ vs $G(Y_{\mathrm{REC}})$ | $Y_{\mathrm{REC}}$ vs $G(F(Y_{\mathrm{REC}}))$ |
|---|---|---|---|---|
| OG | $0.0656 \pm 0.0037$ | $0.2684 \pm 0.0290$ | $0.0796 \pm 0.0047$ | $0.3229 \pm 0.0476$ |
| FR | $0.0585 \pm 0.0034$ | $0.2679 \pm 0.0309$ | $0.0777 \pm 0.0043$ | $0.3192 \pm 0.0477$ |
| AE | $0.0554 \pm 0.0023$ | $0.2677 \pm 0.0282$ | $0.0672 \pm 0.0034$ | $0.3199 \pm 0.0487$ |

Table G.7: The average KL divergence between generated and recorded distributions of Mouse 3 in (a) pairwise correlation, (b) firing rate, and (c) population pairwise van Rossum distance. We compare `Model-AG` results with different neuron ordering methods. Note that we added the identity model (first row of each sub-table) as a baseline comparison and should obtain perfect cycle reconstruction.

| | $X_{\mathrm{rec}}$ vs $F(Y_{\mathrm{rec}})$ | $X_{\mathrm{rec}}$ vs $F(G(X_{\mathrm{rec}}))$ | $Y$ vs $G(X_{\mathrm{rec}})$ | $Y_{\mathrm{rec}}$ vs $G(F(Y_{\mathrm{rec}}))$ |
|---|---|---|---|---|
| | (a) pairwise correlation | | | |
| Identity | $1.0188 \pm 0.5731$ | $0$ | $0.7363 \pm 0.3732$ | $0$ |
| OG | $0.5361 \pm 0.2817$ | $0.5678 \pm 0.3145$ | $0.6975 \pm 0.2202$ | $0.7381 \pm 0.2977$ |
| FR | $0.5021 \pm 0.2596$ | $0.5184 \pm 0.2536$ | $0.6281 \pm 0.2830$ | $0.6616 \pm 0.2850$ |
| AE | $0.5140 \pm 0.2538$ | $0.4751 \pm 0.2421$ | $0.6137 \pm 0.2997$ | $0.4625 \pm 0.2443$ |
| | (b) firing rate | | | |
| Identity | $12.2077 \pm 7.3556$ | $0$ | $12.4075 \pm 7.3156$ | $0$ |
| OG | $1.0164 \pm 0.7129$ | $1.8203 \pm 1.9280$ | $1.2904 \pm 0.9448$ | $1.4786 \pm 1.4374$ |
| FR | $0.9371 \pm 0.6735$ | $1.7893 \pm 2.5419$ | $1.0712 \pm 0.7793$ | $1.2805 \pm 1.5136$ |
| AE | $0.8936 \pm 0.5655$ | $1.1152 \pm 0.6797$ | $1.2114 \pm 0.7281$ | $0.6928 \pm 0.4643$ |
| | (c) pairwise van Rossum distance | | | |
| Identity | $4.2704 \pm 2.0834$ | $0$ | $4.9623 \pm 1.4393$ | $0$ |
| OG | $3.0412 \pm 1.8467$ | $2.0246 \pm 1.3422$ | $4.6059 \pm 2.0664$ | $3.0293 \pm 1.5854$ |
| FR | $2.9009 \pm 1.7587$ | $1.6458 \pm 1.2375$ | $4.1910 \pm 1.7950$ | $2.8613 \pm 1.7788$ |
| AE | $2.8383 \pm 1.5942$ | $1.4747 \pm 1.1150$ | $3.9709 \pm 1.7732$ | $1.4767 \pm 1.0195$ |

### G.5  Mouse 4 results

Table G.8: Cycle-consistent and identity loss of `Model-AG` on Mouse 4 recordings, where neurons were ordered by (OG) original annotation, (FR) firing rate, (AE) `AE` reconstruction error. For reference, $\mathrm{MAE}(X_{\mathrm{rec}}, Y_{\mathrm{rec}}) = 0.4383 \pm 0.2354$ in the test set.

| ORDER | $X_{\mathrm{REC}}$ VS $F(X_{\mathrm{REC}})$ | $X_{\mathrm{REC}}$ VS $F(G(X_{\mathrm{REC}}))$ | $Y_{\mathrm{REC}}$ VS $G(Y_{\mathrm{REC}})$ | $Y_{\mathrm{REC}}$ VS $G(F(Y_{\mathrm{REC}}))$ |
|---|---|---|---|---|
| OG | $0.0443 \pm 0.0015$ | $0.2538 \pm 0.0399$ | $0.0808 \pm 0.0061$ | $0.2403 \pm 0.0395$ |
| FR | $0.0376 \pm 0.0015$ | $0.2511 \pm 0.0389$ | $0.0764 \pm 0.0067$ | $0.2388 \pm 0.0406$ |
| AE | $0.0382 \pm 0.0012$ | $0.2489 \pm 0.0381$ | $0.0764 \pm 0.0053$ | $0.2367 \pm 0.0396$ |

Table G.9: The average KL divergence between generated and recorded distributions of Mouse 4 in (a) pairwise correlation, (b) firing rate, and (c) population pairwise van Rossum distance. We compare `Model-AG` results with different neuron ordering methods. Note that we added the identity model (first row of each sub-table) as a baseline comparison and should obtain perfect cycle reconstruction. Entries with the lowest value are marked in bold.

| | $X_{\mathrm{REC}}$ VS $F(Y_{\mathrm{REC}})$ | $X_{\mathrm{REC}}$ VS $F(G(X_{\mathrm{REC}}))$ | $Y$ VS $G(X_{\mathrm{REC}})$ | $Y_{\mathrm{REC}}$ VS $G(F(Y_{\mathrm{REC}}))$ |
|---|---|---|---|---|
| (A) PAIRWISE CORRELATION | | | | |
| IDENTITY | $0.3724 \pm 0.2169$ | $0$ | $0.5124 \pm 0.3238$ | $0$ |
| OG | $0.2849 \pm 0.1552$ | $0.1735 \pm 0.0918$ | $0.3536 \pm 0.2541$ | $0.5750 \pm 0.2883$ |
| FR | $0.2482 \pm 0.1502$ | $0.1478 \pm 0.0848$ | $0.3482 \pm 0.2561$ | $0.5471 \pm 0.2577$ |
| AE | $0.2096 \pm 0.1155$ | $0.1587 \pm 0.0867$ | $0.3460 \pm 0.2568$ | $0.4795 \pm 0.2457$ |
| (B) FIRING RATE | | | | |
| IDENTITY | $5.8031 \pm 4.8030$ | $0$ | $5.1383 \pm 5.4684$ | $0$ |
| OG | $1.3062 \pm 1.0097$ | $0.6034 \pm 0.6294$ | $1.4253 \pm 1.5599$ | $2.9196 \pm 3.1077$ |
| FR | $1.0818 \pm 0.9274$ | $0.5480 \pm 0.5043$ | $1.2120 \pm 1.2971$ | $2.8206 \pm 2.7266$ |
| AE | $1.0564 \pm 1.1415$ | $0.5474 \pm 0.5223$ | $1.1570 \pm 1.0830$ | $2.1015 \pm 2.2399$ |
| (C) PAIRWISE VAN ROSSUM DISTANCE | | | | |
| IDENTITY | $2.2670 \pm 1.2707$ | $0$ | $2.8134 \pm 1.5536$ | $0$ |
| OG | $1.8698 \pm 1.1525$ | $0.5625 \pm 0.4399$ | $2.4011 \pm 1.4879$ | $3.3849 \pm 1.7608$ |
| FR | $1.5416 \pm 0.9327$ | $0.3931 \pm 0.2821$ | $2.1379 \pm 1.4338$ | $3.3865 \pm 1.9320$ |
| AE | $1.3246 \pm 0.8537$ | $0.4578 \pm 0.3639$ | $2.2134 \pm 1.3838$ | $2.6537 \pm 1.6526$ |

