# OpenReview forum: "Neuronal Learning Analysis using Cycle-Consistent Adversarial Networks"
_TMLR — Rejected by TMLR_

### Review · Reviewer_ikNJ · 2022-09-30

**Summary Of Contributions:**

The authors' goal is to study changes in neural response properties due to learning in settings where no 1:1 correspondence of "before learning" and "after learning" trials exists, because the mouse is moving self-paced and therefore each trial is unique. They frame this problem as unpaired domain translation and use the CycleGAN framework to learn to translate population response between the two conditions.

The work tries to address a very relevant neuroscientific problem, but I have a hard time saying what its scientific contributions are – both from a machine learning perspective and from a neuroscience perspective. The paper reads like a mix and match of several ideas and components without a clear narrative as to why these methods are used and what we learn from it.

**Requested Changes:**

Overall I think this paper would need a major overhaul and rewrite on all fronts in order to be suitable for publication. Honestly, I think it should not have left the PIs desk in this form.

In trying to be constructive, here is a list of things that need to be fixed at the very least in my opinion:

 1. Please explain your rationale for using 2D convolutions over the time and neuron axes. Provide proper baselines and clarify what 1D-AGResNet is. Should it be a 1D convnet that treats each neuron as a channel, explain its architecture and why it performs so poorly.

 1. Provide simpler baselines. For example a simple linear autoencoder that takes a population vector for a single time frame, projects it down to a low-dimensional space and projects it back out to multineuronal space with learned weights. The encoder weights could even be taken from PCA. Alternatively, learn a scalar for each neuron. There are plenty of additional simple models that could serve as baselines and are probably not as easy to beat as one may think.

 1. Provide insights what we learn about learning in the brain. What conclusions can we draw from this model? If there are no concrete conclusions, provide some ideas what these conclusions might be or how the model might enable researches in the future to draw conclusions.

 1. Description of self-attention in Section 2.2 is not useful. A reader cannot deduce from the text what exactly is happening. This section should be rewritten to include all detail necessary to reproduce the work.

 1. Section 2.2 states that you "adapted GradCAM to generate localization maps", but it does not state how you adapted GradCAM. Please provide this information.

 1. Description of optimization procedure in Section 2.2 is incomplete. Which parameters where used for Adam. Which hyperparameters were optimized and over what range of values? What is the rationale for using the same hyperparameters on simulated and real data instead of optimizing hyperparameters for each dataset, as would be more common practice?

 1. I am a bit confused about the Attention Gate layer: Why are a and q added? In (self-) attention blocks, one typically has some form of similarity computation (e.g. dot product). Adding the two inputs seems odd, but I may be missing something. Why don't you use standard query-key-value attention?

 1. The description of the synthetic data (2.7.1) is unclear. I am unable to map the times cited (e.g. "17s to 34s") to what is shown in Fig. 4. Also, I don't understand what solid and dotted lines in the left panel mean. On the right hand side of the figure, it doesn't become clear what pattern 1 and 2 are, and why sometimes blue and orange are active at the same time and sometimes not. Please revise this description so a reader can understand the setup.

 1. I also could not follow the description of the augmented data. What does it mean to "zero out the lower left corners of the signals"? What exactly is shuffled in the training and validation set? Please provide a rationale you what you're doing and explain it more clearly.

 1. The description of the VAE used to order the neurons is incomplete. How do you treat the neurons vs. time dimensions? What dimension do the 1d convolutions go over? What is the dimensionality of the bottleneck layer?

 1. Sections 2.1–2.5 make repeated references to the paradigm, but the reader is not familiar with the experiment at this point, making it very hard to understand what's going on. I therefore suggest you move the description of the experiment to the beginning of the methods, i.e. before the description of CycleGAN (Section 2.1).

	- For instance, it was not clear to me in Section 2.1 why it's an unpaired problem and whether x and y depend on the stimulus. Only after reading 2.6 it became clear that it's an unpaired problem because the animal moves at different speeds in each trial, making the stimulus sequence and behavioral pattern unique in each trial.

	- It was neither clear to me at this point what the dimensionality of x and y are: neurons x timesteps or only one timestep?

 1. What are $\bar X$ and $\bar Y$? First used in Section 2.2. I couldn't find a definition

 1. Fig. 4 (left) has strange axes labels on the y axis (why not 0, 1, 2, 3?) and the numbers for the times cited in the caption seem to be off by a factor of ten with respect to the x axis in the figure and the numbers cited in the text.

 1. Fig. 7: Please explain what is shown here. What does "latent dimension" on the x axis refer to? The y axis is not labeled. From the caption, I'm guessing it's neurons (because the histograms on the right show "neuron-wise attention distribution"), but why are there 25 and 50 "neurons" and why does the number of bars in the histogram not match the number of neurons?

**Strengths And Weaknesses:**


### Strengths

+ Important neuroscientific problem


### Weaknesses

- Lots of ad-hoc architecture choices and no simple baselines
- Claims not clear or not really supported
- Paper is not well organized, quite hard to read and often missing crucial information



The paper makes six claims, none of which I find well supported. I will discuss them in turn.

 1. Claim 1: "We derive an end-to-end procedure to train..." - It's not really derived. It's basically a direct adaptation of CycleGAN, i.e. there is no technical novelty here that is worth claiming. The authors use complex combination of components like residual blocks, attention (see below), convolutions over neurons where convolution seems like an odd choice (see below), none of which is justified by proper ablation studies as necessary.

 1. Claim 2: "We empirically evaluate the CycleGAN performance with 4 different commonly used GAN objective formulations..." - There is one table showing results on one synthetic dataset (that is not really clear; see below). While it's true that the authors evaluated these 4 objectives, it's not clear to me what's the value of this particular comparison given the many very relevant aspects they did not evaluate.

 1. Claim 3: "We validate our method on 3 datasets..." - I find this validation completely insufficient. The two synthetic datasets are highly contrived and it's not clear what they tell us. The evaluation on the real dataset provides numbers, but I find it very hard to interpret these numbers. They are not really put into perspective and not precisely described. For instance, I could only guess what exactly KL divergences of pairwise correlations, firing rates and pairwise van Rossum distances are. Even the most trivial baselines are missing: for instance, what are these metrics if you split the data from one experimental condition into to halves and compare those, i.e. the best one could possibly hope for due to sampling noise? Or: if you learn a simple gain factor for each neuron to turn "before" into "after", what do you achieve?

 1. Claim 4: "We incorporate self-attention and feature-importance visualization ..." - I appreciate the effort, but it is not clear to me how you justify your interpretation of the attention maps. Why do they tell us something about which features the network uses? At the end it's concatenated with the input to the self-attention layer, so the next layer could completely ignore the output of the attention computation. I don't think you can draw any conclusions without causal manipulation of the computation. Also, you emphasize in the text that the regions close to the diagonal masking boundary are important and the network indeed learns this. But how do you interpret the fact that AG2 does not care at all about these regions and in AG1 the last row (25) is completely attended to, not only close to the diagonal? Finally, what does this analysis tell us about how learning changed neural population responses?

 1. Claim 5: "We perform a decoding analysis ..." - I think this claim needs to be put into perspective. How do alternative methods perform? For instance, what if you use the 30 most active neurons? How about doing PCA over neurons and taking the top-30 principal components? Showing that it's better than 30 random neurons is nice, but it doesn't really tell us whether the method is useful since there are no non-trivial baselines.

 1. Claim 6: "We propose a novel neuron ordering ..." - The use of convolutions over the neuron axis seems very odd to me in the first place, so I don't understand why this ordering is even necessary or desirable. What is the motivation to use weight sharing over neurons? There is clearly no simple shift equivariance for neurons. I don't see a reason to assume that neurons can be ordered in 1d such that shift equivariance would hold. Spatially, they're on a 2d lattice. Functionally, it's unknown whether there is a lattice structure. It would seem more principled to use 1d convolutions and treat each neuron as a channel. There seems to be a 1D-AGResNet in the paper, but I could not find out exactly what this is. On p.12 bottom you state that it disregards spatial information. If you treat each neuron as a channel and do 1D convolutions over time, this would not disregard correlations between neurons.

---

> ### Author Response · Authors · 2022-11-17
> **Response to Reviewer ikNJ**
>
> We thank the reviewer for their detailed review and constructive suggestions to improve this work. We have given the manuscript a major overhaul and addressed the requests raised by the reviewer as detailed below:
> 1. We have rewritten the network architecture description (Section 2.6) to clarify the implementation of the generators. Briefly, the $\texttt{1D-AGResNet}$ applies convolution operations over the temporal dimension of the data, i.e. the 2nd dimension in $(N, H=2048, W=102)$, and treats neurons as individual channels. Even though the $\texttt{1D-AGResNet}$ did not achieve the same level of performance as its 2D counterpart, it still performed similarly to the baseline models on the augmented and significantly outperforms them in the recorded datasets. We believe this is due to 2D convolution layers being able to take advantage of the spatial information in the response and learn better representations. We further expand upon this point by grouping neurons (i.e. sorting the 3rd dimension in $(N, H=2048, W=102, C=1)$) that are more informative (e.g. by firing rate, correlation, etc.). We showed that, in the augmented and recorded dataset (Sections 3.2 and 3.3), ordering neurons in the data matrix in a meaningful order does lead to improvement in the transformation result.
> 2. We would like to thank the reviewer for their suggestion on possible baseline models. Based on this comment as well as suggestions by other reviewers, we have introduced two additional baseline models. Briefly, given two distributions $X$ and $Y$, the (1) $\texttt{Linear}$ model first extracts a low-dimensional representation $\hat{x}$ via PCA with $N$ components that explain 95% of the variance over a population vector (i.e. a single time frame), we then project $\hat{x}$ back to the population dimension of $Y$ using a linear decoder; the (2) convolutional-based vanilla $\texttt{VAE}$ model where the encoder learns the mean and log-variance of the conditional distribution of the latent representation $z$ given $x$, and the decoder learns to output parameters of $Y$. Here, the overall optimisation objective is $\mathcal{L}_\texttt{VAE} = \log p (y|z) + \log p(z) - \log q(z | x)$. These two baseline models should represent the common linear and non-linear unsupervised approaches. The detailed descriptions of the two baseline models are available in Section 2.10 in the updated manuscript. We fitted both baseline models on the two synthetic datasets with known ground-truth transformation and the V1 recorded dataset. The results are shown in Sections 3.1, 3.2 and 3.3. Overall, the $\texttt{Linear}$ model failed to learn the transformation in all three datasets, and in some cases, performed worse than the $\texttt{Identity}$ model. The $\texttt{VAE}$ model, on the other hand, achieved similar performance as the CycleGAN methods in the simulated dataset but was unable to capture the more complex transformations in the augmented and recorded dataset as compared to the models trained in the CycleGAN framework.
> 3. This work has two main contributions to computational neuroscience. First, we show that our method is able to identify patterns in the neural activity that highlight task-relevant information in a fully data-driven manner. e.g. positional activation maps identify the reward zones in the VR experiment from segmented neural recordings. This ability is especially attractive for large-scale and unstructured animal experiments. Second, our decoding analysis showed that the method was able to identify a subset of neurons that are informative towards the behavioural variables. In large recordings, this allows experimentalists to shift the focus of their analysis from a large group of hundreds to thousands of neurons to a small group of relevant neurons.  On a technical note, we believe the neuron ordering method can be an easy-to-implement plug-in technique to improve the performance of convolution-based models on neural data. We have updated the manuscript to reflect these points.
>
> [Continue]

---

> > ### Author Response · Authors · 2022-11-17
> > **Response to Reviewer ikNJ (continue)**
> >
> > 4. We have rewritten Section 2.6.1 on the implementation of the additive attention gate (AG) module and updated the overall description of the network architectures and implementation detail. We adapted the additive attention gate from Oktay et al. [1] into our residual connections. Given $q$ and $a$, the outputs of $\texttt{RB}\_9$ and $\texttt{DS}\_2$ respectively, $\texttt{AG}\_1$ first apply a (separate) convolution with a kernel size of 1 and InstanceNorm for each variable, followed by an element-wise summation $s\_\texttt{AG} = \text{InstanceNorm}(\text{CONV}\_{1\times1}(q)) + \text{InstanceNorm}(\text{CONV}\_{1\times1}(a))$. We then apply ReLU to eliminate negative values in $s\_\texttt{AG}$, then learn a sigmoid mask $m_\texttt{AG} = \text{Sigmoid}(\text{InstanceNorm}(\text{CONV}_{1\times1}(\text{ReLU}(s\_\texttt{AG}))))$ via a convolution layer with a kernel size of 1 and InstanceNorm which has the same shape as $q$ and $a$. Finally, we apply the sigmoid mask $m_\texttt{AG}$ to $a$ and concatenate with $q$ in the channel dimension and pass its output to $\texttt{US}_1$. If all of $a$ is relevant, then this formulation works like \texttt{ResNet}. AG is easy to implement and can be a simple replacement for any block-wise residual concatenation layer. In addition, since $m_\texttt{AG}$ has the same shape as $a$ and $q$, we can overlay the sigmoid mask over either of the two variables and visualise the level of spatial-temporal attention learned by the AG module, thus improving the interpretability of the method.
> > 5. We have added Section 2.7 to briefly describe the GradCAM algorithm and discuss how we adapted the visual explanation technique to our use case. Briefly, GradCAM computes the gradient information flow between logits $y^c$ of class $c$ and the feature map $A^k \in \mathcal{R}^{u \times v}$ of a specific convolutional layer (usually the final layer) with $k$ filters. The gradients are then pooled over the spatial dimensions to calculate neuron importance weights $\alpha_k^c = \frac{1}{N} \sum_i^u \sum_j^v \frac{\partial y^c}{\partial A^k_{i,j}}$ where $N = u \times v$. The GradCAM activation map is the weighted combination of the feature maps followed by ReLU activation to eliminate features with negative influence $M_\text{GradCAM} = \text{ReLU}(\sum_k \alpha^c_k A^k)$. Similar to a sigmoid mask in the AG module (see Section 2.6.1), here, we can overlay the activation map $M_\text{GradCAM}$ on top of the input to visually interpret the region(s) that the model is focussing on. We applied the same method on the two discriminators to monitor the final convolution layer, and instead of a scalar prediction $y_c$, we compute the gradient flow between the feature maps of the target layer and the output. In the case of the generators, we instead monitor the gradient between the feature maps of the convolution layer in the last residual block $\texttt{RB}_9$ and the output, where the feature maps should learn the low-dimensional representation of the input after the down-sampling and residual processing block.
> > 6. We have updated Table B.3 with the hyperparameters of the optimizer. In addition, we added Table B.2 with random search space details for each objective function.
> > 7. Conceptually, we learn the attention mask $m_\texttt{AG}$ to eliminate irrelevant information in $a$ (i.e. a representation closer to the input) with respect to $q$ (i.e. a representation after a number of low-dimensional processing blocks). We, therefore, sum over $q$ and $a$ such that overlapping regions are amplified and apply ReLU activation to eliminate negative values. The attention-gated formulation in combination with convolutional layers has also been used in other works, especially in medical imaging [2, 3, 4].
> > 8. We thank the reviewer for pointing out our mistake in the description of the firing patterns. We have updated the simulated data description (Section 2.3.1) with the correct firing intervals for $X_\text{sim}$ and $Y_\text{sim}$. The synthetic traces in Figure 2 show a random example of 4 neurons from the simulated populations $X_\text{sim}$ and $Y_\text{sim}$. Each population consists of two firing patterns, for instance, neurons in $X_\text{sim}$ are either high firing between 1s to 2s or 8s to 9s, which we denote as Pattern 1 and Pattern 2 in Figure 2. In addition, we set the background spiking activity to 0.1Hz. Hence it is possible to have spikes outside of the firing window mentioned above. We have updated the caption and description to reflect these points.
> >
> > [continue]

---

> > > ### Author Response · Authors · 2022-11-17
> > > **Response to Reviewer ikNJ (continue)**
> > >
> > > 9. We thank the reviewer for their suggestion and agree that the original description of the augmented dataset was unclear. We have now rewritten the augmented dataset description (Section 2.3.2) with further details and precise formulation. The simulated dataset tests CycleGAN’s performance in transforming the overall activity patterns of two datasets. However, we are not able to evaluate the neuron-wise transformation performance. In order to test CycleGAN’s ability to recover calcium responses from unpaired data, we constructed the augmented dataset which consists of a handcrafted transformation $\Phi$ to the recordings obtained on the first day of the experiment. We shuffle the pairing of $X_\text{aug}$ and $Y_\text{aug}$ such that $x^i_\text{aug} \neq \Phi(y^i_\text{aug})$ for any sample $i$ in the training set. This forces the model to learn from unpaired samples but allows us to measure the transformation error in $F(Y_\text{aug})$ and $G(X_\text{aug})$ against their ground-truth data $X_\text{aug}$ and $Y_\text{aug}$ using common distance metrics on the test set.
> > > 10. We have revised the implementation detail in the Neuron ordering section. The \texttt{AE} model consists of three (encoder) down-sampling and (decoder) up-sampling convolution blocks, and a bottleneck layer of dimension $(256, 128)$. The down-sampling block consists of a 1D convolution layer followed by InstanceNorm, GELU activation, and Spatial Dropout, whereas a 1D transpose convolution is used in the up-sampling block instead. We fit the model by minimising the reconstruction loss $\mathcal{L}_\text{AE} = \texttt{MSE}(X, \texttt{AE}(X)) + \texttt{MSE}(Y, \texttt{AE}(Y))$. Then we compute the per-neuron reconstruction error on the validation set and sort the neurons in ascending order. That is, we rearrange the order in the neuron (3rd) dimension of the data matrix of shape $(N, H, W, C=1)$ where $H=2048$ and $W=102$ for $X_\text{rec}$ and $Y_\text{rec}$. It is important to note that the proposed neuron ordering process is an optional data preprocessing step that allows 2D convolution-based models to take advantage of the spatial information presented in neuronal responses and is not mandatory for the rest of this work to function. We also compare our method against neurons ordered by their original annotation, as well as neurons ordered by their average firing rate, and pairwise correlation.
> > > 11. We thank the reviewer for their suggestion and we agree that it would make more sense to first discuss the datasets as the subsequent Method sections made repeated references to the dataset. We rearranged the Method section by moving the dataset descriptions (Sections 2.2 and 2.3)  to the beginning of the Method section and also updated related paragraphs to improve the flow of the paper.
> > > 12. $\bar{X} = F(G(X))$ and $\bar{Y} = G(F(Y))$ refer to the cycle transformation/reconstruction, which is used to compute the cycle-consistent loss $\texttt{MAE}(X, \bar{X})$ and $\texttt{MAE}(Y, \bar{Y})$. We have updated the main text and also the notation table in the Appendix to make this information clear.
> > > 13. We have updated Figure 2 (previously Figure 4) to have integer intervals in the y-axis and the caption for the x-axis labels.
> > > 14. As per point 4, we have rewritten the description of the AG module to clarify its implementation detail and the dimension of each module. Briefly, the two attention gate modules $\texttt{AG}_1$ and $\texttt{AG}_2$ sit at the connection between the residual blocks ($\texttt{RB}$) and the first upsampling block $\texttt{US}_1$, and between the first upsampling block $\texttt{US}_1$ and the second upsampling block $\texttt{US}_2$, respectively. They thus operate at a lower dimension than the original input of shape $(2048, 102, 1)$. The output of $\texttt{AG}_1$ has shape $(512, 26, \text{hidden channels})$ and output of $\texttt{AG}_2$ has shape $(1024, 51, \text{hidden channels})$. Figure 5 illustrates the generator architecture (Figure 2, previously). The histogram to the right of each panel illustrates the distribution of the attention over the spatial dimension (i.e. 2nd dimension) and was intended to show the changes in attention distributions when training the model with and without neuron ordering (e.g. Figure 11 in Section 3.3). We included the histogram in AG plots for the augmented dataset for completeness. We have updated the caption of Figure 7 and Figure 11 to clarify this point.
> > >
> > > [continue]

---

> > > > ### Author Response · Authors · 2022-11-17
> > > > **Response to Reviewer ikNJ (continue)**
> > > >
> > > > In addition to the number of changes mentioned above, we have also introduced numerous updates to the manuscript which are summarised below. Large changes in the main text are written in red font for clarity:
> > > > - We updated the Introduction section with more recent works in neuronal analysis.
> > > > - We clarified the generator architectures with a more precise description in Section 2.6. In order to avoid any possible confusion with the naming of the model with established architectures, we have renamed $\texttt{ResNet}$ to $\texttt{Model-R}$, $\texttt{AGResNet}$ to $\texttt{Model-AG}$ and the 1D variant of $\texttt{AGResNet}$ to $\texttt{Model-AG-1D}$.
> > > > - As the comparison of different GAN objectives in the CycleGAN setting is not the main focus of this work, but more of an additional hyperparameter, we have moved the formulation and discussion of the various GAN objectives to Appendix B.
> > > > - In the Introduction and Discussion sections, we reiterated the ability to identify trial-relevant information in a fully data-driven manner making this work suitable for large-scale unstructured datasets.
> > > >
> > > > We would like to thank the reviewer for their time and constructive comments, we hope that the updated manuscript addresses the concerns raised by the reviewer.
> > > >
> > > > [1] Oktay, Ozan, et al. "Attention u-net: Learning where to look for the pancreas." arXiv preprint arXiv:1804.03999 (2018).
> > > >
> > > > [2] Schlemper, Jo, et al. "Attention gated networks: Learning to leverage salient regions in medical images." Medical image analysis 53 (2019): 197-207.
> > > >
> > > > [3] Woo, Sanghyun, et al. "Cbam: Convolutional block attention module." Proceedings of the European conference on computer vision (ECCV). 2018.
> > > >
> > > > [4] Guo, Meng-Hao, et al. "Attention mechanisms in computer vision: A survey." Computational Visual Media (2022): 1-38.

---

> > ### Comment · Reviewer_ikNJ · 2023-01-30
> > **Still not clear what exactly the paper contributes**
> >
> > I would like to thank the authors for the detailed response and the revision of the paper.
> >
> > They addressed most of my points about documenting what was done. A reader can now indeed understand much better what was done, except for a number of remaining issues I will list in the next comment.
> >
> > **Unfortunately, the major issue with the paper remains: It's still not clear to me what exactly the paper contributes.** The authors removed some of the claims and rephrased others, but even in the revision I find the claims not well supported or don't understand their relevance. It's now a better documented mix and match of several ideas and components, but the "why?" and "what do we learn?" remain as unclear as in the first version.
> >
> > Let me explain:
> >
> > ### New claim 1 - "We introduce two synthetic datasets..."
> >
> > As in my previous review I maintain that these two datasets are highly contrived and do not appear to be indicative of performance on a real dataset where ground truth is lacking. They are specifically designed in a way that simple transformations (e.g. gain modulation) don't work, but the authors do not present convincing evidence that this is the case in real datasets.
> >
> >
> > ### New claims 2 ("We demonstrate that our method is able to identify neurons...") and 3 ("We perform a decoding analysis...")
> >
> > These two claims seem to be the same thing to me. The authors are trying to make a point that the method identifies "important"/"relevant" neurons. But what is the evidence for this claim?
> >
> > - That decoding from the top-30 neurons is better than a random set of 30? That's the weakest possible baseline. That can probably be achieved trivially by taking the top-30 most active neurons or any number of alternative methods.
> >
> > - Why couldn't the same be achieved by traditional methods? Fig. 13 shows the data aligned to the position of the mouse along the path. Why couldn't you simply align the responses like this, then average within each bin and compare pre/post learning? Sorting by biggest (absolute or relative) difference between pre and post will probably reveal a similar set of "important" neurons.
> >
> >
> > ### New claim 4 - "We propose a novel neuron ordering method"
> >
> > I still don't understand why one needs to "order" neurons and why convolution over neurons would be a meaningful thing to do. The authors present results that it helps in their hands, but it's not clear *why* and the *how* is also quite strange. For instance:
> >
> > - The neurons seem to be sorted by reconstruction error of a VAE, firing rates or correlations. Why would any of these bring similar neurons close to each other? What does reconstruction quality have to do with similarity in responses? Why would ordering neurons by how correlated they are on average with other neurons bring similar neurons together (think of a population with two groups that are uniformly positively correlated within and uncorrelated across - what would happen?)? Same for firing rates? Why would more active neurons be more similar to each other?
> >
> > - The authors' evidence that simpler baselines don't work is questionable. Judging from Table 4, all of these methods perform worse than the identity, suggesting that they did not learn anything, i.e. were not properly optimized and/or regularized. Unfortunately I still don't quite understand how exactly they were trained: the same GAN objective as the complicated architecture proposed by the authors or something else?

---

> ### Comment · Reviewer_ikNJ · 2023-01-30
> **Miscellaneous other issues**
>
> - I couldn't find a description how the evaluation in Table 4 was done: Were these metrics computed on a held-out test set not used for training the model? If so, how were the splits done? I'm somewhat worried that the model memorized the training set. Did you look for the nearest neighbors for F(X) in Y?
>
> - P.3 first line: The explanation in parentheses for why it's unpaired seems to be bogus. The real answer that's not stated in this paragraph is that the mouse is moving at its own pace and therefore never sees the same stimulus sequence twice.
>
> - P 5. above Eq. 2: H and W seem to be backwards. At least Fig. 3 shows H=neurons and W=time.
>
> - Eq. 2: less than should be greater than (I think).
>
> - Below Eq. 3: The description of the shuffling is unclear. It's not clear what is being shuffled and what the superscript i (overloaded compared to above, Eq. 2) refers to.
>
> - P.9, 2.9: Description of AE: Not clear how the bottleneck can be (256,128) [what's the input dimension?] when it's using 1D convolutions. This is a recurring issue with the manuscript that dimension and indices are not always properly defined (it's better but still not clear at many places).
>
> - P.9, 2.10: Why is the VAE now using 2D convolutions whereas the linear model is only operating on individual timepoints?
>
> - 2.10: It's not clear what objectives these models are trained on? How do they translate from X to Y?
>
> - P 13: "Overall, models trained on sorted neurons achieved better results with ordering neurons by AE reconstruction loss being the most performant." <-- all models with ordering perform equally if you take the error bars into account. What do the error bars report? SD or SEM or 95% confidence intervals?
>
> - Table 4: Not clear what the van Rossum distances are. How are they computed and what do they measure exactly in the context of your questions? Would need to be explained in order to be useful.

---

### Review · Reviewer_bZvP · 2022-10-31

**Summary Of Contributions:**

The authors provide a framework for mapping neuronal activity measurements before and after learning/adapting to a task. They use a CycleGAN framework for this purpose. Their methods are developed for calcium signaling data where a number of neurons are imaged for segments of time.They use a convolutional architecture for the generator that treats the spatio temporal data as an image. Further they employ self-attention which can be used for interpretation important regions for the task, they also augment this with GRADCAM.

**Broader Impact Concerns:**

No ethical concerns.

**Requested Changes:**

1. Formulate this problem clearly, explain what is unpaired, is it the timepoints, is it the stimuli or what?
2. Compare to other architectures, not just variants of cycleGAN.
3. Contribute the dataset.

**Strengths And Weaknesses:**

Strengths:
1. I think the overall idea of using a generative model to predict neuronal activation after "learning" is interesting, and likely useful in neuroscience for predicting responses to tasks.
2. The idea of looking at segments of particular length as forming a pattern (and thus being convolved over) is useful here because of the high degree of autocorrelation in time-steps of neuronal data.

Weaknesses:
1. The writing does not make the problem formal or mathematically clear. There are several things that are unspecified. First, what exactly is unpaired in this training? It seems like the training data before and after learning a task is measured on the SAME mouse. So the entity is indeed paired. Is it timepoints within a segment that are unpaired? Moreover, formulating this problem clearly for the ML audience will enable others to develop methods for this question.
2.Given my questions about pairing, it does seem as though a CycleGAN may not be needed.  To make it clear, CycleGANs map from distribution to distribution when an alignment is not known, thus they use population level distribution matching as a proxy for penalizing individual datapoint domain transfer. In fact, this approach is extremely under-constrained and even when there is no paired measurement, but there are known correspondences one can do better [see MAGAN Amodio et al. ICML 2018]. However, here it seems that data are recorded before and after learning and thus there are correspondences in this task.  Second, CycleGANs don't use stochasticity, which could be important for this type of task. I would urge the authors to try or at least compare to something like a VAE model, or at least an encoder-predictor model that encodes a pre-learning neuronal images and then predicts post learning.
2. There are no ablations or tests of non-GAN models since GANs may not be needed here. I suggest testing other transformation models, and potentially latent space arithmetic models.
3. One of the contributions to machine learning could indeed be the datasets and even synthetic dataset, I would add clearer descriptions of these.

---

> ### Author Response · Authors · 2022-11-17
> **Response to Reviewer bZvP**
>
> We thank the reviewer for the overall positive assessment of our work and informative suggestions to improve this work. We have addressed each point and question raised by the reviewer below:
> - We have updated the Introduction to cover more recent work in analysing how neural activities reshape with experience and highlighted our motivation to tackle this problem as an unsupervised translation problem. In addition, we added Section 2.1 to formalise our problem setting with the following paragraph: “Our goal is to model the transformation between pre-learning and post-learning activities. Given two sets of neural recordings $X$ and $Y$ which correspond to the pre-learning and post-learning activities from the same behaving animal, one could learn the transformation between the two sets in a supervised manner, i.e. given a trial $i=1$ in $x^i \in X$, learn model $G: X \rightarrow Y$ to minimise the error in $G(x^i)$ and $y^i \in Y$. However, due to trial-to-trial variability in neuronal responses [1], as well as external factors that can influence the recording session (e.g. small changes in lighting or level of attention by the animal), trial-to-trial pairings of $X$ and $Y$ tend to be noisy (i.e. we cannot ensure that trial $i=1$ in $X$ corresponds to trial $i=1$ in $Y$). Moreover, combinations of these factors are unlikely to occur multiple times, leading to a small number of samples for training our models. Instead, we investigate the problem of learning the $X \rightarrow Y$ transformation with unknown pairing as an unsupervised translation task. In other words, given the neural recordings of a novice animal, can we translate the responses that correspond to the animal with expert-level performance, and vice versa?”
> - We would like to thank the reviewer for their suggestion on possible baseline models and non-GAN methods. Based on this comment as well as suggestions by other reviewers, we have introduced two additional baseline models. Briefly, given two distributions X and Y, the (1) $\texttt{Linear}$ model first extracts a low-dimensional representation $\hat{x}$ via PCA with N components that explain 95% of the variance over a population vector (i.e. a single time frame), we then project $\hat{x}$ back to the population dimension of Y using a linear decoder; the (2) convolutional-based vanilla $\texttt{VAE}$ model where the encoder learns the mean and log-variance of the conditional distribution of the latent representation z given x, and the decoder learns to output parameters of Y. Here, the overall optimisation objective is $\mathcal{L}_\texttt{VAE} = \log p (y|z) + \log p(z) - \log q(z | x)$. These two baseline models represent the common linear and non-linear unsupervised approaches. The detailed descriptions of the two baseline models are available in Section 2.10 in the updated manuscript. We fitted both baseline models on the two synthetic datasets with known ground-truth transformation and the V1 recorded dataset. The results are shown in Sections 3.1, 3.2 and 3.3. Overall, the $\texttt{Linear}$ model failed to learn the transformation in all three datasets, and in some cases, performed worse than the $\texttt{Identity}$ model. The $\texttt{VAE}$ model, on the other hand, achieved similar performance as the CycleGAN methods in the simulated dataset but was unable to capture the more complex transformations in the augmented and recorded dataset as compared to the models trained in the CycleGAN framework.
> - We thank the reviewer for this suggestion. The recorded dataset will be made publicly available upon acceptance. The code that was used to generate the two synthetic datasets are already part of the supplementary file and will be made available on a publicly accessible Git repository. We have updated the manuscript to reflect our commitment.
>
> [Continue]

---

> > ### Author Response · Authors · 2022-11-17
> > **Response to Reviwer bZvP (continue)**
> >
> > In addition to the number of changes mentioned above, we have also introduced numerous updates to the manuscript which are summarised below. Large changes in the main text are written in red font for clarity:
> > - We updated the Introduction section with more recent works in neuronal analysis.
> > - We rearranged the Method section by moving the dataset descriptions (Sections 2.2 and 2.3) to the beginning of the Method section as subsequent sections referenced to the datasets. Moreover, we rewrote part of the description for the augmented dataset (Section 2.3.2) to clarify its construction.
> > - We clarified the generator architectures with a more precise description in Section 2.6. In order to avoid any possible confusion with the naming of the model with established architectures, we have renamed $\texttt{ResNet}$ to $\texttt{Model-R}$, $\texttt{AGResNet}$ to $\texttt{Model-AG}$ and the 1D variant of $\texttt{AGResNet}$ to $\texttt{Model-AG-1D}$.
> > - We added Section 2.7 to describe the implementation of GradCAM in our work.
> > - As the comparison of different GAN objectives in the CycleGAN setting is not the main focus of this work, but more of an additional hyperparameter, we have moved the formulation and discussion of the various GAN objectives to Appendix B.
> > - In the Introduction and Discussion sections, we reiterated the ability to identify trial-relevant information in a fully data-driven manner making this work suitable for large-scale unstructured datasets.
> >
> > We would like to thank the reviewer for their time and constructive comments, we hope that the updated manuscript addresses the concerns raised by the reviewer.
> >
> > [1] Carandini, Matteo. "Amplification of trial-to-trial response variability by neurons in visual cortex." PLoS biology 2.9 (2004): e264.

---

### Review · Reviewer_4Xms · 2022-11-03

**Summary Of Contributions:**

This paper applies cycle-consistent losses to learn a mapping between neural activity before and after learning. The method is applied on both synthetic datasets and real calcium imaging recordings from the primary visual (V1) region of the mouse brain while the animal is learning to master a visual-based behaviour task. The authors use a number of different visualization methods including attention visualization and GradCAM to show that CycleGAN models could possibly be used to identify neurons and subgroups of neurons that are relevant to the learning process.



**Broader Impact Concerns:**

None noted.

**Requested Changes:**

*Requested Changes*

_1. Related work and background_
- The authors should discuss more recent research works on neural activities, as highlighted in the main review.
- The authors should formally define the abovementioned “CycleGAN framework”, and explicitly state how different GAN objectives are used in the framework, and how using them in the proposed way distinguish them from their original formulations. When referring to them in the other sections, please clearly state if the original formulation in the original paper is used or if the modified variant of the corresponding loss function is used.
- The authors also should cover more recent related works that are related to CycleGAN.

_2. Baselines and comparisons_
- To understand the advantages of the CycleGAN framework over other unsupervised models, the reviewer thinks it would be useful to have some comparisons with alternative models.

_3. Description of methodology_
- The authors should clarify how GradCAM and the Autoencoders are used, and provide corresponding evidence. Specifically, the authors should justify 1) how GradCAM could be used in generative models and why using it in the proposed way would yield the desired results; and 2) how training an autoencoder with reconstruction loss would re-order the neurons.
- Please clarify the definition of ResNet and AGResNet in the methodology section based on what is mentioned above.



**Strengths And Weaknesses:**


*Strengths*
- The motivation of the paper is clear and of interest to the community.
- The use of cycle-consistency to learn the unknown transformation and build interpretability into the learning process could be beneficial to the field in the long term.
- The application experiments (VR experiments on rodents) on neural recordings is relatively novel and interesting.
- The authors applied many methods to visualize and interpret the learned representation, which could lend themselves to biological insight.

*Weaknesses*

_1. Novelty of the proposed method._
- The proposed method isn’t novel. It is almost entirely based on existing works including CycleGAN, LSGAN, WGANGP, and DRAGAN with few modifications. The work also did not propose new architecture, new loss function, or new methods for visualization and analysis. The method is mostly just a combination of existing methods and applying them on calcium images.

_2. Lack of comparisons with other unsupervised models for neural analysis_
- There are a number of different tools for many session data analysis, including simple baselines like CCA (as in Gallego et al, [4]). How well do these methods perform in the tasks described?

_3. The related work is not complete and doesn’t discuss work that has a lot of common motivations_
- When discussing deep learning methods for neuronal activities, very few existing works are mentioned. There has been a lot of work in this area over the past few years that has not been included in related work or discussion (see [1-8], see pg. 23 of 8 for a summary of deep learning and other latent variable models for neural data).
- There is a line of work focused on across-session alignment that hasn’t been discussed [1-4]. The authors should more clearly establish the gap and discuss why the proposed work is novel in light of these studies.
- For general data-driven deep learning analysis of neural activities, [5,6] aim to use VAE to learn identifiable latent representation for neural activities, [7] use more recent architecture (transformer) to analyze neural activities.

_4. The description of the methodology is not clear or complete._
- “We adapted GradCAM to generate localization maps (or activation maps) for all four networks.” - Class Activation Mapping is a method that generates activation maps for classification models, as a class label is required when using this method. How is that used for generative networks?
- Section 2.3 described many details of the networks architectures, but also stated “We denote the level-wise residual network as ResNet.” and “We denote the attention-gated ResNet as AGResNet.” Commonly used architectures such as ResNet and AGResNet share generally-agreed architecture details in the machine learning field. The authors should use the commonly agreed architecture detail, and share information about which part is different, instead of using their own definition.
- In section 2.5, the authors trained an autoencoder to re-order the neurons, but the loss function of the autoencoder is just MSE loss (reconstruction loss). How are neurons re-ordered when the model is trained to reconstruct for both populations?
- The authors claim that the CycleGAN is a GAN-based unsupervised learning framework, which is a debatable statement/description. CycleGAN is a GAN-based model that utilizes cycle consistency to form parts of their loss function. When referring to CycleGAN, it is more typical that one specifically refers to the model with their proposed loss function. Thus, it is not clear and not common to plug in different loss functions into GAN objectives and refer to it as CycleGAN variants. The authors should re-word the section and state their loss functions more clearly.

_5. Minor comments and questions_
- Introduction: “A major hurdle in this endeavor was the difficulty in obtaining high-quality neural recordings of the same set of neurons across an extended period of learning a task.” Please consider citing more papers to support your view here.
- The loss function in Table 1 LSGAN is incorrect. There should be a “)” after G(x). It is also not the general LSGAN loss function but a specific version as the a-b coding scheme of the discriminator is set to be 1-1, please add a sentence to clearly note this.
- Methods:  “However, DY can only verify if y \in Y, though cannot ensure that y^ is the corresponding expert activity of the novice recording x.” Depending on the difference between distributions, sometimes it is better to use style transfer techniques instead of image translation techniques if the distribution discrepancy is relatively low. Have the authors considered this? If not, is there a reason?
- Why is the loss for the identity loss objective defined as a MAE loss? Does that encourage the generator to generate a X domain object for a generator that is supposed to generate Y domain object?
- In Table B.1, is LReLu leaky ReLu? Please add either a reference or a piece of introduction.

*References*

[1] Degenhart, Alan D., et al. "Stabilization of a brain–computer interface via the alignment of low-dimensional spaces of neural activity." Nature biomedical engineering 4.7 (2020): 672-685.

[2] Farshchian, Ali, et al. "Adversarial domain adaptation for stable brain-machine interfaces." arXiv preprint arXiv:1810.00045 (2018).

[3] Jude, Justin, et al. "Robust alignment of cross-session recordings of neural population activity by behaviour via unsupervised domain adaptation." arXiv preprint arXiv:2202.06159 (2022).

[4] Gallego, Juan A., et al. "Long-term stability of cortical population dynamics underlying consistent behavior." Nature neuroscience 23.2 (2020): 260-270.

[5] Liu, Ran, et al. "Drop, swap, and generate: A self-supervised approach for generating neural activity." Advances in Neural Information Processing Systems 34 (2021): 10587-10599.

[6] Zhou, Ding, and Xue-Xin Wei. "Learning identifiable and interpretable latent models of high-dimensional neural activity using pi-VAE." Advances in Neural Information Processing Systems 33 (2020): 7234-7247.

[7] Ye, Joel, and Chethan Pandarinath. "Representation learning for neural population activity with Neural Data Transformers." arXiv preprint arXiv:2108.01210 (2021).

[8] Pei, Felix, et al. "Neural Latents Benchmark'21: Evaluating latent variable models of neural population activity." Advances in Neural Information Processing Systems 34 (2021) Track on Datasets and Benchmarks. (https://arxiv.org/pdf/2109.04463.pdf)

---

> ### Author Response · Authors · 2022-11-17
> **Response to Reviewer 4Xms**
>
> We thank the reviewer for their detailed review and helpful suggestions for improvement. We have updated the manuscript to address the issues and concerns raised in the review.
>
> Please find our response to each point of the reviewer’s Weaknesses section below:
> 1. We agree with the reviewer that the individual techniques used in this work are based upon previous works in deep learning and computational neuroscience. However, our work provides a number of novel contributions when using these techniques in combination, including the problem setting where we pose the neuronal learning analysis as an unsupervised machine translation problem, as well as the visual explanation techniques to identify neurons and responses that are of interest in a completely data-driven manner. Moreover, even though it is not the main focus of this work, the proposed neuron ordering methods are general and can be easily adapted to work that involves convolutional-based networks modelling neural data.
> 2. We would like to thank the reviewer for their suggestion on possible baseline models. Based on this comment as well as suggestions by other reviewers, we have introduced two additional baseline models. Briefly, given two distributions X and Y, the (1) $\texttt{Linear}$ model first extracts a low-dimensional representation $\hat{x}$ via PCA with N components that explain 95% of the variance over a population vector (i.e. a single time frame), we then project $\hat{x}$ back to the population dimension of Y using a linear decoder; the (2) convolutional-based vanilla $\texttt{VAE}$ model where the encoder learns the mean and log-variance of the conditional distribution of the latent representation z given x, and the decoder learns to output parameters of Y. Here, the overall optimisation objective is $\mathcal{L}_\texttt{VAE} = \log p (y|z) + \log p(z) - \log q(z | x)$. These two baseline models represent the common linear and non-linear unsupervised approaches. The detailed descriptions of the two baseline models are available in Section 2.10 in the updated manuscript. We fitted both baseline models on the two synthetic datasets with known ground-truth transformation and the V1 recorded dataset. The results are shown in Sections 3.1, 3.2 and 3.3. Overall, the $\texttt{Linear}$ model failed to learn the transformation in all 3 datasets, and in some cases, performed worse than the $\texttt{Identity}$ model. The $\texttt{VAE}$ model, on the other hand, achieved similar performance as the CycleGAN methods in the simulated dataset but was unable to capture the more complex transformations in the augmented and recorded dataset as compared to the models trained in the CycleGAN framework.
> 3. We thank the reviewer for providing the list of publications with similar motivation in neural analysis. We have updated the background section in the Introduction to discuss more recent work in this area, including deep latent variable models, generative adversarial networks and domain adaptation models. In addition, we have added Section 2.1 to formalise our problem setting and clarify our motivation to treat the pre-learning to post-learning analysis task as an unsupervised translation problem which slightly deviates from previous work in this area.
>
> [Continue]

---

> > ### Author Response · Authors · 2022-11-17
> > **Response to Reviewer 4Xms (continue)**
> >
> > 4.
> > - We have added Section 2.7 to briefly describe the GradCAM algorithm and discuss how we adapted the visual explanation technique to our use case. Briefly, GradCAM computes the gradient information flow between logits $y^c$ of class $c$ and the feature map $A^k \in \mathcal{R}^{u \times v}$ of a specific convolutional layer (usually the final layer) with $k$ filters. The gradients are then pooled over the spatial dimensions to calculate neuron importance weights $\alpha_k^c = \frac{1}{N} \sum_i^u \sum_j^v \frac{\partial y^c}{\partial A^k_{i,j}}$ where $N = u \times v$. The GradCAM activation map is the weighted combination of the feature maps followed by ReLU activation to eliminate features with negative influence $M_\text{GradCAM} = \text{ReLU}(\sum_k \alpha^c_k A^k)$. Similar to a sigmoid mask in the AG module (see Section 2.6.1), here, we can overlay the activation map $M_\text{GradCAM}$ on top of the input to visually interpret region(s) that the model is focussing on. We applied the same method on the two discriminators to monitor the final convolution layer, and instead of a scalar prediction $y_c$, we compute the gradient flow between the feature maps of the target layer and the output. In the case of the generators, we instead monitor the gradient between the feature maps of the convolution layer in the last residual block $\texttt{RB}_9$ and the output, where the feature maps should learn the low-dimensional representation of the input after the down-sampling and residual processing block.
> >  - We thank the reviewer for pointing out the potential confusion with commonly used architecture names. We have updated the network architecture section (Section 2.6) to provide a more precise description of the two generator architectures, their exact formulation and their distinction. In addition, to avoid any possible confusion with the naming of the model with established architectures, we have renamed $\texttt{ResNet}$ to $\texttt{Model-R}$, $\texttt{AGResNet}$ to $\texttt{Model-AG}$ and the 1D variant of $\texttt{AGResNet}$ to $\texttt{Model-AG-1D}$.
> >  - We fit the autoencoder $\texttt{AE}$ by minimising the reconstruction loss $\mathcal{L}_\text{AE} = \texttt{MSE}(X, \texttt{AE}(X)) + \texttt{MSE}(Y, \texttt{AE}(Y))$. Then we compute the per-neuron reconstruction error over the validation set of $X$ and $Y$ and sort the neurons in ascending order. That is, we rearrange the order in the neuron (3rd) dimension of the data matrix of shape $(N, H, W, C=1)$, i.e. $H=2048$ and $W=102$ in the recorded dataset. We have also experimented with training separate autoencoders for $X$ and $Y$, which would then sort the neurons in $X$ and $Y$ separately, though the performance gain over a single autoencoder was minimal and we, therefore, present the simpler approach.\
> > 5.
> >  - We have added a number of previous works that discuss the difficulties in obtaining large-scale cross-day neuronal recordings [1,2,3].
> >  - We thank the reviewer for pointing out the typo in the Table and the unique case of LSGAN in this setting. We have updated the Table and the main text to reflect the changes.
> >  - The observed difference between pre- and post-learning activity (c.f. Figure G.1 top) suggests relatively large distribution discrepancies. Therefore, we deemed a simpler style transfer approach is not likely to be successful. We would also like to note that image style transfer is one of the demonstrated tasks in the original CycleGAN paper [4].
> >  - As $G$ learns $X \rightarrow Y$, we expect that $y - G(y)$ should be small given $y \in Y$. Therefore, identity loss was introduced to enforce the generator to apply no transformation to the input if the input is already in its target distribution.
> >  - We thank the reviewer for pointing out the typo in Table B.3. We actually used GELU as the activation function which was reflected in the main text and Figure 5 but we overlooked Table B.3 in the Appendix. We have updated the Table with the exact hyperparameter settings with references.
> >
> > [Continue]

---

> > > ### Author Response · Authors · 2022-11-17
> > > **Response to Reviewer 4Xms (continue)**
> > >
> > > In addition to the number of changes mentioned above, we have also introduced numerous updates to the manuscript which are summarised below, large changes in the main text are written in red font for clarity:
> > > - We formalised the problem setting of this work in Section 2.1 and also explained it in the Introduction.
> > > - We rearranged the Method section by moving the dataset descriptions (Section 2.2 and 2.3) to the beginning of the Method section as subsequent sections referenced to the datasets. Moreover, we rewrote part of the description for the augmented dataset (Section 2.3.2) to clarify its construction.
> > > - We clarified the generator architectures with a more precise description in Section 2.6 and added Section 2.7 to describe the implementation of GradCAM in our work.
> > > - As the comparison of different GAN objectives in the CycleGAN setting is not the main focus of this work, but more of an additional hyperparameter, we have moved the formulation and discussion of the various GAN objectives to Appendix B.
> > > - In the Introduction and Discussion sections, we reiterated the ability to identify trial-relevant information in a fully data-driven manner making this work suitable for large-scale unstructured datasets.
> > >
> > > We again would like to thank the reviewer for their time and comprehensive review.
> > >
> > > [1] Stevenson, Ian H., and Konrad P. Kording. "How advances in neural recording affect data analysis." Nature Neuroscience 14.2 (2011): 139-142.
> > >
> > > [2] Dhawale, Ashesh K., et al. "Automated long-term recording and analysis of neural activity in behaving animals." eLife 6 (2017): e27702.
> > >
> > > [3] Lütcke, Henry, David J. Margolis, and Fritjof Helmchen. "Steady or changing? Long-term monitoring of neuronal population activity." Trends in Neurosciences 36.7 (2013): 375-384.
> > >
> > > [4] Zhu, Jun-Yan, et al. "Unpaired image-to-image translation using cycle-consistent adversarial networks." Proceedings of the IEEE international conference on computer vision. 2017.

---

### Decision · Action_Editors · 2023-02-05

**Recommendation:** Reject

**Comment:**

I could get feedback from only two reviewers among the three. Both reviewers acknowledge that good work has been done in the process. However, the consensus is that the paper does not bring a clear contribution that would make it suitable for publication in TMLR. This judgment mostly comes from the experimental comparisons carried out in the analysis.
While much of the paper is technically sound, the experiments presented contain results that do not make sense (baseline methods performing worse than chance).
As a consequence, the paper does not bring strong enough insights. We suggest that the authors rework their experimental setup, but also clarify the motivations for their contribution, and the justifications for the various choice performed.
The consensus among reviewers is that a fresh submission is more useful than another iteration on the current document.

**Audience:**

As outline above, the main issue with the paper as it is, is its usefulness. It has too many loose ends for readers of this community.

**Claims And Evidence:**

The paper has been found to be too weak. This judgment mostly comes from the experimental comparisons carried out in the analysis.
While much of the paper is technically sound, the experiments presented contain results that do not make sense (baseline methods performing worse than chance).
As a consequence, the paper falls short of demonstrating the proposed approach.